# Alterations of redox and iron metabolism accompany the development of HIV latency

Iart Luca Shytaj[1,2,†,*] (iD), Bojana Lucic[1,2,†], Mattia Forcato[3], Carlotta Penzo[1], James Billingsley[4],
Vibor Laketa[1,2], Steven Bosinger[4,5], Mia Stanic[1], Francesco Gregoretti[6] (iD), Laura Antonelli[6],
Gennaro Oliva[6], Christian K Frese[7], Aleksandra Trifunovic[7] (iD), Bruno Galy[8], Clarissa Eibl[9,10],
Guido Silvestri[4,5], Silvio Bicciato[3] (iD), Andrea Savarino[11] & Marina Lusic[1,2,**] (iD)

## Abstract

HIV-1 persists in a latent form during antiretroviral therapy, mainly in CD4$^+$ T cells, thus hampering efforts for a cure. HIV-1 infection is accompanied by metabolic alterations, such as oxidative stress, but the effect of cellular antioxidant responses on viral replication and latency is unknown. Here, we show that cells survive retroviral replication, both *in vitro* and *in vivo* in SIVmac-infected macaques, by upregulating antioxidant pathways and the intertwined iron import pathway. These changes are associated with remodeling of promyelocytic leukemia protein nuclear bodies (PML NBs), an important constituent of nuclear architecture and a marker of HIV-1 latency. We found that PML NBs are hyper-SUMOylated and that PML protein is degraded via the ubiquitin–proteasome pathway in productively infected cells, before latency establishment and after reactivation. Conversely, normal numbers of PML NBs were restored upon transition to latency or by decreasing oxidative stress or iron content. Our results highlight antioxidant and iron import pathways as determinants of HIV-1 latency and support their pharmacologic inhibition as tools to regulate PML stability and impair latency establishment.

**Keywords** HIV-1 latency; iron; oxidative stress; promyelocytic leukemia protein; proteasome
**Subject Categories** Metabolism; Microbiology, Virology & Host Pathogen Interaction; Molecular Biology of Disease
**The EMBO Journal (2020) 39: e102209**

## Introduction

The activation status of T cells, and thus their metabolic activity, is recognized as primary determinants of HIV-1 latency (Williams & Greene, 2007). Latency is the main obstacle to HIV-1 eradication and is characterized by persistence of integrated viral DNA that is transcriptionally silent, but replication competent. During antiretroviral therapy (ART), integrated viral DNA mainly persists in memory CD4$^+$ T lymphocytes (Chomont *et al*, 2009; Hiener *et al*, 2017) where proviral reservoirs can become transcriptionally activated to trigger systemic infection upon ART discontinuation (Chun *et al*, 1998, 1999). Latency is established early upon infection *in vitro* (Chavez *et al*, 2015) and *in vivo* (Chun *et al*, 1998). Several concepts have been proposed to explain the development of HIV-1 latency and, among them, the reversal of activated cells to a resting state after infection is broadly accepted (Siliciano & Greene, 2011). However, it is still unclear why only a portion of HIV-1-infected cells develop latent infection, while others do not survive the productive infection phase.

Among the different factors involved in the virus-host interplay, oxidative stress and the cellular antioxidant response have been linked to productive HIV-1 infection (Gorrini *et al*, 2013) and to HIV-1 reactivation from latency (Piette & Legrand-Poels, 1994; Bosque & Planelles, 2009; Shytaj *et al*, 2013). Oxidative stress is characterized by the generation and accumulation of byproducts of oxygen metabolism, reactive oxygen species (ROS), which can play a double-faced toxic and/or regulatory role (Benhar *et al*, 2016). To survive and regulate the accumulation of ROS, cells employ a plethora of antioxidant mechanisms. Nuclear

1  Heidelberg University Hospital, Heidelberg, Germany
2  German Center for Infection Research, Heidelberg, Germany
3  Department of Life Sciences, University of Modena and Reggio Emilia, Modena, Italy
4  Division of Microbiology and Immunology, Yerkes National Primate Research Center, Emory University, Atlanta, GA, USA
5  Department of Pathology and Laboratory Medicine, Emory University, Atlanta, GA, USA
6  Institute for High Performance Computing and Networking, ICAR-CNR, Naples, Italy
7  CECAD Research Center, University of Cologne, Cologne, Germany
8  Division of Virus-Associated Carcinogenesis, German Cancer Research Centre, Heidelberg, Germany
9  Leibniz-Forschungsinstitut für Molekulare Pharmakologie, Berlin, Germany
10  Institute of Biology, Cellular Biophysics, Humboldt Universität zu Berlin, Berlin, Germany
11  Italian Institute of Health, Rome, Italy
   *Corresponding author. Tel: +49 6221 564128; E-mail: Luca.Shytaj@med.uni-heidelberg.de
   **Corresponding author. Tel: +49 6221 565007; E-mail: Marina.Lusic@med.uni-heidelberg.de
   †These authors contributed equally to this work

factor erythroid 2-related factor 2 (Nrf2) is the master transcriptional regulator that controls the production of several antioxidant factors, including key redox modulators such as glutathione (GSH), nicotinamide adenine dinucleotide phosphate (NADPH), the thioredoxin/thioredoxin reductase (Trx/TrxR1) axis, the quinone detoxifying agent NAD(P)H dehydrogenase [quinone] 1 (NQO1), and several molecules involved in cellular iron metabolism (Gorrini et al, 2013; Ma, 2013). While therapeutic manipulation of antioxidant responses to complement ART is under pre-clinical and clinical investigation (Benhar et al, 2016; Diaz et al, 2019), only sporadic studies have examined the cellular regulation of antioxidant responses upon HIV-1 infection, reaching opposite conclusions (Zhang et al, 2009; Gill et al, 2014; Furuya et al, 2016). Thus, identifying the antioxidant species and pathways which regulate the response to HIV-1-induced oxidative stress is required to define and characterize their role in the transition from productive to latent infection. Moreover, aside from specialized antioxidant factors, increased ROS levels are known to alter the cellular proteome, mainly through cysteine oxidation (van der Reest et al, 2018). Whether HIV-1-induced oxidative stress can regulate the expression of cellular proteins involved in HIV-1 latency establishment or maintenance is unknown.

Promyelocytic leukemia nuclear bodies (PML NBs) are multimolecular structures formed predominantly by the PML protein. The antiviral activity of the PML protein and PML NBs has been reported for a number of different viruses (Lallemand-Breitenbach & de Thé, 2018; Castro & Lusic, 2019; Torok et al, 2009). In the case of HIV-1, PML was shown to play an important role in the epigenetic control of viral transcription, contributing mainly to latency maintenance through its interaction with histone methyltransferase G9a (Lusic et al, 2013). Moreover, PML is the main therapeutic target of arsenic trioxide (ATO), a drug approved for the treatment of acute promyelocytic leukemia and shown to have anti-HIV-1 latency potential in vitro (Lusic et al, 2013) and in vivo (Yang et al, 2019). ATO can induce oxidative stress, and it has been recently proposed that the oxidation process per se can modulate the biogenesis and turnover of PML NBs through post-translational modifications (Sahin et al, 2014a,b; Niwa-Kawakita et al, 2017). However, the role of oxidative stress on PML stability in the setting of either productive or latent HIV-1 infection has not been explored.

Here, we use omics data from a primary CD4[+] T-cell model and ex vivo transcriptomic profiles derived from rhesus macaques infected with the HIV homolog SIVmac (Micci et al, 2015) to characterize the transition between productive and latent infection. We identify a pathway initiated by HIV-1 replication and oxidative stress prompting increased expression of Nrf2-regulated antioxidant pathways and iron import. We further show that oxidative stress and iron import during productive HIV-1 infection mediate depletion of PML NBs through SUMO modification and ubiquitin-mediated proteasomal degradation. In line with this, latency establishment or blocking HIV-1 replication through ART leads to PML reformation, while reactivation from latency induces PML degradation. The pathway here reconstructed highlights metabolic correlates of latency and shows that Nrf2-signaling, iron import, and PML are intertwined, thus providing potential druggable targets for combined therapeutic applications.

# Results

## Antioxidant response is up-regulated during productive in vitro and in vivo infection

To study the expression of antioxidant genes and proteins during the different stages of HIV-1 infection, we used a primary CD4[+] T-cell model similar to those previously adopted by several groups (Bosque & Planelles, 2009; Lusic et al, 2013; Martins et al, 2016). In this model, CD4[+] T cells are activated through stimulation of CD3/CD28 receptors, infected with HIV-1[NL4-3], and monitored over time until returning to a resting state (Fig EV1A). This time course mimics distinct features of HIV-1 infection in vivo, i.e., a rapid initial growth of viral replication (Fig EV1B and C) accompanied by cell death or establishment of a small pool of cells harboring integrated viral DNA, a fraction of which can be reactivated (Fig EV1D–H). Donor-matched mock-infected controls are used to standardize gene and protein expression levels, while sampling of mRNA and proteins at different time points enables specific analysis of the various infection stages.

In order to elucidate the relationship between viral replication and oxidative stress, we conducted RNA-Seq (Figs 1 and EV2A, and Table EV1) and proteomic analyses (Fig EV2B) of a macro-pathway containing genes ($n = 184$) related to the response to oxidative stress. We observed enriched expression of antioxidant genes and proteins in infected compared to mock-infected control cells. Interestingly, the most significant enrichment was observed at 7 days post-infection (henceforth, dpi) (Fig 1A), corresponding to the peak of HIV-1 replication in our model (Fig EV1B and C). To estimate the biological impact of this enrichment as compared to other cellular pathways modulated by HIV-1, we analyzed all significantly up-regulated genes during infection at 7 dpi using the entire Gene Ontology—Biological Process collection available at MSigDb (containing 4,436 gene sets). The enrichment of the genes included in the antioxidant pathway retained its statistical significance after correction for multiple testing ($q$-val = 1.71E-08; 6[th] percentile of all 4,436 gene sets ranked by $q$-values).

To investigate the in vivo relevance of our findings, we further analyzed the expression of genes involved in oxidative stress response using an RNA-Seq dataset from an animal model closely recapitulating the main features of HIV infection (Evans & Silvestri, 2013; Micci et al, 2015). This dataset is derived from peripheral blood mononuclear cells (PBMCs) of macaques chronically infected with the HIV homolog virus SIVmac in the presence or absence of ART. The use of this animal model provides an optimal in vivo parallel of our in vitro system in which macaques can be standardized for viral inoculum, time/route of infection, and time points of analysis, with each animal acting as its own internal control before ART initiation (Evans & Silvestri, 2013). In agreement with our in vitro data, a Gene Set Enrichment Analysis (GSEA) showed enriched expression of antioxidant genes before administration of ART ($P = 0.035$; Fig 1B).

In conclusion, both in vitro and ex vivo data show that the expression of antioxidant genes is enriched during the productive stage of viral infection.

**A**

**Pathway:**
**GO CELLULAR RESPONSE TO OXIDATIVE STRESS**

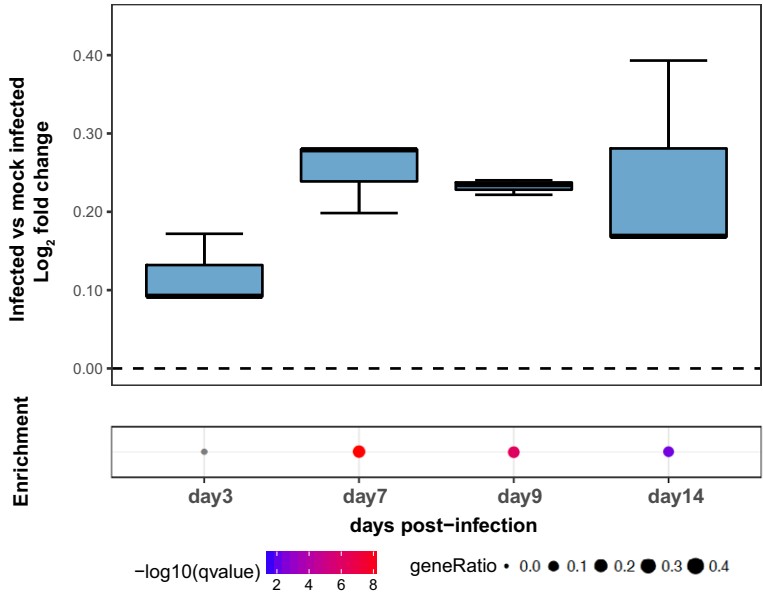

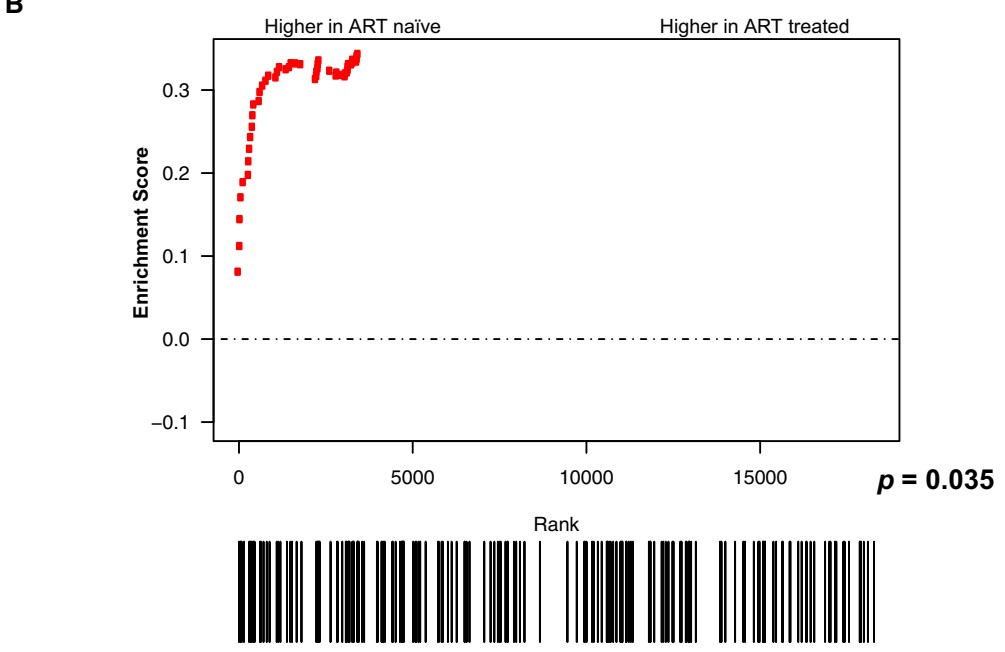

**Figure 1. Viral replication induces upregulation of antioxidant genes *in vitro* and *in vivo*.**

A, B  RNA-Seq analysis of different time points of primary CD4[+] T cells infected *in vitro* with HIV-1 or mock-infected (A) and PBMCs of macaques infected with SIVmac239 before and after suppression of viremia with ART (B). Expression levels were normalized as $\log_2$ fold change in infected vs. matched mock-infected controls (A) or as enrichment score in ART-treated vs. ART-naïve animals (B). For (A), data were analyzed by Fisher test (number of donors = 3 biological replicates). Boxplots in (A) depict median and 25–75 percentiles, while whiskers extend from the hinge to the highest or lowest value that is within 1.5 * IQR (inter-quartile range) of the hinge. Data beyond the end of the whiskers are outliers and plotted as points. For each time point, dots illustrate the pathway enrichment analysis of genes up-regulated in infected vs. matched mock-infected controls. Dots are color-coded based on the enrichment $q$-values, and their size indicates the fraction of differentially expressed genes (DEGs) in the pathway; gray dots are not statistically significant; for (B), data were analyzed by Gene Set Enrichment Analysis (GSEA, number of animals = 8). Symbols in the GSEA enrichment plots represent the position of the gene set members in the transcriptome ranked by differential expression between ART-naïve and ART-treated time points. Red color indicates leading edge genes. All analyses were conducted on the genes of the pathway *GO cellular response to oxidative stress* (184 genes; GO:0034599).

## HIV-1 replication and latency reversal drive activation of Nrf2-regulated antioxidant pathways

The transcriptomic and proteomic profiling of infected CD4$^+$ T cells pointed to an enrichment of antioxidant defenses accompanying the increase in HIV-1 replication (Figs EV2 and 1). In line with this, infected cells exhibited specific markers of redox dysregulation (Figs EV3A and 2A–D). In particular, HIV-1 infection was associated with decreased ratio of total to oxidized glutathione (GSH/GSSG) (Fig 2A) as well as increased ROS (Fig 2B–D) and phosphorylated eukaryotic initiation factor 2 alpha (eIF2$\alpha$, Fig EV3A) content, three well-described markers of oxidative stress (Pace & Leaf, 1995; Giustarini *et al*, 2013; Cnop *et al*, 2017). Moreover, increased ROS content could be highlighted in cells harboring HIV-1 DNA during productive, but not latent, infection corroborating HIV-1 replication as the driving cause of oxidative stress generation (Fig 2D).

To reconstruct the molecular steps leading to the development of antioxidant responses upon HIV-1 infection, we focused on the master transcription factor Nrf2. We found that productive HIV-1 infection was accompanied by Nrf2 nuclear translocation (Fig 2E), a required step for the activation of this transcription factor (Ma, 2013). In line with this, single-molecule HIV-1 RNA FISH combined with immunofluorescence for Nrf2 showed that the nuclear content of Nrf2 was higher in cells actively producing the virus (Fig 2F). Furthermore, knock-down of Nrf2 expression in HIV-1-infected J-Tag T cells was associated with increased *gag* mRNA levels (Fig EV3B and C), suggesting an inhibitory role of Nrf2 on HIV-1 transcription as previously shown (Zhang *et al*, 2009).

We next analyzed the expression of six representative Nrf2-target genes involved in the utilization, synthesis, or detoxification of redox species such as GSH, NAPDH, thioredoxin (Trx), iron, and quinones (Gorrini *et al*, 2013). All Nrf2 targets considered were upregulated both at the transcriptional (Figs 2G and EV3D) and translational (Fig 2H) levels. In all cases, a trend toward normalization to mock-infected control levels was observed upon establishment of latency (Figs 2G and H, and EV3D). Upregulation of the Nrf2 targets was highly dependent on the time post-infection ($P < 0.0001$), irrespective of the gene analyzed or of the housekeeping control used for data normalization (Figs 2G and EV3D). Moreover, when gene expression was assayed after enriching for viable cells (Fig EV3E and F), the enhanced antioxidant responses upon infection were confirmed. These results suggest that oxidative stress generated by HIV-1 replication is accompanied by a broad upregulation of Nrf2-regulated pathways.

We next analyzed whether antioxidant gene upregulation could also characterize HIV-1 reactivation from latency. As only a small fraction of primary cells in our model harbors reactivable HIV-1 (Fig EV1), we employed the J-Lat T-cell line, i.e., a widely adopted latency model in which each cell harbors an integrated HIV-1 DNA copy (Jordan *et al*, 2003). To potently induce latency reactivation, we stimulated cells with tetradecanoylphorbol-13-acetate (TPA) (Jordan *et al*, 2003). We observed significant upregulation of Nrf2 downstream antioxidant genes (Fig 2I), with a similar pattern to that displayed by primary cells during productive infection (Fig 2G). In line with a specific effect of HIV-1 reactivation, TPA treatment of uninfected Jurkat T cells did not significantly alter the expression of the same genes (Fig EV3G).

Overall, these data prove that HIV-1 replication induces oxidative stress, followed by the activation of Nrf2 and upregulation of its downstream pathways. These effects are partially reversed during latency, while they can be induced upon viral reactivation.

## HIV-1-induced upregulation of Nrf2 targets is associated with increased iron import capacity

Intracellular iron can play a pro-oxidant role in cells, and its metabolism is directly controlled by Nrf2 (Gorrini *et al*, 2013). Our data highlighted heme oxygenase-1 (HMOX-1), the enzyme responsible for the catabolism of iron-containing heme (Gozzelino *et al*, 2010), as a highly up-regulated antioxidant factor during HIV-1 replication and latency reactivation (Figs 2G–I, and EV3D and F). Indeed, silencing HMOX-1 expression in the J-Tag T-cell line (Fig EV4A)

---

**Figure 2. Activation of Nrf2 drives antioxidant responses to HIV-1-induced oxidative stress.**

A Total-to-oxidized glutathione ratio (GSH/GSSG) in mock-infected or HIV-1-infected CD4$^+$ T cells (7 dpi, $n = 4$ biological replicates; mean $\pm$ SEM).

B–D Representative picture (B) and quantification (C, D) of ROS content in HIV-1-infected or mock-infected primary CD4$^+$ T cells at 7 dpi (productive infection) and 14 dpi (latent infection). ROS content was measured by immunofluorescence; HIV-1 DNA$^+$ cells were identified by FISH (D). The black dash in (C) indicates the median. Box plots in (D) depict median and 25–75 percentile, while whiskers extend from min to max. $n$ = number of cells. Scale bar = 2 $\mu$m.

E Subcellular localization of Nrf2 in CD4$^+$ T cells infected with HIV-1 or mock infected (3 dpi) as analyzed by biochemical fractionation and Western blot, whole cell extract (WCE), cytoplasm (Cyt), nucleoplasm (Nuc).

F Nuclear localization (left) and content (right) of Nrf2 in HIV-1 RNA$^+$ and HIV-1 RNA$^-$ cells (7 dpi) as measured by combining immunofluorescence and HIV-1 RNA FISH. Black dash indicates median, $n$ = number of cells. Scale bar = 2 $\mu$m.

G, H Time course of the mRNA (G) and protein (H) levels of main downstream antioxidant targets of Nrf2 during the transition from productive (3–9 dpi) to latent (14 dpi) infection, as measured by qPCR (G) and Western blot (H). Data in (G) are depicted as mean $\pm$ SEM of 3 biological replicates. GCLC and HMOX-1 in (H) were probed upon membrane stripping.

I Relative mRNA expression of Nrf2 downstream antioxidant targets in J-Lat 9.2 cells left untreated (latent) or treated for 24 h with 10 $\mu$M TPA (reactivated). Data are depicted as mean $\pm$ SEM of 3 biological replicates.

Data information: Values shown in (C, D and F) were calculated as nuclear corrected total cell fluorescence (as in McCloy *et al*, 2014) and analyzed by Kruskal–Wallis multiple comparison test followed by Dunn's post-test (C) or by two-tailed Wilcoxon signed-rank test (D, F). For (G, I), raw data were first normalized using 18S as housekeeping control and then expressed as log$_2$ fold mRNA expression in infected vs. mock-infected cells (G), or TPA-reactivated vs. latent cells (I) which were calculated as in Livak and Schmittgen (2001). Data were analyzed by two-way ANOVA followed by Tukey's post-test for multiple comparisons. *$P < 0.5$; **$P < 0.01$; ***$P < 0.001$; ****$P < 0.0001$. Trx = thioredoxin; NQO1 = NAD(P)H [quinone] dehydrogenase 1; HMOX-1 = heme oxygenase 1; G6PD = glucose-6-phosphate dehydrogenase; GCLC = glutamate-cysteine ligase; TrxR1 = thioredoxin reductase 1.

Source data are available online for this figure.

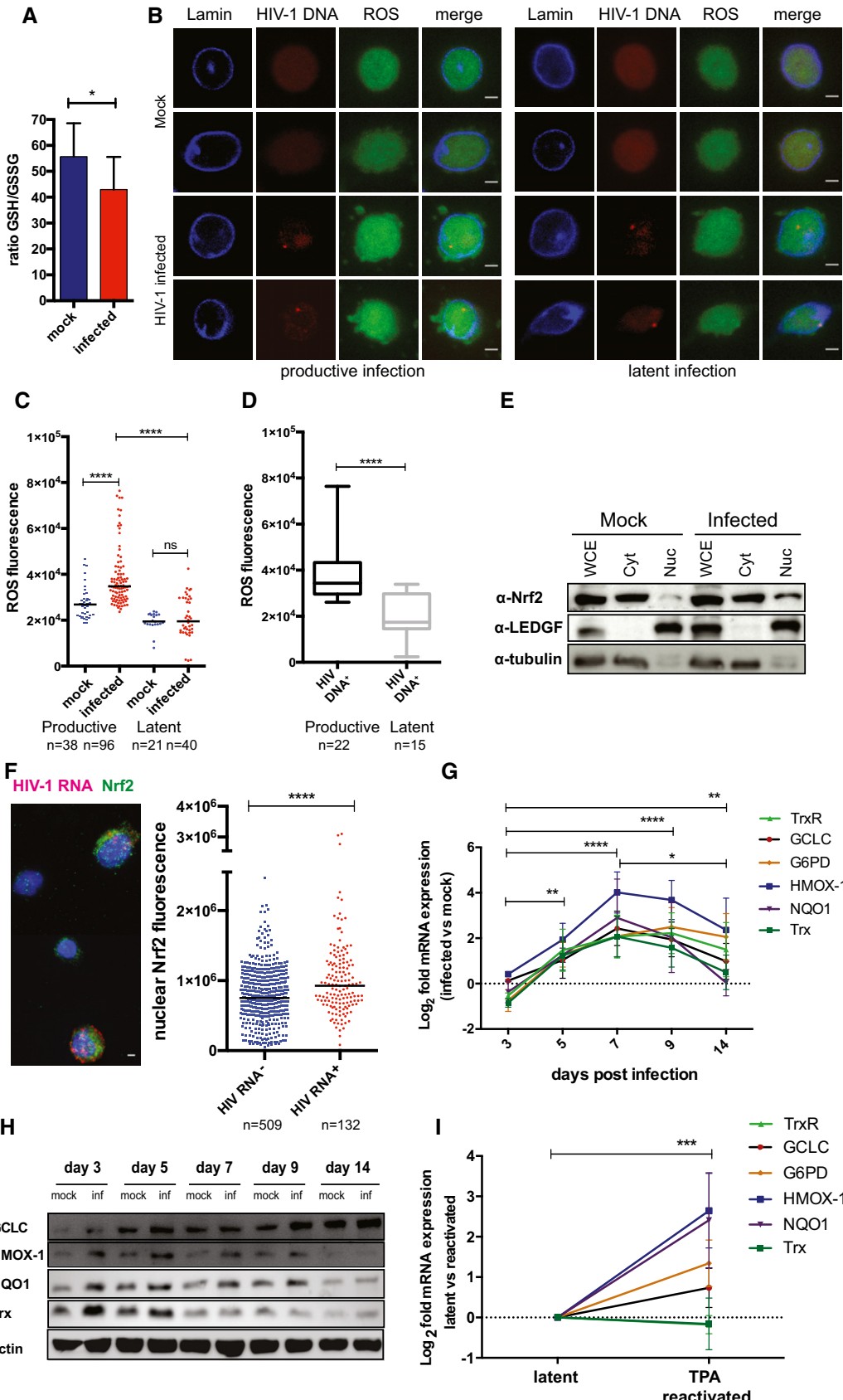

Figure 2.

upon HIV-1 infection led to increased viral transcription and mortality of infected cells (Fig 3A), suggesting that HMOX-1 upregulation plays a pro-survival role.

To further explore the role of iron metabolism regulation during the different infection stages, we analyzed the expression of a pathway comprising the main iron homeostasis genes (Fig EV4B–E and Table EV2). RNA-Seq and proteomic data showed a trend toward enriched expression of mRNA (Fig EV4B and C) and proteins (Fig EV4D) regulating iron metabolism during infection (RNA-Seq $q$-value for enriched genes at 7 dpi after correction for multiple testing = 0.07; Fig EV4B). The *ex vivo* RNA-Seq data derived from SIV-infected macaques further indicated significant enrichment of iron homeostasis genes after ART administration ($P = 0.008$; Fig EV4E).

As iron homeostasis can be regulated by mutually counteracting mechanisms, such as import and export, (Muckenthaler *et al*, 2017), we narrowed our analysis to iron import regulation. Both *in vitro* (Fig 3B and C, and Table EV3) and *ex vivo* (Fig EV4F and Table EV3) data indicated enriched expression of iron import genes toward the transition to latency or after ART administration [*in vitro* data $q$-value for enriched genes at 9 dpi = 0.027 (Fig 3B); *ex vivo* GSEA $P$-value = 0.035; Fig EV4F)]. In addition, the two iron-transport proteins that could be detected in our proteomic analysis (transferrin receptor 1 (TfR1) and phosphatidylinositol binding clathrin assembly protein (PICALM)) showed a similar trend (Fig EV4G and Table EV4). TfR1 was the iron import marker that could be readily detected in all omics datasets. We thus verified its expression independently using immunofluorescence and flow cytometry (Figs 3D and EV4H). We observed an early decrease in TfR1 expression in infected cells at 3 dpi (Fig 3D). As viral replication progressed (7–9 dpi), an opposite pattern was observed (Figs 3D and EV4H), showing enhanced iron import capacity via TfR1 upregulation (Figs 3D and EV4H). Although higher TfR1 expression in infected cells persisted upon latency establishment, absolute levels of this marker were decreased

(14 dpi, Fig 4D), in line with the low expression of TfR1 in resting lymphocytes (Fig EV4I). Apart from TfR1 upregulation, productively infected cells were characterized by significantly decreased expression of FTH-1 (Fig 3E). As FTH-1 binds iron converting it to a nonreactive form (Muckenthaler *et al*, 2017), its downregulation suggests reduced intracellular storage. This reduced storage, however, was also accompanied by lower expression of SLC40A1, suggesting decreased iron export (Fig EV4J).

Overall, the expression of these markers suggests enhanced iron import capacity at peak productive infection and utilization of iron by replicating HIV-1. In line with this hypothesis, long-term incubation with low concentrations (Fig EV4K) of the iron chelator deferiprone (L1) increased mortality of HIV-1-infected, but not mock-infected cells (Fig EV4L). Previous studies have demonstrated that iron can increase HIV-1 replication (Chang *et al*, 2015), likely by enhancing viral transcription (Debebe *et al*, 2007; Xu *et al*, 2010) or reverse transcription (van Asbeck *et al*, 2001; Traoré & Meyer, 2004). Indeed, administration of nontoxic concentrations (Fig EV4K) of the iron donor ferric nitriloacetate (Fe-NTA) was associated with increased frequency of HIV-1 gag p24$^+$ cells (Fig 3F), which was more pronounced when the initial percentage of p24$^+$ cells was higher (Fig 3F). Consistently, treatment with Fe-NTA displayed only minor effects in reactivating HIV-1 expression in two different J-Lat clones but its co-administration with TPA significantly enhanced viral reactivation (Fig 3G).

Overall, these data show that iron can be exploited as a co-factor by replicating HIV-1 and that iron import capacity is increased during productive infection, possibly facilitating cell survival before latency establishment.

### Oxidative stress leads to depletion of the latency marker PML during productive HIV-1 infection

The induction of oxidative stress and iron import can alter the function and stability of metal-containing proteins (Imlay, 2014).

---

**Figure 3. HIV-1 replication alters iron metabolism and enhances iron import during transition to latency.**

A   *gag* mRNA expression (left) and relative viability (right) of HIV-1-infected Jurkat-TAg cells transfected with non-targeting siRNA (NC) or siRNAs targeting the *HMOX-1* gene. Cells were infected 24 h post-transfection and assayed for mRNA expression (by qPCR) or viability (by MTT assay) 48 h post-infection. Data (mean ± SEM; $n = 3$ technical replicates) were normalized using the NC control and analyzed by unpaired $t$-test. *$P < 0.05$; **$P < 0.01$.

B, C   RNA-Seq analysis of different time points of primary CD4$^+$ T cells infected *in vitro* with HIV-1 or mock infected. The gene set used for the analyses is *GO iron ion import* (12 genes, GO:0097286). Data in (B) were analyzed by Fisher test. Boxplots depict median and 25–75 percentiles, while whiskers extend from the hinge to the highest or lowest value that is within 1.5 * IQR (inter-quartile range) of the hinge. Data beyond the end of the whiskers are outliers and plotted as points. Dots below boxplots illustrate, for each time point, the pathway enrichment analysis of genes up-regulated in infected vs. matched mock-infected controls. Dots are color-coded based on the enrichment $q$-values, and their size indicates the fraction of DEGs in the pathway; gray dots are not statistically significant. (C) Heatmaps of the standardized expression of the *GO iron ion import* pathway genes in HIV-1-infected and mock-infected samples over time. Expression levels were standardized [(mean gene expression − SD)/SD] for each gene in each time point. The TFRC (TfR1) gene, which was considered for further analysis, is highlighted.

D   TfR1 expression in HIV-1-infected or mock-infected CD4$^+$ T cells as visualized by confocal microscopy. Fluorescent values shown were calculated as described in McCloy *et al* (2014). Data were analyzed by one-way ANOVA followed by Sidak's post-test to compare corresponding mock-infected vs. infected pairs. Black dash indicates median, $n$ = number of cells. Scale bar = 2 μm.

E   Western blot (upper panel) and relative expression (lower panel) of FTH-1 protein in HIV-1-infected vs. mock-infected CD4$^+$ T cells over time. Western blot quantification was performed with Fiji-Image J (Schindelin *et al*, 2012), normalized to the housekeeping protein beta-actin and expressed as log$_2$ fold change in HIV-1-infected vs. mock-infected cells (mean ± SEM; $n = 3$ biological replicates). $P < 0.05$.

F   Percentage of intracellular p24$^+$ gag CD4$^+$ T cells infected with HIV-1 and left untreated or treated for 48 h with the iron donor Fe-NTA at 150 μM concentration ($n = 3$; mean ± range of three biological replicates).

G   Percentage of GFP$^+$ J-Lat cells (clones 9.2 and 15.4) left untreated or treated for 48 h with TPA (10 μM), the iron donor FeNT (150 μM), or a combination of the two. Data are expressed as mean ± SEM of four technical replicates for each cell line and were analyzed by one-way ANOVA followed by Tukey's post-test. *$P < 0.05$; ***$P < 0.001$; ****$P < 0.0001$.

Source data are available online for this figure.

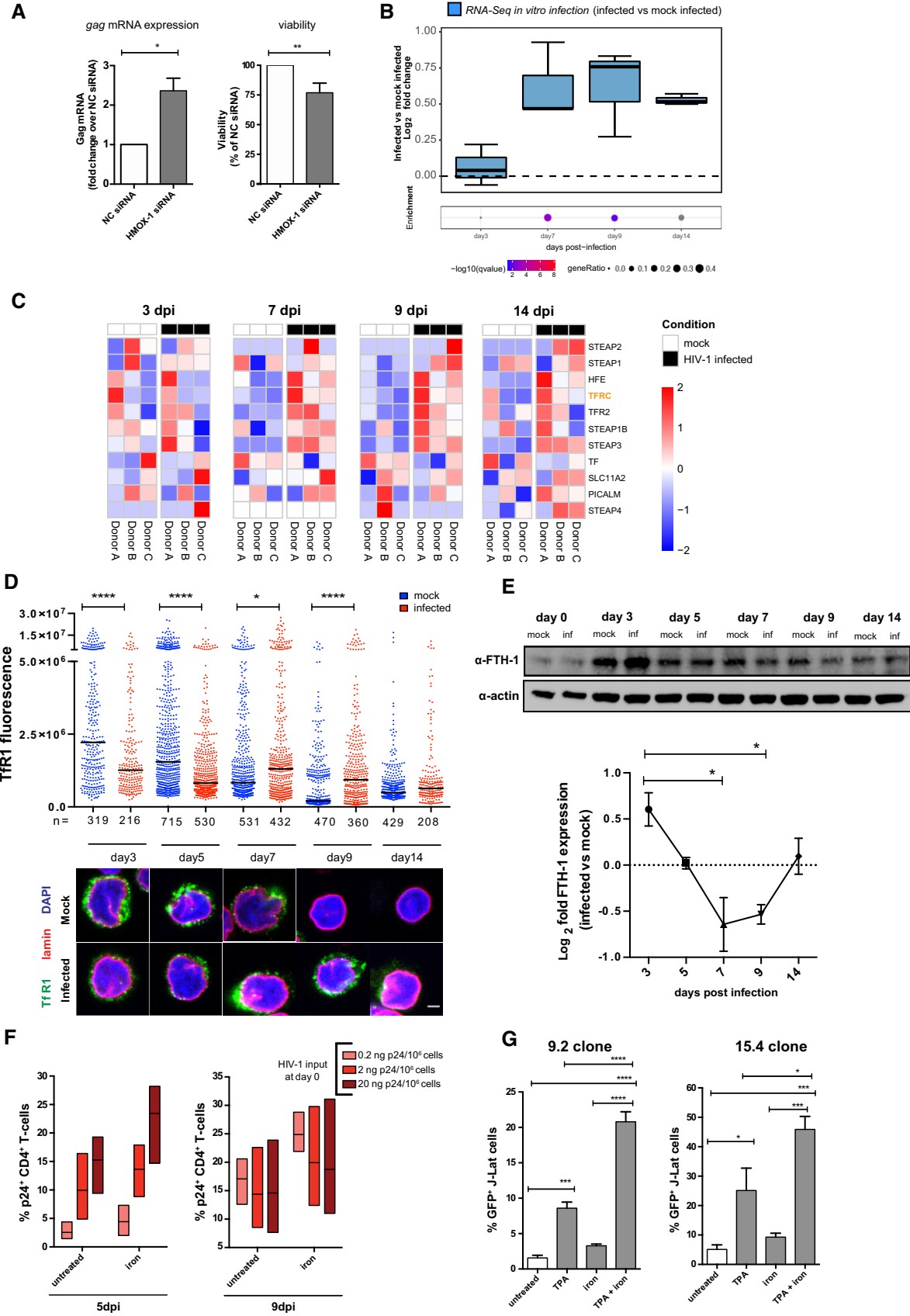

**Figure 3.**

Among these, PML is a potential target of interest as it has been described to colocalize with latent HIV-1 DNA (Lusic *et al*, 2013) and to be redox-sensitive (Sahin *et al*, 2014b; Niwa-Kawakita *et al*, 2017).

The use of drugs increasing oxidative stress (Fig EV5A–C) or iron content (Fig EV5D–F) resulted in depletion of PML and PML NBs in uninfected primary CD4$^+$ T cells. The effect of iron was, however, less pronounced than that of ATO, which can target PML by direct binding and destabilization (Zhang *et al*, 2010; Fig EV5D–F). This evidence suggests an indirect mechanism of action of iron, as previously shown with ROS (Sahin *et al*, 2014b; Niwa-Kawakita *et al*, 2017) and in line with *in silico* analysis predicting no destabilization of PML upon iron binding (Fig EV5G).

PML NBs have been previously described as possible restriction factors of HIV-1 (Turelli *et al*, 2001; Dutrieux *et al*, 2015; Kahle *et al*, 2015). We thus analyzed the proportion of productively infected cells in our primary CD4$^+$ T-cell model where PML was silenced with the recently developed FANA antisense oligo technology (Souleimanian *et al*, 2012). Albeit full silencing could not be achieved in primary cells, only FANA oligos inducing partial depletion of PML were associated with higher proportion of p24$^+$ T cells in the culture (Appendix Fig S1A–C).

We then explored the possible influence of HIV-1 infection in modulating PML expression. We first analyzed the regulation of the PML pathway (comprising PML and the main proteins recruited in the NBs) during the different stages of the infection (Fig 4A and B, Appendix Fig S1D and Table EV5). Transcriptional enrichment of the PML pathway was observed in infected cells, peaking during productive infection (Fig 4A; *q*-value for enriched genes at 7 dpi after correction for multiple testing = 0.0007; first percentile of 217 pathways in the Biocarta pathway collection). A similar trend was observed in *ex vivo* samples from SIVmac-infected macaques (albeit without reaching statistical significance, $P = 0.17$; Appendix Fig S1E). Conversely, protein levels displayed the highest enrichment in infected cells at the latent time point (Fig 4B).

A mirroring trend between protein and mRNA expression was also observed when PML was analyzed separately by Western blot and qPCR (Fig 4C and D). In particular, the data showed a progressive decrease in the amount of PML protein alongside the increase in viral replication (5–9 dpi), followed by replenishment of PML upon transition to latency (14 dpi) (Fig 4C). This pattern was confirmed also when assaying protein extracts obtained from cells previously enriched for their viability (Appendix Fig S2A). The relative level of PML mRNA displayed an opposite trend to protein expression, irrespective of the housekeeping gene used for normalization (Fig 4D), suggesting a compensatory increase to counteract ongoing protein degradation. To investigate causality between HIV-1 replication and PML depletion, we suppressed viral production using ART (5 dpi). The addition of ART abrogated HIV-1 replication in less than 48 h (data not shown) and restored PML protein to mock-infected control levels (Appendix Fig S2B and C). Conversely, PML depletion could be induced by reactivating latent HIV-1 transcription in two different clones of J-Lat cells (Fig 4E).

Finally, to assess the impact of HIV-1-induced oxidative stress/iron import on PML depletion, we incubated mock-infected and HIV-1-infected cells with either NAC, to reduce ROS content, or L1, to chelate iron (Fig 4F and G). Both antioxidant treatment and iron chelation were able to restore PML content to levels comparable to mock-infected controls.

On the whole, these results prove that PML protein is depleted through oxidative stress induced by replicating HIV-1 and reformed upon latency establishment.

### PML is preferentially depleted in cells harboring actively transcribing HIV-1

Oxidative stress can be associated with bystander effects (Klammer *et al*, 2015). We combined HIV-1 DNA FISH with immunofluorescence for PML NBs to specifically investigate PML NB content in cells harboring the virus (Fig 5). Our analysis highlighted a decrease in PML NBs during productive HIV-1 infection as well as its reformation in cells harboring latent HIV-1 (Fig 5A). Furthermore, single-cell HIV-1 RNA FISH and immunofluorescence staining for PML NBs proved that CD4$^+$ T cells infected with actively transcribing HIV-1 are characterized by lower numbers of PML NBs within the same cell culture (Fig 5B). In this regard, identifying 3D objects, such as PML bodies, by projecting *z*-stacks to 2D can potentially bias the counting of such objects. Hence, we used an algorithm (described in the Materials and Methods section) that automatically generates 3D reconstructions of the nuclei and

---

**Figure 4. HIV-1 replication drives depletion of the redox-sensitive marker PML.**

A, B  RNA-Seq (A) or proteomic (B) analyses of different time points in primary CD4$^+$ T cells infected *in vitro* with HIV-1 or mock infected. Data were analyzed by Fisher test (number of donors = 3 biological replicates). Boxplots depict median and 25–75 percentiles, while whiskers extend from the hinge to the highest or lowest value that is within 1.5 * IQR (inter-quartile range) of the hinge. Data beyond the end of the whiskers are outliers and plotted as points. For each time point, dots below boxplots illustrate the pathway enrichment analysis of genes or proteins up-regulated in infected vs. matched mock-infected controls. Dots are color-coded based on the enrichment *q*-values, and their size indicates the fraction of differentially expressed genes or proteins in the pathway; gray dots are not statistically significant. The gene set considered for the analyses is *Biocarta_PML_pathway* (17 genes, M4891).

C, D  Western blot (C) and relative expression (D) of PML protein and mRNA in HIV-1-infected vs. mock-infected CD4$^+$ T cells over time. Raw data were normalized using actin (protein) or GAPDH or 18S (mRNA) as housekeeping control and expressed as log$_2$ fold change expression in infected vs. mock-infected cells. Data were analyzed by two-way ANOVA followed by Tukey's post-test (mean $\pm$ SEM of 3 and 4 biological replicates for PML mRNA and PML protein, respectively).

E  Western blot and quantification of PML protein in two different clones of J-Lat cells left untreated or treated with TPA for 24 h. Data were analyzed by unpaired *t*-test (mean $\pm$ SEM; $n = 3$ technical replicates for each cell line).

F, G  PML protein expression in CD4$^+$ T cells infected with HIV-1 or mock infected at 7 dpi as measured by Western blot. Cells were left untreated or treated with the antioxidant compound NAC (10 mM, F) or the iron chelator deferiprone (L1, 50 µM, G).

Data information: *$P < 0.05$; **$P < 0.01$.
Source data are available online for this figure.

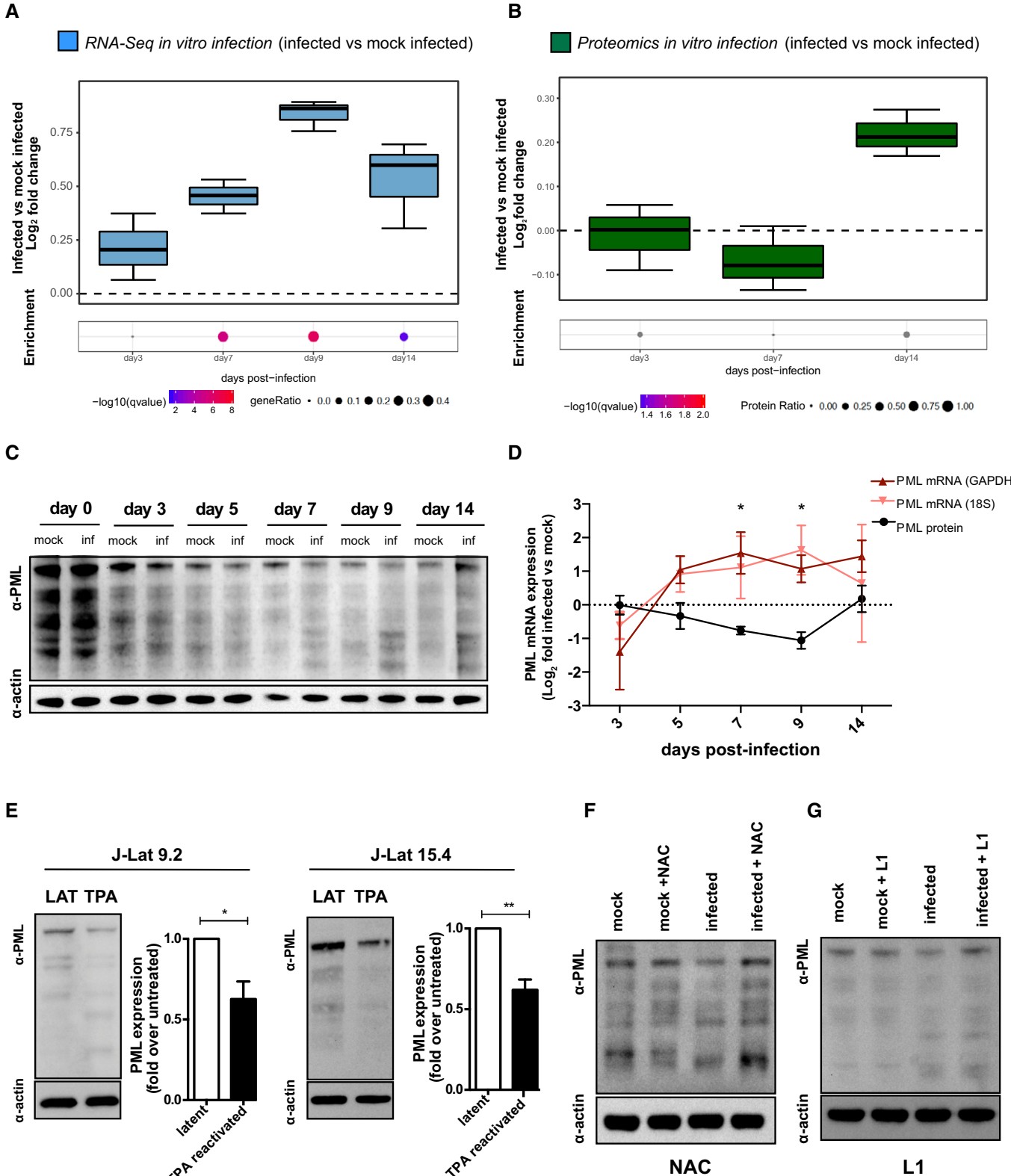

**Figure 4.**

identifies HIV-1 RNA and PML NB signals. The results confirmed significantly lower numbers of PML NBs in cells in which HIV-1 RNA could be detected (Fig 5C).

These data prove that PML is preferentially depleted in cells characterized by ongoing viral replication and is reformed in cells harboring latent HIV-1 DNA.

## PML degradation is associated with nuclear body SUMOylation and mediated by the ubiquitin–proteasome axis

The PML protein has been described as a substrate for degradation by the ubiquitin–proteasome pathway during the response to oxidative stress (Sahin *et al*, 2014b). We tested this hypothesis in the setting of HIV-1 infection by using compounds to inhibit two main steps of the pathway (Appendix Fig S3A). To block proteasome function, we employed MG132 and bortezomib, two well-characterized inhibitors (Goldberg, 2012). Immunofluorescence analysis for PML NBs showed that both proteasome inhibitors could rescue PML expression in productively infected primary CD4$^+$ T cells (Appendix Fig S3B and C), as well as in J-Lat 9.2 cells in which HIV-1 expression was reactivated (Fig 6A and B). In line with this, inhibiting proteasome function before latency reactivation with TPA decreased GFP protein expression and *gag* transcription (Appendix Fig S3D and E).

The recognition of protein substrates by the proteasome system involves several steps of ubiquitination, starting with the activation of ubiquitin by the Ub-activating enzyme (UAE). In line with this, blocking ubiquitination with the selective UAE inhibitor TAK243 (Appendix Fig S3A; Hyer *et al*, 2018) counteracted PML NB degradation after latency reactivation in J-Lat 9.2 cells (Fig 6C and D).

Finally, we tested the colocalization of PML with Small Ubiquitin-like Modifier 2/3 (SUMO 2/3), a post-translational modification that can prelude to proteasomal degradation (Becker *et al*, 2013). By combining STED and confocal microscopy, we could show that, upon latency reactivation, SUMO 2/3 colocalizes with PML (Fig 6E) and its expression is increased in PML NBs (Fig 6F). Consistently, SUMO 2/3 expression was also higher in productively infected primary CD4$^+$ T cells (Fig 6G and H).

Taken together, our results show that replication of HIV-1 induces SUMOylation of PML NBs and degradation of PML through the ubiquitin–proteasome axis.

## Discussion

The pathway here reconstructed characterizes the interplay between oxidative stress, PML NBs, and iron metabolism during all main stages of HIV-1 infection (Fig 6I). Our results are consistent with a model in which T-cell activation allows efficient HIV-1 infection and replication, leading to increased ROS generation. Oxidative stress in turn causes PML degradation, potentially favoring further viral production. At the same time, PML depletion is known to increase nuclear translocation and stability of the antioxidant master transcription factor Nrf2, thus leading to the activation of its downstream targets (Guo *et al*, 2014; Niwa-Kawakita *et al*, 2017). This

concept is supported by the opposite expression trends of PML and the Nrf2 targets that we observed. Upon latency establishment, ROS content and antioxidant responses are decreased, albeit full normalization of the latter is not achieved. This is in line with a previous study showing upregulation of antioxidant defenses in latently infected cell lines, as compared to their uninfected counterparts (Bhaskar *et al*, 2015).

While the interdependence between redox and iron metabolism is well established (Kerins & Ooi, 2018), we here show for the first time that iron can play a key role in mediating PML depletion during productive HIV-1 infection. According to our data, cells surviving the early phase of the infection are characterized by increased iron import capacity. It is tempting to speculate that upregulation of TfR1 selects cells that can survive the increased iron consumption imposed by the infection and therefore develop latency. This hypothesis is in line with our results showing cytotoxicity of prolonged iron chelation during infection, and with a previous study conducted in tumor cell lines, showing that downregulation of TfR1 is associated with death of HIV-infected cells (Savarino *et al*, 1999). In this regard, increased iron utilization and import might be necessary for infected cells to sustain both cellular and viral replication through the role of iron as co-factor of ribonucleotide reductase, as previously suggested (Drakesmith & Prentice, 2008). On the other hand, imported iron can increase ROS content through the Fenton reaction (Fenton Chemistry, 2006), thus indirectly favoring PML depletion. In particular, oxidation of disulfide bonds, formed in response to ROS, is required for PML SUMOylation and proteasomal degradation (Sahin *et al*, 2014b). Accordingly, our data show that PML depletion upon HIV-1 replication is accompanied by nuclear body SUMOylation and degradation through the ubiquitin–proteasome pathway. The iron-containing heme is a potent pro-oxidant known to play an important role in disulfide bond formation. Heme is also the main target of HMOX-1 (Gozzelino *et al*, 2010), an antioxidant factor which was highly up-regulated in response to HIV-1 replication in our experiments. Intriguingly, the PML-depleting drug ATO is known to induce upregulation of HMOX-1 (Meyer *et al*, 2018), further suggesting the possibility that HMOX-1 expression might respond to lowered PML levels.

Despite the possibility that iron may directly bind PML, our molecular modeling does not predict this binding to be destabilizing, unlike that of arsenic (Zhang *et al*, 2010). However, the spatial proximity of iron to PML might increase the susceptibility of this protein to ROS, as previously described for ATO (Zhou *et al*, 2015). As PML is a relevant therapeutic target *per se*, further studies on this topic may be warranted.

The pattern of increased oxidative stress and iron import induced by HIV-1 replication was reversed upon latency establishment, leading to PML reformation. Accordingly, iron chelation was previously

---

**Figure 5. PML is preferentially depleted in cells with detectable HIV-1 mRNA.**

A, B   Representative image and quantification of the number of PML NBs in HIV-1 DNA$^+$ vs. mock-infected (A) and HIV-1 RNA$^+$ vs. HIV-1 RNA$^-$ (B) CD4$^+$ T cells. Scale bars = 2 μm.

C   Representative 3D reconstruction and quantification of the number of PML NBs in HIV-1 RNA$^+$ vs. HIV-1 RNA$^-$ CD4$^+$ T cells. *n* = number of cells.

Data information: Infected cells were identified with either DNA (A) or RNA (B, C) FISH, and PML NBs were stained by IF. The algorithm used for automatic PML counting and 3D reconstruction is detailed in the Materials and Methods section. Data are depicted as mean ± SEM and were analyzed by unpaired *t*-test. **$P < 0.01$; ***$P < 0.001$; ****$P < 0.0001$.

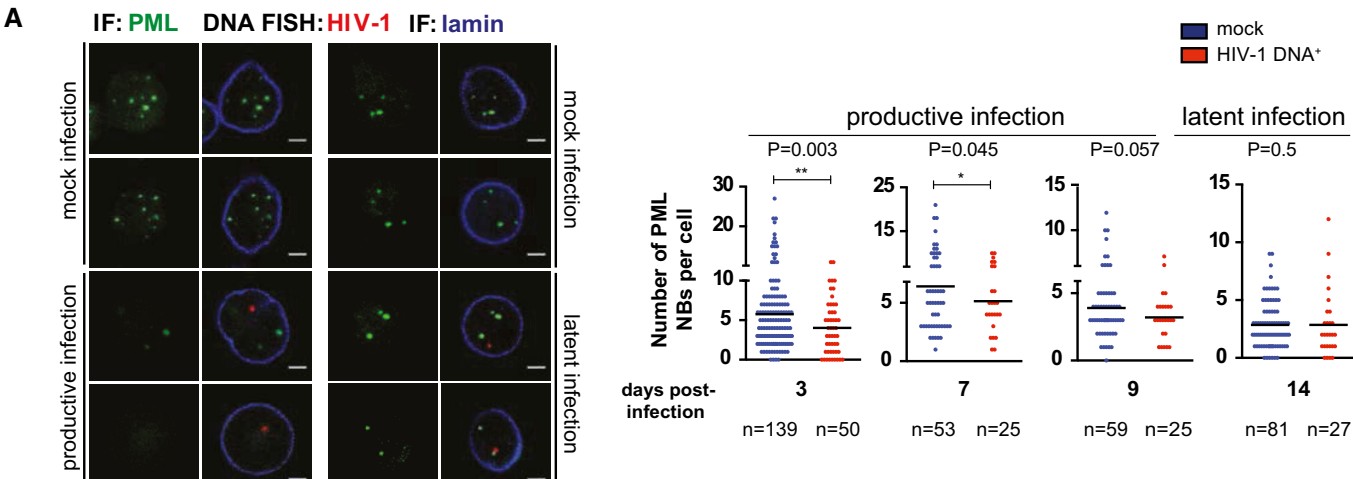

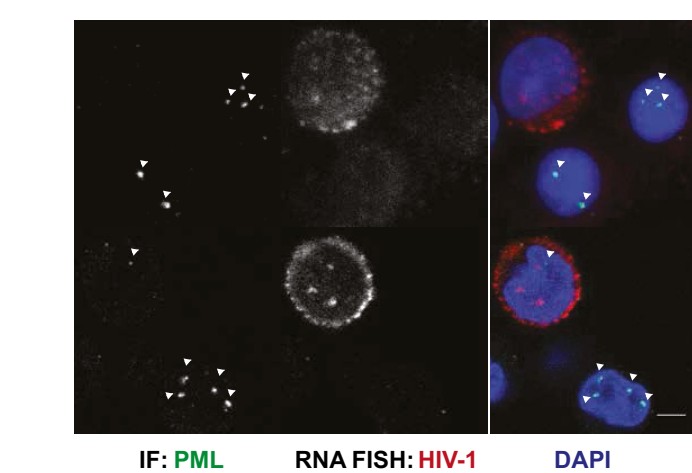

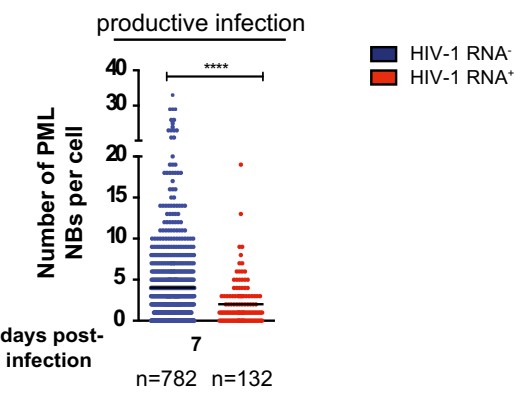

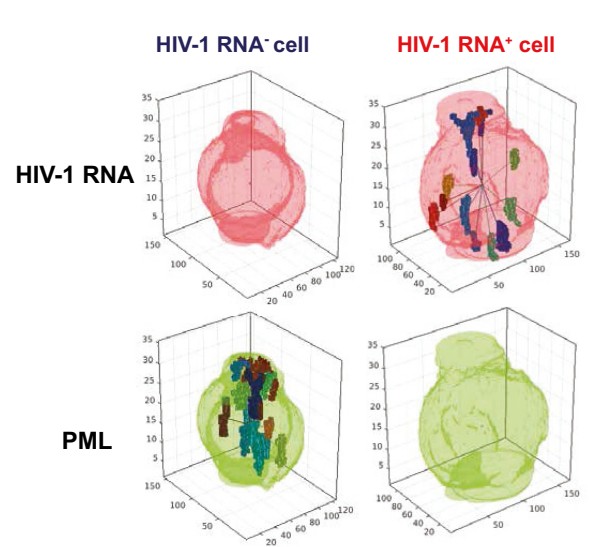

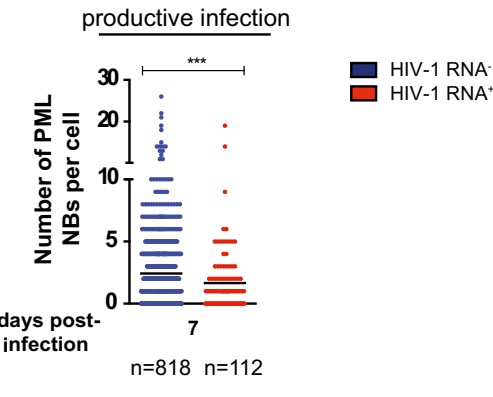

**Figure 5.**

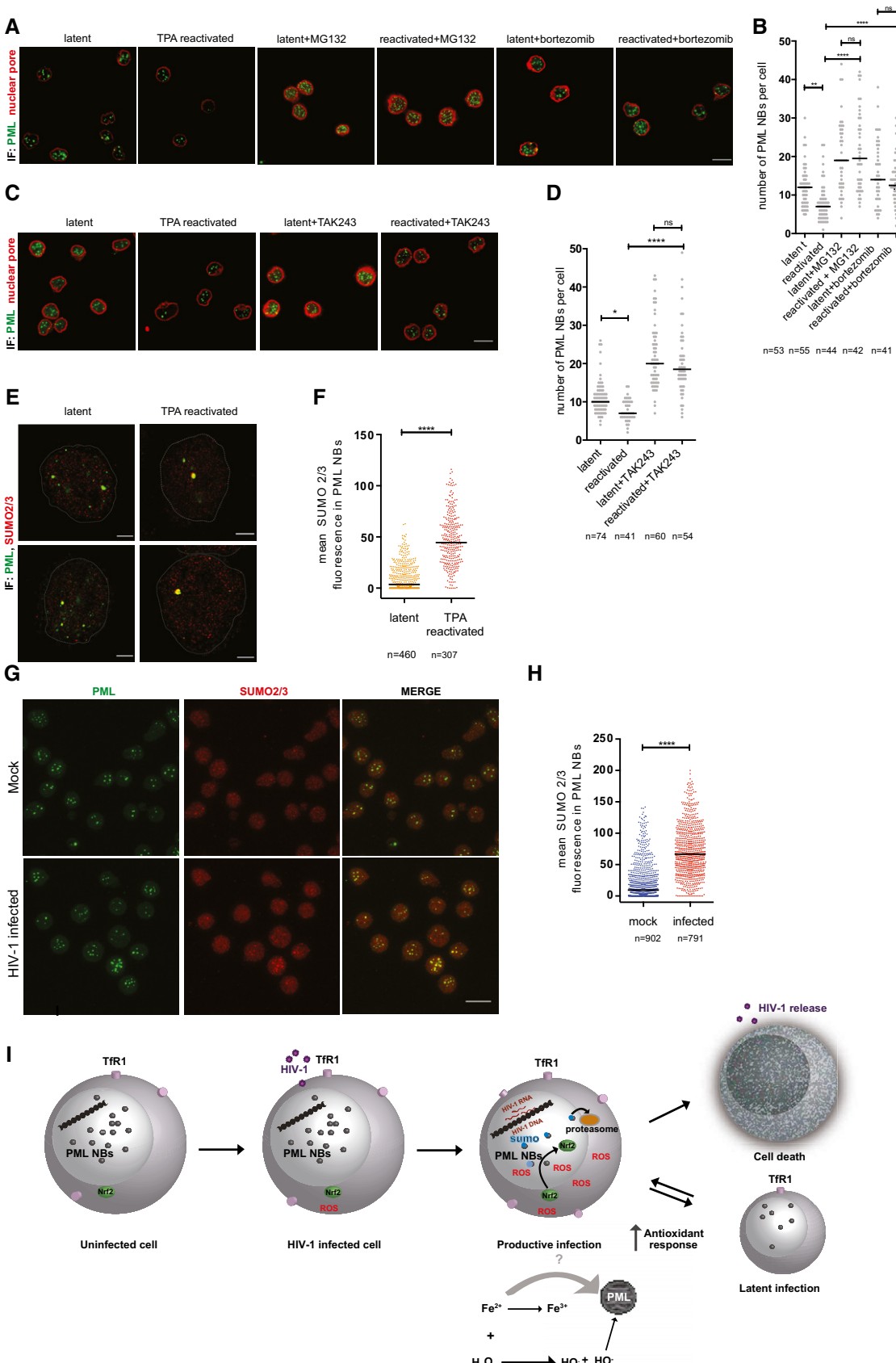

**Figure 6.**

**Figure 6.  PML degradation is mediated by the ubiquitin–proteasome axis.**

A–D    Representative immunofluorescence images (A, C) and quantifications (B, D) of PML NBs (green, maximal projection) and mab414/nuclear pore (red, one *Z* stack) in
       J-Lat 9.2 cells. Cells were left latent or reactivated for 24 h with 10 μM TPA. Proteasome function was blocked for 4 h in both latent and TPA-reactivated cells
       using MG132 (10 μM) or bortezomib (100 nM) (A, B). Alternatively, ubiquitination was blocked for 4 h using TAK243 (10 μM) (C, D). Scale bar 10 μm.
E, F    Representative immunofluorescence images (E) and quantification (F) of PML NBs (green) and SUMO 2/3 (red) in J-Lat 9.2 cells left latent or reactivated for 24 h
       with 10 μM TPA. Cells were imaged by STED microscopy, scale bar 2 μm.
G, H    Representative immunofluorescence images (G) and quantification (H) of PML NBs (green, maximal projection) and SUMO 2/3 (red, maximal projection) during
       mock and productive HIV-1 infection of primary CD4$^+$ T cells at 7 dpi. Scale bar 10 μm.
I       Schematic representation of the interplay between metabolic imbalances induced by HIV-1 replication and PML stability. Upon HIV-1 infection of activated CD4$^+$ T
       cells, ROS content is increased by HIV-1 replication and potentially by iron through the Fenton reaction, leading to oxidative stress. Oxidative stress is accompanied
       by PML NB SUMOylation and PML protein degradation via the ubiquitin–proteasome axis. Upregulation of Nrf2-mediated antioxidant responses in infected cells
       facilitates cell survival and reversal to a resting state where ROS content is decreased and PML reformed, while latency reactivation reignites oxidative stress and
       PML degradation.

Data information: Data were analyzed by Kruskal–Wallis test (B, D) or unpaired Mann–Whitney test (F, H). Black dash indicates median, *n* = number of PML NBs.
*$P < 0.05$; **$P < 0.01$; ****$P < 0.0001$.

shown to decrease HIV-1 replication (Debebe *et al*, 2007) and our data show that drugs decreasing oxidative stress or iron content could rescue HIV-1-induced PML degradation. Conversely, reactivation from latency was associated with PML depletion and increased antioxidant gene expression. This observation suggests that latency reversal recapitulates the metabolic changes observed upon productive infection and that the pathways contributing to induction of HIV-1 latency are also involved in its maintenance. It is noteworthy that iron supplementation could enhance HIV-1 latency reversing in our experiments. As drugs inducing oxidative stress and/or causing PML depletion have been proposed as latency-reversing agents (Lusic *et al*, 2013; Khan *et al*, 2015), combining compounds targeting these pathways might significantly enhance their anti-latency potential.

Our study presents some discrepancies from previously published papers as well as some limitations. In particular, previous evidence has been reported of both activation (Zhang *et al*, 2009) and inhibition of the Nrf2-regulated pathways upon HIV-1 infection (Gill *et al*, 2014). Cell type differences might explain these discrepancies in which inhibition of Nrf2 has been reported in myeloid cells, in particular in the central nervous system (Gill *et al*, 2014). A limitation of the present study resides in circumscribing the analyses to bulk CD4$^+$ T cells or PBMCs. Different T-cell subsets are characterized by variable baseline antioxidant defenses (Chirullo *et al*, 2013) and could thus contribute differently to the observed effects. Of note, a recent study has elegantly characterized the metabolic activity of T-cell subsets and shown the correspondence with their different predisposition to HIV-1 infection (Valle-Casuso *et al*, 2018). Another potential limitation of our study is the combined analysis of all different isoforms of PML, which can independently influence its function and localization (Condemine *et al*, 2006). Although *ex vivo* data from SIVmac-infected macaques were in good agreement with the main findings of our work, the low frequency of infected cells *in vivo* suggests an important role for bystander effects in the phenomena that we observed. However, despite this potential confounding factor, our single-cell analyses support the idea that productively infected cells are the drivers of ROS generation and PML degradation.

Some of the pathways highlighted in the present study have translational potential. The permanent upregulation of antioxidant defenses that we observed in latently HIV-1-infected cells parallels a condition observed in cancer cells (Benhar *et al*, 2016). Pre-clinical and clinical studies suggest that this upregulation might render HIV-infected and cancer cells more susceptible to pharmacologically induced oxidative bursts, namely through the inhibition of the Nrf2 targets TrxR1 and GCLC (Benhar *et al*, 2016). Moreover, the PML-depleting drug ATO has recently advanced to clinical trials enrolling HIV$^+$ individuals (NCT03980665). Finally, our data using iron chelators open a possibility for the selective killing of latently infected cells that might be included to currently studied "shock and kill" strategies for an HIV cure.

In conclusion, the present work demonstrates that HIV-1 can alter the nuclear structure of the host and favor its own replication by increasing oxidative stress and intracellular iron import, while the cellular response to these effects preludes to latency development. These data open the possibility of combining drugs to increase oxidative stress and modulate iron content to favor PML depletion and target the HIV-1 reservoir.

## Materials and Methods

Whole blood of healthy donors was provided by Heidelberg University Hospital Blood Bank following approval by the local ethics committee.

### Method details

*Cell lines*

The T lymphocyte cell lines J-Lat 9.2 and 15.4, harboring one latent HIV-1 DNA copy per cell (Jordan *et al*, 2003), Jurkat E6.1 (ATCC), or Jurkat-Tag cells (Northrop *et al*, 1993) were cultured in RPMI + 10% fetal bovine serum and kept at a concentration of 0.25–0.5 × 10$^6$ cells/ml in an incubator at 37°C in a 5% CO$_2$ atmosphere.

*Electroporation with siRNAs*

Prior to electroporation, 1 × 10$^6$ of Jurkat-TAg cells were centrifuged at pelleted and resuspended in 100 μl of the electroporation buffer from the Cell Line NucleofectorTM Kit (VVCA-1003, Lonza). Cell suspension was then mixed with 300 nM-specific targeting siRNA [Silencer Select HMOX1, Ambion #4390824; Silencer Select NFE2L2 (Nrf2), Ambion# 4392420] or non-targeting control siRNA (Silencer Select Negative control, Ambion #4392420). The resuspended cells

were transferred to cuvettes and immediately electroporated using the program X-01. After electroporation, cells were transferred into pre-warmed RPMI medium supplemented with 10% FBS and penicillin/streptomycin and incubated at 37°C, 5% $CO_2$ for 48–72 h.

### Primary CD4$^+$ T-cell isolation and culture

CD4$^+$ T cells were isolated using the RosetteSep$^{TM}$ Human CD4$^+$ T Cell Enrichment Cocktail (STEMCELL Technologies Inc., Vancouver, British Columbia, Canada) following the manufacturer's instructions. After isolation, cells were activated by adding the Dynabeads$^®$ Human T-Activator CD3/CD28 using a bead/cell ratio of 1:2. Cells were cultured in RPMI 1640 supplemented with 20% FBS, penicillin/streptomycin, and 10 ng/ml IL-2 and kept at a concentration between 1 and $2 \times 10^6$ cells/ml in an incubator at 37°C in a 5% $CO_2$ atmosphere. Three days after activation, cells were infected and/or treated with drugs as described below.

### FANA oligo-mediated silencing of primary CD4$^+$ T cells

Custom-made PML FANA ASOs were reconstituted by using an appropriate volume of sterile water. CD4$^+$ T cells were seeded prior to addition of the silencing oligos at the concentration of $2 \times 10^6$ cells/ml. Three different FANA PML ASOs or control oligo was added drop wise to the cell suspension at the concentration of 1 μM or 5 μM. Cells were cultured in RPMI medium supplemented with 20% FBS, penicillin/streptomycin, and 10 ng/ml IL-2 and incubated at 37°C, 5% $CO_2$ for 48–72 h.

### Enrichment of live primary CD4$^+$ T cells

The EasySep$^{TM}$ Dead Cell Removal (Annexin V) Kit (Stemcell Technologies Inc., Vancouver, British Columbia, Canada) was used to deplete apoptotic CD4$^+$ T cells according to the manufacturer's instructions. Briefly $10^7$ cells were labeled with a mix of antibodies specific for the apoptosis marker Annexin V. After incubation with the antibody mix, unwanted (apoptotic) cells were labeled with biotin and removed by magnetic selection (EasySep$^{TM}$ magnet).

### In vitro infection with HIV-1

For in vitro infection, $5–20 \times 10^6$ activated CD4$^+$ T cells were used. Cells were pelleted and infected by incubation with wt HIV-1$_{NL4-3}$ (0.2, 2, or 20 ng p24/$10^6$ cells) for 2–4 h in an incubator at 37°C and 5% $CO_2$. Following infection, cells were washed and resuspended at $1 \times 10^6$ cells/ml in RPMI 1640 supplemented with 20% FBS and 10 ng/ml IL-2.

### Drug treatments

For testing pharmacologically induced PML depletion or iron overload/chelation, CD4$^+$ T cells at day 3 post-activation were left untreated or treated with (i) arsenic trioxide (As$_2$O$_3$; Sigma-Aldrich, Saint Louis, MI, USA, ref: 356050) at 5 μM concentration, (ii) iron (III) chloride hexahydrate (FeCl$_3$ 6H$_2$O; Sigma-Aldrich, Saint Louis, MI, USA, CAS: 10025-77-1) at 500 μM, (iii) deferoxamine mesylate salt (DFOA, Sigma-Aldrich, Saint Louis, MI, USA, CAS: 138-14-7) at 1 μM concentration, (iv) deferiprone (L1, Sigma-Aldrich, Saint Louis, MI, USA, CAS: 30652-11-0) at 50 μM concentration, and (v) H$_2$O$_2$ at 100 μM concentration (H1009; Sigma-Aldrich, Saint Louis, MI, USA). Cells treated with FeCl$_3$ 6H$_2$O were harvested 48 h post-treatment and used for Western blot analyses. Cells treated with As$_2$O$_3$ and DFOM were infected and used for further analyses as

described below. For specifically blocking HIV-1 replication, CD4$^+$ T cells were incubated with a three-drug ART 5 days post-infection with wt HIV-1. The antiretroviral combination was composed of T-20 (10 μM), raltegravir (10 nM), and efavirenz (100 nM) (kindly provided by the National Institutes of Health AIDS Research and Reference Reagent Program). For reactivation experiments, J-Lat cells were incubated with (i) 12-O-tetradecanoylphorbol-13-acetate at 10 μM concentration (TPA; Sigma-Aldrich, Saint Louis, MI, USA), (ii) H$_2$O$_2$ at 10 μM concentration, (iii) ferric nitriloacetate (Fe-NTA) at 150 μM concentration: J-Lat 9.2, while primary CD4$^+$ T cells were incubated at 14 dpi, with the Dynabeads$^®$ Human T-Activator CD3/CD28 using a bead/cell ratio of 1:2 or with phytohemagglutinin (PHA) at 10 μg/ml. Fe-NTA was prepared as described in Awai et al (1979). For proteasome inhibition experiments, cells were treated with 10 μM MG132 (carbobenzoxy-Leu-Leu-leucinal) (M8699-1MG, Sigma-Aldrich) or 100 nM bortezomib (Cell Signalling, #2204) for 4 h. For inhibition protein ubiquitination, cells were treated with 10 μM UAE inhibitor MLN-7243 (TAK 243) (A-1384, Active Biochem) for 4 h.

### RNA and DNA extraction and quantification

Total RNA and genomic DNA were extracted using, respectively, the InviTrap$^®$ Spin Universal RNA Mini Kit (Stratec Biomedical, Germany) and the DNeasy Blood & Tissue Kits (Qiagen, Germany) according to the manufacturers' instruction. When extracting DNA from cells fixed with paraformaldehyde (PFA) after p24$^+$ cell sorting, an initial decrosslinking step (10 min at 70°C) was included, followed by overnight incubation with proteinase K and lysis buffer at 56°C. Nucleic acids were quantified using a P-class P 300 NanoPhotometer (Implen GmbH, Munich, Germany).

### RNA-Seq analyses

Libraries for RNA-Seq of CD4$^+$ T cells infected in vitro with HIV-1 or mock infected were created using the Illumina TruSeq RNA library Preparation Kit with Ribo-Zero Gold (Human/Mouse/Rat). 450 ng of total RNA was used for library generation. Before and after library preparation, the quality of the samples was analyzed with the Agilent Bioanalyzer and the concentrations determined with the Qubit fluorometer. Equimolar sample contents were pooled and loaded onto the Illumina NextSeq 500 for sequencing in high output (> 75 cycles paired end). CASAVA was used to perform base calling. The quality of the $2 \times 75$ bp reads was assessed with FastQC (https://www.bioinformatics.babraham.ac.uk/projects/fastqc/). STAR (Dobin et al, 2013) version 2.5 was used to align raw reads to build version hg38 of the human genome and to calculate counts for GENCODE (release 25) basic annotated genes. Normalization and differential analysis were carried out using the DESeq2 R package (Love et al, 2014). Genes were considered differentially expressed with an adjusted P-value lower than 0.05. Gene sets representing the activity of pathways were derived from the Molecular Signatures Database (Liberzon et al, 2011). The statistical overrepresentation of gene sets among differentially expressed genes and proteins was assessed with a one-tailed Fisher exact test. R (version 3.3.1) was used for statistical analyses. GEO GSE127468.

RNA-Seq data for SIVmac239-infected rhesus macaques were taken from previously published work describing changes in gene expression in PBMCs collected from SIV-infected macaques prior to and during ART treatment (Micci et al, 2015) (GEO GSE73232). To

identify pathways differentially modulated between the ART-naïve and ART-treated time points, Gene Set Enrichment Analysis (Subramanian *et al*, 2005) was performed using RNA-Seq expression data as follows. Transcripts were ranked by differential expression between ART-naïve and ART-treated time points using the Signal2-Noise metric. GSEA was performed using the desktop module available from the Broad Institute (www.broadinstitute.org/gsea/). GSEA was performed on the ranked transcript lists using 1,000 gene set permutations, collapse of duplicates to Max probe, and random seeding. Gene sets investigated included C2: curated, and C5: GO gene sets. All gene sets were retrieved from the Molecular Signatures Database (MSigDB v6.2).

### Proteomic analysis

Sample preparation was performed using the Single-Pot Solid-Phase-enhanced Sample Preparation approach SP3, as described elsewhere (Hughes *et al*, 2019). In brief, 2 μl of a 1:1 mixture of hydrophilic and hydrophobic carboxylate-coated paramagnetic beads (SeraMag Speed Beads, #44152105050250 and #24152105050250, GE Healthcare, Little Chalfont, UK) was added to 30 μg protein of each sample. Acetonitrile was added to achieve a final concentration of 50% organic solvent. Bound proteins were washed with 70% ethanol and 100% acetonitrile. Beads were resuspended in 5 μl 50 mM triethylammonium bicarbonate buffer containing 0.6 μg trypsin (SERVA, Heidelberg, Germany) and 0.6 μg endopeptidase Lys-C (Wako). Digestion was carried out for 16 h at 37°C in a PCR cycler. Recovered peptides were resuspended in 1% formic acid/5% DMSO and stored at 20°C prior MS analysis. All samples were analyzed on a Q-Exactive Plus (Thermo Scientific) mass spectrometer that was coupled to an EASY nLC 1200 UPLC (Thermo Scientific). Peptides were loaded with solvent A (0.1% formic acid in water) onto an in-house packed analytical column (50 cm × 75 μm I.D., filled with 2.7 μm Poroshell EC120 C18, Agilent) equilibrated in solvent A. Peptides were chromatographically separated at a constant flow rate of 250 nl/min using the following gradient: 3–5% solvent B (0.1% formic acid in 80% acetonitrile) within 1 min, 5–30% solvent B within 91 min, 30–50% solvent B within 17 min, followed by washing at 95% for 10 min. For library generation, the mass spectrometer was operated in data-dependent acquisition mode. The MS1 survey scan was acquired from 350 to 1,300 $m/z$ at a resolution of 70,000. The top 10 most abundant peptides were isolated within a 2 Th window and subjected to HCD fragmentation at a normalized collision energy of 27%. The AGC target was set to 5e5 charges, allowing a maximum injection time of 55 ms. Product ions were detected in the Orbitrap at a resolution of 17,500. Precursors were dynamically excluded for 20 s. A data-independent acquisition method was employed for protein quantification. The mass spectrometer was operated in data-independent acquisition (DIA) mode. The MS1 scan was acquired from 400 to 1,220 $m/z$ at a resolution of 140,000. MSMS scans were acquired for 10 DIA windows at a resolution of 35,000. The AGC target was set to 3e6 charges. The default charge state for the MS2 was set to 4. Stepped normalized collision energy was set to A, B, C = 23.5, 26, 28.5%. The MSMS spectra were acquired in profile mode. The raw data of the pooled library samples were processed with Maxquant (version 1.5.3.8) using default parameters. Briefly, MS2 spectra were searched against the UniProt human database, including a list of common contaminants. False discovery rates on protein and PSM

level were estimated by the target-decoy approach to 1% protein FDR and 1% PSM FDR, respectively. The minimal peptide length was set to seven amino acids, and carbamidomethylation at cysteine residues was considered as a fixed modification. Oxidation (M) and acetyl (protein N-term) were included as variable modifications. The match-between run option was disabled. For DIA quantification, raw data were processed with Spectronaut Pulsar X (version 11) using default parameters. Briefly, MS2 spectra were searched against the previously generated library. The maximum of major group top N was set to 6; the decoy method to inverse and the data filtering were set to $q$-value.

### Measurement of reduced (GSH) and oxidized (GSSG) cell content

Measurement of GSH/GSSG content was performed using the GSH/GSSG-Glo™ Assay (Promega; Madison, WI, USA) according to the manufacturer's instruction. Briefly 5 × 10⁵ CD4⁺ T cells were lysed and incubated with a specific probe emitting luciferin in glutathione-dependent reaction. For detection of GSSG content, N-ethylmaleimide was included to block the GSH-luciferin reaction. After addition of luciferase, a luminescent signal was developed and acquired with an Infinite 200 PRO (Tecan, Männedorf, Switzerland) multimode plate reader. A standard curve was used to calculate the total glutathione and GSSG content, while the GSH/GSSG ratio was calculated as follows: [μM total glutathione − (μM GSSG)/μM GSSG].

### Flow cytometry

For surface staining, 500 × 10⁵ cells were pelleted and fixed with 4% PFA in PBS. Cells were then washed twice with cold FACS buffer (2% FCS in PBS) and incubated in the dark at 4°C with an anti-TfR1 (CD71) antibody (clone OKT9; Thermo Fisher Scientific, Waltham, MA, USA). Cells were then washed with PBS and resuspended in FACS buffer. For intracellular staining, the two initial washes were performed with 0.5% Triton X-100 in PBS and cells were stained for 30 min with an α-p24-PE (gag) antibody (Coulter Clone KC57-RD1; Beckman Coulter). Data were acquired with a BD FACSVerse and BD FACSCelesta (Becton Dickinson, Franklin Lakes, NJ, USA) flow cytometers and analyzed using the FlowJo package (FlowJo LLC, Ashland, Oregon, USA v7.6.5).

For cell sorting, 10 × 10⁶ CD4⁺ cells were pelleted and fixed in PBS + 3% PFA on ice for 15 min. Cells were then permeabilized and stained with an α-p24-PE antibody as described in the paragraph above. Sorting of p24⁺ cells was performed on a BD FACSAria II flow cytometer using an 85 μM nozzle.

### Reverse transcription and quantitative polymerase chain reaction (qPCR)

Total RNA was retrotranscribed to cDNA using the SuperScript III Reverse Transcriptase Kit (Thermo Fisher Scientific, Waltham, MA, USA) according to the manufacturer's instructions. Briefly, 500 ng of RNA was mixed with 1 μl random primers (3 μg/μl) and 1 μl dNTPs (10 mM) and incubated at 65°C per 5 min for a predenaturation step. For primer extension, 6 μl of 5× First Strand Buffer, 1 μl DTT (0.1 M), and 0.5 μl of protector RNase inhibitor (40 U/μl, Hoffmann-La Roche, Basel, Switzerland) were added and the mix incubated at 37°C per 2 min. Finally, 1 μl of MMLV RT enzyme (200 U/μl) was added and samples reverse transcribed with the following conditions: 10 min at 25°C followed by 50 min at 37°C—and 15 min at 70°C. For qPCR, a master mix was prepared containing, per each

sample, 10 μl of Taq PCR Iq supermix (Bio-Rad Laboratories, Hercules, CA, USA), 1 μl of primer/probe set (Applied Biosystems, Thermo Fisher), and 8 μl H$_2$O. Commercially available primer-probe mixes (Single Tube TaqMan Gene Expression Assays; Thermo Fisher Scientific, Waltham, MA, USA) were used for thioredoxin (Trx; hs01555214), thioredoxin reductase (TrxR1; hs00917067), glucose-6-phosphate dehydrogenase (G6PD hs00166169), heme oxygenase 1 (HMOX-1; hs01110250), glutamate-cysteine ligase (GCLC; hs00155249), NADPH dehydrogenase [quinone] 1 (NQO1; hs02512143), and promyelocytic leukemia protein (PML; hs00231241). For U1A (gag), the following primers and probe were used forward ACAT CAAGCAGCCATGCAAAA (position 543), reverse CAGAATGGGATA GATTGCATCCA (position 629), probe AAGAGACCATCAATGAG GAA (position 605). For all qPCRs, 1 μl of cDNA was added to the mix and qPCR was performed using a CFX96/C1000 Touch qPCR system with PCR program: polymerase activation/DNA denaturation 98°C 3 min, followed by 45 cycles of denaturation at 98°C for 10 s; annealing/extension at 60°C for 40 s. Final extension was performed at 65°C for 30 s, followed by slow cool down to 4–0.5°C/s.

### Integrated HIV DNA measurement

The integrated HIV-1 DNA content was assessed using a nested Alu-LTR PCR assay as previously described (Tan et al, 2006). Briefly, in the first round of PCR, Alu-LTR fragments were amplified starting from 250 ng of genomic DNA. For the second round, products of the first PCR were diluted 1:50 in H$_2$O and LTR-LTR fragments amplified in a qPCR. A housekeeping gene, i.e., lamin B2, B13 region, was run in parallel in the second round of PCR using 10 ng of genomic DNA and used to normalize the data as described in the Statistical Analysis section.

### MTT assay

Viability upon treatment with the drugs employed was measured through the CellTiter 96® Non-Radioactive Cell Proliferation Assay (MTT) (Promega; Madison, WI, USA) similarly to Shytaj et al (2015). Briefly, 300 × 10$^5$ cells were resuspended in 100 μl RPMI + 10% FCS and transferred to a 96-well plate. To each well was added the MTT solution (15 μl), and after 2–4 h, the reaction was stopped by the addition of 100 μl of the solubilization/stop solution. Absorbance values at 570 nm were acquired with an Infinite 200 PRO (Tecan, Männedorf, Switzerland) multimode plate reader. Reactions were conducted in triplicate, and the averages of the triplicates were normalized over the matched untreated controls and expressed as percentage.

### Immunofluorescence and immuno-HIV-1 DNA FISH

Approximately 3 × 10$^5$ uninfected or HIV-1-infected CD4$^+$ T cells were plated on the PEI-coated coverslips placed into a 24-well plate for 1 h at 37°C. Cells were treated with 0.3× PBS to induce hypotonic shock and fixed in 4% PFA in PBS for 10 min. Coverslips were extensively washed with PBS, and cells were permeabilized in 0.5% Triton X-100/PBS for 10 min. After three additional washings with PBS-T (0.1% Tween-20), coverslips were blocked with 4% BSA/PBS for 45 min at RT and primary antibody anti-rabbit lamin B1 (Abcam ab16048), mab414 (1:500) (Abcam 24609), anti-mouse PML (Santa Cruz (PG-M3): sc-966 or Bethyl Lab A301-167A) (1:500 in 1% BSA/PBS), or TfR1 (1:200) (Abcam ab8598) was incubated overnight at 4°C. Following three washings with PBS-T,

fluorophore-coupled secondary antibodies (anti-mouse, coupled to Alexa 488, anti-rabbit, coupled to Alexa 568 or Alexa 647, diluted 1:1,000 in 1% BSA/PBS) were incubated for 1 h at RT. Nuclear counterstaining was performed using 1:10,000 Hoechst 33342 in PBS followed by two washings in PBS and mounting the coverslips with Mowiol. For Immuno-HIV-1 DNA FISH experiments, following fluorophore-coupled secondary antibody incubation, coverslips were extensively washed and post-fixed with EGS in PBS. Coverslips were washed three times with PBS-T and incubated in 0.5% Triton X-100/0.5% saponin/PBS for 10 min. After three washings with PBS-T, coverslips were treated with 0.1 M HCl for 10 min, washed three times with PBS-T, and additionally permeabilized step in 0.5% Triton X-100/0.5% saponin/PBS for 10 min. After extensive PBS-T washings, RNA digestion was performed using RNAseA (100 μg/ml) for 45 min at 37°C. Coverslips were equilibrated for 5 min in 2× SSC and put in hybridization solution over night at 4°C. HIV-1 FISH probes were generated by labeling HIV-1 DNA (pHXB2) plasmid. Biotin-dUTP nucleotide mix containing 0.25 mM dATP, 0.25 mM dCTP, 0.25 mM dGTP, 0.17 mM dTTP, and 0.08 mM biotin-16-dUTP in H$_2$O was prepared. 3 μg of pHXB2 was diluted with H$_2$O in a final volume of 12 μl, and 4 μl of each nucleotide mix and nick translation mix was added. Labeling was performed at 15°C for 5 h. For ROS detection experiments, the HIV-1 probe was labeled directly. 3 μg of HIV-1 DNA (pHXB2) plasmid was diluted in a final volume of 22.5 μl H$_2$O. 2.5 μl of 0.2 mM orange fluorophore-coupled dUTP; 5 μl of 0.1 mM dTTP; 10 μl of dNTP mix containing 0.1 mM of each dATP, dCTP, and dGTP; and 5 μl of 5× nick translation buffer (Abbott) were added, and reagents were mixed well by vortexing. The reaction was started by addition of 5 μl Nick translation enzymes (Abbott) and incubated at 15°C for 13–14 h. The probes were precipitated in 100% ethanol with sodium acetate overnight and resuspended in 2× SSC/10% dextran sulfate/50% formamide, denatured, and stored at −20°C until use. Probe hybridization was performed with 2 μl of HIV-1 probe in 6 μl reaction with 2× SSC/10% dextran sulfate/50% formamide, denatured at 95°C for 5 min, and then kept on ice for 1 min. After spotting on a glass slide in a metal chamber, the probe was denatured at 80°C for 8 min and subsequently hybridized to the sample at 37°C for 40–65 h in a water bath. Probe detection was carried out by washing the coverslips with 2× SSC and 0.5× SSC at 37 and 65°C, respectively, 1 h blocking in TSA blocking buffer (TNB) and detection with streptavidin-HRP in TNB for 40 min at 37°C. Coverslips were then washed with TNT wash buffer at RT, before incubation with fluorescein plus amplification reagent (1:1,500 in TSA Plus amplification diluent, part of TSA Plus Fluorescein kit) for 5 min at RT. Coverslips were then washed five more times in TNT buffer. For directly labeled probes, detection was carried out by washing the coverslips with 2× SSC and 0.5× SSC at 37 and 65°C, respectively. Nuclear counterstaining was performed using 1:10,000 Hoechst 33342 in PBS followed by two washings in PBS and mounting the coverslips with Mowiol. For ROS detection, CellROX® Green dye from Thermo Fisher Scientific was added to the coverslips at 5 μM concentration for 30 min at 37°C. Two washings in PBS were subsequently performed, followed by mounting the coverslips with Mowiol. For stimulated emission depletion (STED) microscopy, IF was performed with the following primary antibodies: lamin B1 (1:100) (Abcam ab16048), TfR1 (1:100) (Abcam ab8598), PML (1:100) (Santa Cruz (PG-M3): sc-

966, and anti-SUMO 2/3 (1:100) (kind gift from Dr Frauke Melchior). Following three washings with PBS-T, fluorophore-coupled secondary antibodies (Anti-Mouse IgG—Atto 594 antibody produced in goat, 1 mg/ml (76085, Sigma-Aldrich) and STAR RED Goat anti-rabbit IgG, 500 µg (2-0012-011-9, Abberior) diluted 1:100 in 1% BSA/PBS) were incubated for 1 h at RT. Nuclear counterstaining was performed using 1:10,000 Hoechst 33342 in PBS followed by two washings in PBS and mounting the coverslips with Mowiol.

### Single-molecule HIV-1 RNA FISH coupled to immunofluorescence

A custom-made Stellaris HIV-1 RNA FISH probe Fluor Red 610, dissolved to 12.5 µM in TE buffer, was used for the RNA FISH experiments. Approximately $5 \times 10^5$ HIV-1-infected PBMCs were washed twice in RNA grade PBS and fixed in 3.7% formaldehyde (FA) solution in PBS for 10 min. After extensive washings with PBS, cells were permeabilized in suspension in 70% ice-cold ethanol for at least 1 h. Cells were then adhered to PEI-coated slides for 30 min at RT and washed with RNA grade PBS for three times 3 min at RT. Antibody incubation for immunofluorescence for Nrf2 (mab3925, 1:250; R&D systems, Minneapolis, MN, USA) or PML (sc/966Å~, 1:500; Santa Cruz Biotechnology, Dallas, TX, USA) was performed at RT for 1–2 h. After PBS washings, secondary antibody anti-mouse, coupled to Alexa 488 (1:1,000 dilution), was incubated 1 h at RT light protected.

After extensive PBS washings, cells were post-fixed with 3.7% FA in RNA grade PBS at room temperature for 10 min, washed with PBS, and permeabilized with 70% ice-cold ethanol for 30 min. Cells were washed/equilibrated in buffer A [Stellaris RNA FISH Wash Buffer A (Biosearch Tech. Cat# SMF-WA1-60)] with 10% formamide in water. 1–3 µl of HIV-1 RNA probe diluted in hybridization buffer [Stellaris RNA FISH Hybridization Buffer (Biosearch Tech. Cat# SMF-HB1-10)] and 10% formamide were spotted on coverslips and incubated overnight in humid conditions at 37°C. Coverslips were subsequently washed with buffer A four times at 37°C and one time at RT, and nuclear counterstaining was performed using 1:10,000 Hoechst 33342 in buffer A followed by two washings in buffer B [Stellaris RNA FISH Wash Buffer B (Biosearch Tech. Cat# SMF-WB1-20)] and mounting the coverslips with Mowiol.

### Stimulated emission depletion (STED) microscopy

STED imaging was performed with a $\lambda = 775$ nm STED system (Abberior Instruments GmbH, Göttingen, Germany), containing an easy 3D optic module (Abberior Instruments) and the = 640 nm excitation laser line using a 100× Olympus UPlanSApo (NA 1.4) oil immersion objective. Images were acquired using the 590- and 640-nm excitation laser lines.

Deconvolution of STED images was performed in Imspector software (Abberior Instruments) via the linear deconvolution tool.

Images were further processed in Fiji-Imagej (Schindelin et al, 2012) by using the Gaussian-Blur Filter and setting the sigma radius to 1 to further increase signal/noise ratio. Different analyzed excitation channels from the same sample were also merged into one image by using this software.

### Confocal microscopy

Data were acquired using a Leica TCS SP8 confocal microscope (Leica Microsystems GmbH, Wetzlar, Germany) with a 63×

objective immersed in oil. A distance (z-step) of 500 nm was used to acquire three dimensional stacks with zoom set at 1× or 3×.

Image analysis was performed with Fiji-Imagej (Schindelin et al, 2012) using the following macros to perform automatic segmentation and counting of the number of cells and PML bodies:

Segmentation:

```
//run("Brightness/Contrast...");
setMinAndMax(0, 150);
run("Gaussian Blur...", "sigma = 2 stack");
//run("Threshold...");
setAutoThreshold("RenyiEntropy dark");
setThreshold(40, 255);
//setThreshold(40, 255);
setOption("BlackBackground", true);
run("Convert to Mask", "method = RenyiEntropy
background = Dark black");
roiManager("Show None");
roiManager("Show All");
```

count:

```
run("Duplicate...", "duplicate");
run("3D Objects Counter", "threshold = 1 slice = 16min. = 1
max. = 356796 objects statistics summary");
```

### 3D FISH PML analysis algorithm

We developed an algorithm, implemented in MATLAB, in order to automatically analyze 3D RNA FISH and PML proteins in fluorescence cell image z-stacks. The algorithm is derived from Cortesi et al (2019) with some adaptations. It performs the 2D segmentation of cell nuclei and the detection of FISH and PML spots for each slice of the stack followed by the 3D reconstruction and identification of nuclei and spots. It then counts and measures the 3D FISH and PML spots and calculates the total intensity value of FISH spots. We recognized as infected those cells with HIV-1 RNA FISH signals satisfying these constraints: (i) at least one signal with size greater than 200 voxels and (ii) total intensity of FISH signals greater than 15,000. The algorithm can be sketched as follows:

```
for each slice n of the stack
    I_vol_FISH = I_FISH,n(:,:)
    nuclei_n = nuclei_seg(I_DAPI,n) %performs 2D nuclei segmentation
    nuclei_vol(:,:,n) = nuclei_n(:,:)
    fish_n = detect_spot(I_FISH,n) %performs 2D FISH spot detection
    fish_vol(:,:,n) = fish_n(:,:)
    pml_n = detect_spot(I_PML,n) %performs 2D PML spot detection
    pml_vol(:,:,n) = pml_n(:,:)
endfor
nuclei_CC = bwconncomp(nuclei_vol)
nuclei_L = labelmatrix(nuclei_CC)
compute volume for each nucleus object in nuclei_CC
exclude nuclei whose volume is less than 10% of mean volumes
{NCL}_M < - identified 3D nuclei pixel data
for each nucleus m in {NCL}_M
    NCL_m.FISH = {NCL}_M .* fish_vol %3D positions of detected
    FISH spots within NCL_m
```

$NCL_m.FISH = \textbf{bwareaopen}(NCL_m.FISH, 17, 6)$

$FISH\_CC_m = \textbf{bwconncomp}(NCL_m.FISH)$

compute volume for each FISH spot object in $FISH\_CC_m$

exclude FISH spots whose volume is below 100 voxels

$NCL_m.FISH_{PICK} <$ - final FISH spot objects

count final FISH spot objects

$NCL_m.Intensity_{FISH} = I\_vol_{FISH} .* NCL_m.FISH_{PICK}$ %intensity values of FISH spots

$NCL_m.TotIntensity_{FISH} = sum(sum(sum(NCL_m.Intensity_{FISH})));$ %total intensity value of FISH spots

$NCL_m.PML = \{NCL\}_M .* pml\_vol$ %3D positions of detected PML spots within $NCL_m$

$NCL_m.PML = \textbf{bwareaopen}(NCL_m.PML, 17, 6)$

$PML\_CC_m = \textbf{bwconncomp}(NCL_m.PML)$

compute volume for each PML spot object in $PML\_CC_m$

exclude PML spots object whose volume is below 0.15 micron cubes

$NCL_m.PML_{PICK} <$ - final PML spot objects

count final PML spot objects

endfor

$I_{DAPI}$, $I_{FISH}$, and $I_{PML}$ are the stack images with DAPI, RNA FISH, and PLM staining, respectively. The function *nuclei_seg* performs a partition of cell image in nuclei regions and background implementing a region-based segmentation algorithm (Chan *et al*, 2006; Goldstein *et al*, 2009; Antonelli & De Simone, 2018). Output image *nuclei_n* highlights foreground objects (nucleus regions). The function *detect_spot* has four major steps. It first filters the input image $I_{FISH,n}$ ($I_{PML,n}$) applying the Laplacian of Gaussian (LoG) operator (*fspecial* MATLAB function) of size 9 and standard deviation 7. This enhances the signal in the areas where objects are present. Then, the function applies the h-dome transformation (Vincent, 1993) that extracts bright structures by cutting off the intensity of height h from the top, around local intensity maxima. We used h = 0.5 with a neighborhood size of 15 × 15. We decided to not use a global operator after having observed that a FISH spot (PML spot) in one part of the image could be lighter or darker than the background in another part. This is due to the facts that spots have inhomogeneous intensity distribution over the image and that the image may have an uneven background. In the third step, the function performs a thresholding on h-dome image that excludes pixels whose intensity values are below a threshold. The FISH and PML threshold values are defined as 1.96 and 1 standard deviations above the mean of dome intensity values, respectively. We therefore assumed that spot areas have significant intensity disparity with respect to other bright areas present in cell nucleus. Lastly, the function applies a thresholding operation based on the surface areas of the spots, in order to discard too small objects which are probably just noise. It filters out spots smaller than a surface area of 8. *detect_spot* produces an accurate set of FISH and PML spots.

*fish_vol*, *pml_vol*, and *nuclei_vol* are 3D arrays that contain the positions of the detected FISH and PML spots and nuclei from all slices.

3D reconstructions of nuclei are obtained through the connected components algorithm (*bwconncomp* MATLAB function, using a connectivity of 26). 3D nuclei are then labeled by applying the *labelmatrix* MATLAB function so they can easily separated each from the others.

The algorithm computes the volume of each 3D reconstruction, discarding, as noise, objects whose volume is less than 10% of mean volumes.

The algorithm uses the *bwareaopen* function in order to discard too small detected spot objects which are probably just noise. 3D reconstructions of spots are obtained through the connected components algorithm (*bwconncomp* MATLAB function, using a connectivity of 6). Then, a threshold operation is performed on the 3D spots to obtain the more significant ones: It keeps in all the PML spots whose volume is above 0.15 micron cubes and all the FISH spots whose volume is above 100 voxels.

### Biochemical fractionation, SDS–PAGE, and Western Blot

Cytoplasmic and nuclear cell fractions were prepared from 10 to $20 × 10^6$ cells. Briefly, cell pellets were resuspended in cytoplasmic buffer: 10 mM Tris pH 7.9; 3 mM $CaCl_2$, 2 mM $MgCl_2$, 0.1 mM EDTA, 0.34 M sucrose, and 1 mM DTT supplemented with protease inhibitors (Roche). After swelling in cytoplasmic buffer on ice for 5 min, Triton X-100 was added to a final concentration of 0.1%. Cytoplasm was extracted after 10-min incubation and 10-min centrifugation at 900 *g* at 4°C. Nuclei were washed in the same buffer without the detergent and lysed in nuclei lysis buffer: 20 mM HEPES pH 7.9; 2.5 mM $MgCl_2$; 0.5 mM EDTA; 150 mM KCl; 10% glycerol; and 0.5% NP-40 plus protease inhibitors (add fresh) on ice for 10 min. Nucleic acids were removed with 250 U of benzonase (Roche) for 1 h at 4°C. Nuclear soluble fraction was extracted by the addition of high salt buffer to achieve 250 mM KCl for 1 h at 4°C rotating, followed by 45-min centrifugation at maximum speed at 4°C. Alternatively, cytoplasmatic and nuclear cell fractions were isolated using the REAP method, according to the published protocol (Suzuki *et al*, 2010).

For SDS–PAGE experiments, $1–5 × 10^6$ cells were harvested and homogenized in lysis buffer (20 mM Tris–HCl, pH 7.4, 1 mM EDTA, 150 mM NaCl, 0.5% Nonidet P-40, 0.1% SDS, and 0.5% sodium deoxycholate supplemented with protease inhibitors (Sigma-Aldrich, Saint Louis, MI, USA) for 10 min at 4°C. For further homogenization, lysates were sonicated for 5 min using a Bioruptor® Plus sonication device (Diagenode, Liège, Belgium). Protein concentration was assayed using the Micro BCA Protein Assay Kit (Thermo Fisher Scientific, Waltham, MA, USA) or Bradford method (Bio-Rad Laboratories, Hercules, CA, USA) and acquired through a Implen NanoPhotometer® Pearl (Implen GmbH, Munich, Germany). Equal amounts of total cellular proteins (5 or 25 μg) were loaded and run on a precast NuPAGE Bis-Tris 4–12% (Thermo Fisher Scientific, Waltham, MA, USA) SDS–PAGE at 120 V. Proteins were then transferred onto a nitrocellulose membrane (GE Healthcare, Little Chalfont, UK) for 3 h using a Trans–Blot device for semi-dry transfer (Bio-Rad Laboratories, Hercules, CA, USA). Alternatively, wet transfer was performed for 3 h at 4°C (Bio-Rad Laboratories, Hercules, CA, USA). Membranes were then blocked for 1 h at RT with 5% skim milk in 0.1% PBS-Tween and incubated overnight at 4°C with the following primary antibodies diluted in 5% milk: α-beta-actin (1:1,000), (Sigma-Aldrich, Saint Louis, MI, USA), α-Trx (1:500; sc-58440), α-NQO1 (1:500, sc-32793) (Santa Cruz Biotechnology, Dallas, TX, USA), α-TrxR1 (1:500, MAB7428) (R&D systems, Minneapolis, MN), α-HMOX-1 (1:500, ab13248), FTH-1 (1:100, ab75972), α SLC40A1 (1:500, ab78066) (Abcam, Cambridge, UK), α-PML (1:500) (A301-167A Bethyl Laboratories, Montgomery, TX,

USA), mono- and polyubiquitinylated conjugate monoclonal antibody (FK2) (1:1,000 BML-PW8810-0100 Enzo), phospho-eIF2α (Ser51) (D9G8) XP® Rabbit mAb (1: 1,000 #3398 Cell Signaling), and eIF2α antibody (1:1,000 #9722 Cell Signaling). Membranes were then washed three times with 0.1% PBS-Tween and incubated for 1 h with a horseradish-conjugated secondary antibody (anti-mouse or anti-rabbit, West Grove, PA, USA). Proteins were visualized with the Amersham ECL Prime Kit (GE Healthcare, Little Chalfont, UK). For membrane reprobing, stripping buffer was used (2% SDS, 8% upper Tris buffer and 110 mM β-mercaptoethanol) for 15 min at 65°C. After extensive washing with 0.1% PBS-Tween, membrane was blocked and reincubated with desired antibodies.

### Molecular modeling

The crystal structure of PML-Ring domain bound to Zn(II) (pdb code 5yuf) was used to model Fe(II) and As binding into the two binding sites (CCCC and HCCC). To this aim, Zn(II) was replaced by Fe(II) and As using Coot (version 0.8.9) (Emsley *et al*, 2010). While Fe(II) showed no clashes as analyzed by MolProbity (Chen *et al*, 2010), As had to be placed manually outside the binding pocket with typical binding distances for As, in order to avoid clashes. Figures were prepared using PyMOL (version 2.2.2, Schrödinger LLC, USA).

### Statistical analysis

Gene set enrichment analyses were performed by Fisher's test (*in vitro* CD4$^+$ T cells) or GSEA (*ex vivo* macaque PBMCs). The Fisher test was used to compare, for each time point, the significance of the expression enrichment of differentially expressed genes or proteins in infected vs. matched mock-infected controls. The GSEA enrichment analysis (Subramanian *et al*, 2005) was used to compare the possible differential expression of the pathways of interest between the ART-naïve and ART-treated time points of a macaque cohort. qPCR and Alu PCR values were normalized using the 2(−ΔΔ C(T)) method (Livak & Schmittgen, 2001). Small datasets ($n \leq 4$) were expressed as mean ± SEM and analyzed using the parametric *t*-test (for comparisons between two samples) or one- and two-way ANOVA when one or two variables, respectively, were compared in multiple groups. To further compare two or more specific subgroups, *ad hoc* post-tests correcting for multiple comparisons were applied (Tukey's and Sidak's). For larger datasets (such as microscopy data) where a normality test could be reliably performed, the D'Agostino-Pearson test was applied to assess the distribution. Data were then represented as scatter dot plots depicted the mean or median or boxplots and whiskers, showing the median and percentile ranges. Where there was no normal distribution, data were analyzed by the unpaired Mann–Whitney test (two groups) or Kruskal–Wallis tests followed by Dunn's post-test to correct for multiple comparisons (more than two groups). All analyses were performed using GraphPad Prism v6 (GraphPad Software, San Diego, CA, USA).

## Data availability

RNA-Seq datasets analyzed in this article have been deposited to the Gene Expression Omnibus (GEO) repository:

1. *In vitro* primary human CD4$^+$ T cells: GEO GSE127468 (https://www.ncbi.nlm.nih.gov/geo/query/acc.cgi?acc = GSE127468).
2. *Ex vivo* macaques PBMCs: GEO GSE73232 (https://www.ncbi.nlm.nih.gov/geo/query/acc.cgi?acc = GSE73232).

The proteomic dataset has been deposited to the ProteomeXchange Consortium via the PRIDE (Perez-Riverol *et al*, 2019) partner repository with the dataset identifier PXD012907 (http://www.ebi.ac.uk/pride/archive/projects/PXD012907). PML molecular modeling was performed using structure pdb code: 5yuf (http://www.rcsb.org/pdb/explore/explore.do?structureId = 5yuf) from Protein Data Bank.

Expanded View for this article is available online.

## Acknowledgements

The authors thank Dr. Lara Manganaro, Dr. Shawon Gupta, and Dr. Zunamys Carrero for proofreading and helpful suggestions. We acknowledge microscopy support from the Infectious Diseases Imaging Platform (IDIP) at the Center for Integrative Infectious Disease Research, Heidelberg. This work was supported by German Center for Infection Research (DZIF) grant 04.704 and the Gilead Sciences GmbH funding to ML. ILS acknowledges the post-doctoral fellowship and funding provided by the Humboldt Foundation.

## Author contributions

ILS, BL, AS, and ML conceived the project. ILS, BL, BG, GS, AS, and ML designed the experiments. ILS, BL, and CP performed *in vitro* experiments and analyzed *in vitro* data. MF, SBi, JB, and SBo performed bioinformatics analysis. CE performed the molecular modeling. FG, LA, and GO performed 3D cell reconstructions. CKF and AT performed proteomic analysis. MS analyzed STED microscopy data. VL performed confocal microscopy image analysis. ILS, BL, AS, and ML wrote the manuscript.

## Conflict of interest

The authors declare that they have no conflict of interest.

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
