## [Review Process File · The EMBO Journal]

Alterations of redox and iron metabolism accompany development of HIV latency

Iart Luca Shytaj, Bojana Lucic, Mattia Forcato, Carlotta Penzo, James Billingsley, Vibor Laketa, Steven Bosinger, Mia Stanic, Francesco Gregoretti, Laura Antonelli, Gennaro Oliva, Christian K Frese, Aleksandra Trifunovic, Bruno Galy, Clarissa Eibl, Guido Silvestri, Silvio Bicciato, Andrea Savarino, Marina Lusic

Review timeline:

Submission date:	9th Apr 2019
Editorial Decision:	8th May 2019
Revision received:	16th May 2019
Editorial Decision:	24th May 2019
Revision received:	24th Nov 2019
Editorial Decision:	17th Dec 2019
Revision received:	1st Feb 2020
Editorial Decision:	10th Feb 2020
Revision received:	11th Feb 2020
Accepted:	14th Feb 2020

Editor: Ieva Gailite

Transaction Report:

1st Editorial Decision

8th May 2019

Thank you for submitting your manuscript to The EMBO Journal. We have now received three referee reports on your manuscript, which are included below for your information. Based on these comments, we unfortunately had to conclude that the study is not a sufficiently strong candidate for publication in The EMBO Journal at this point.

As you will see from the comments, the reviewers in principle express interest in the work and the topic. However, all referees point out that the various aspects of the manuscript are not developed to a sufficient detail to support the proposed mechanism and conclusions, and point out numerous issues with the interpretation and conclusiveness of the data. Given these opinions from three good experts in the field, and since extensive revision work would be required to address all referee concerns, I am afraid that we cannot offer to consider a revised manuscript.

Thank you in any case for the opportunity to consider this manuscript. I am sorry that I cannot communicate more positive news, but I nevertheless hope that you will find the comments of our reviewers helpful.

REFeree REPORTS:

Referee #1:

In the manuscript entitled "Alterations of redox and iron metabolism accompany development of HIV latency", Luca et al. investigate the effects of HIV infection in primary CD4+ T cell, distinguishing productive phase from latency, focusing on both oxidative stress, iron metabolism and PML nuclear bodies. The authors report induction of oxidative stress and NRF2 signaling upon productive infection that decrease after HIV infection, at time of latency. The authors report an activation of anti-oxidant pathway accompanied by enriched expression of iron import genes based on RNAseq and proteomic analyses, as well as a transient decrease in PML protein and nuclear bodies during productive infection. Although use of primary cells favors physiological relevance of their findings, at present, the data presented is too descriptive and correlative for firm conclusions to be drawn.

Major issues:

Interpretation of the data seems often overstretched and many differences are of borderline significance. For unknown reasons, there are often important basal variations in the uninfected controls (Fig. 3D, 4E, 5D), complicating interpretation of the experiments shown.

Many of the reported effects (induction of oxidative stress by HIV infections, PML restriction of HIV, role of PML in HIV latency) have been, at least partially, described, although previously not well-characterized, reducing impact of the data shown.

The mechanism by which HIV replication alters iron metabolism remains still speculative, resulting from complex effects on Tfr1 mainly affected by cell culture and weak effects, if any on FTH-1 or SLC40A1 expression (a decrease in Fig. 4E rather an increase being much more convincing, and starting before latency at D7).

Although the decrease in PML level upon HIV production or FeCl₃ is clear (Fig. 5 and 6A), the result presented in Fig. 6E attempting to link oxidative stress (or iron imbalance) induced by HIV productive infection to PML degradation is not.

Unexpectedly, no inhibition of ROS (using NAC for example) was performed to assess the role of NRF2 on HIV production and on PML degradation. The close reciprocal connections between iron metabolism and oxidative stress renders very challenging attempts to order these events. Similarly, the role of anti-oxidant response in establishment of HIV latency is not explored (nor in PML restoration).

Finally, how HIV mechanistically induces PML decrease is not investigated. This could have allowed the authors to block this HIV effect and assess whether it affects viral production and latency.

The model of insertion of Fe atoms within PML RING seems much too speculative to be presented at this point.

Additional points:

Productive HIV infection is expected to induce cell death, which could account for oxidative stress induction. How the authors avoid these non-specific effects of cell mortality?

In most if not all the immunoblot analyses, the anti-actin loading control is overexposed, precluding any comparison between cell experimental conditions.

A₂O₃ cannot account here as simple "PML depletion" in HIV (Fig. S5D) since arsenic also potently activates oxidative stress. Concentration of A₂O₃ used on those primary cells is very high (5μM), likely inducing broad unspecific effects, in addition to PML targeting.

Referee #2:

Luca and colleagues aim at understanding the mechanisms regulating HIV-1 cytotoxicity and transition to latency. The authors combined multiple methods, including RNAseq, Proteomic, flow cytometry and microscopy to demonstrate that HIV-1 infection upregulates antioxidant and iron import pathways, leading to remodeling of promyelocytic leukemia protein nuclear bodies (PML NBs). As PML NBs inhibit viral replication, its decrease in infected cells favors HIV-1 spread. Interestingly, this also makes infected cells more susceptible to iron chelation, providing a potential target to eliminate infected cells. This is an impressive amount of work. Unfortunately, the presentation of the results is confusing, and the data are sometimes over-interpreted. The work is promising, but fails to be fully convincing at this stage.

General comments:

1. In kinetic experiments day 0 is never shown. This makes difficult the interpretation of the results. The results are sometimes but not always normalized to the mock condition. Overall, it is difficult to know what are the actual levels of expression of the genes/proteins studied.
2. The authors attempted to confirm the *in vitro* RNAseq findings with an *in vivo* dataset in macaques. However, the *in vivo* results were obtained on bulk PBMCs, in which the frequency of infected cells is known to be very low. Thus, the authors should be more cautious when concluding on gene expression in infected cells *in vivo*.
3. Addition of gene knock down or silencing to decrypt the role of specific proteins/pathways would help understanding their exact contribution. For instance, targeting TfR1, Nrf2, HMOX or PML knock would be of high value.

Specific comments

1. Figure 1: The proportion of HIV-1 latently infected cells in the *in vitro* model is unclear. HIV-1 DNA should be expressed as copies per million of cells rather than normalized to day 3. PHA reactivation of Gag- cells at day 14 could be performed to assess latency.
2. Figure 2B: the authors conclude that the cellular response to oxidative stress is decreased with establishment of latency. They should tune down this conclusion. Figure 2B clearly shows that day 14 is not different from day 9.
3. Associated to Figure 3: The author report that GSH/GSSG ratio is decreased in infected cells versus mock control cells. The data should be shown, and the number of donors clearly stated.
4. Figure 3E: it is not surprising that delta Env and delta Tat viruses display no difference at day 5. These viruses do not replicate. It is likely that samples from day 5 and 7 do not carry infected cells. The frequency of infected cells in the culture should be depicted. Overall, it is unclear whether viral replication by itself is needed or if it is only a matter of number of infected cells and assay sensitivity.
5. Figure 4: How do the author explain the discrepancy between TfR1 measurement by FACS and IF?
6. Figure 4C: in the text the authors state "We observed an early decrease in TfR1 expression in infected cells at 3 dpi, which was most likely secondary to intracellular iron release through the action of HMOX-1 " There is no data on intracellular iron release supporting this claim.
7. Figure 4E: the increase of FTH-1 "early upon infection" is not convincing. It seems to be the case only at day 5. Quantification with several donors is required to support this claim.
8. Figure 4F does not display an "enhanced HIV-1 gag p24 production" but rather an increase in the frequency of Gag+ cells. Moreover, the enhancing effect of Fe is a key result, it should be validated by adding more time points and various doses of viruses.
9. Figure S5: why productively infected cells (GFP+) are not all positive for viral integration?

Referee #3:

Shytaj et al. studied oxidative stress, iron transport and PML in primary CD4 T cells infected with HIV-1. They used a model of infection that extended for 14 days when HIV-1 latency started to become apparent. Although most cells died over the 14 days period, a few seemed to survive that contained an integrated yet transcriptionally silent provirus. Samples were analysed by RNAseq and proteomics to compare changes of expression of specific gene pathways (i.e. GO "cellular response to oxidative stress" or "PML pathway"). Overall, the oxidative stress response pathway, controlled by transcription factor Nrf2, was upregulated in productively infected cells but this tended to normalize at 14 days post infection. Similar results were observed in PBMCs from SIVmac infected macaques before and after antiretroviral therapy (ART). Analysis of a subset of 6 oxidative stress genes by qPCR supported the RNAseq data and infection with two different defective viruses demonstrated that the oxidative stress response was triggered only by viral replication. Iron homeostasis was also examined, showing an overall signature of enhanced intracellular iron uptake during productive infection. Iron controls PML stability and PML has been proposed to promote latency. Thus, the authors examined mRNA and protein levels of PML in their experimental setting. PML degraded during active virus replication but it stabilized at 14 days post-infection. The authors draw on these lines of evidence to argue that HIV-1 replication triggers an oxidative stress response and induce iron uptake in CD4 T cells, converging on the degradation of PML, which in turn may regulate entry into latency.

The study reports a series of interesting observations on the relationship between HIV-1 infection, the oxidative stress response and iron homeostasis, an area that is little investigated. The notion that cells more resistant to oxidative stress might go on to survive and become latently infected is intriguing.

Major concerns.

Unfortunately, the mechanistic connection with latency itself is tenuous and mostly unsupported by the data presented. Indeed, there is no clue that perturbing the oxidative stress response or iron homeostasis affects either the establishment of latency or exit from it. Some of the results seem to be overinterpreted and would benefit from clarification.

1. Figure 2A shows a modest upregulation of the oxidative response pathway (mock relative to infected), which appears to be stable until day 14. Figure 2B then shows progressive downregulation of the same pathway when comparing "productive vs latent" cells. However, here they perform the analysis up to day 9, before latency becomes apparent (Figure 1B-D). Furthermore, there is no way the authors can separate latently infected cells from uninfected cells from dying cells in the experimental setting described, even at day 14. In contrast to the data in primary human CD4 T cells, the data on simian PBMCs are clearer but of course again there is no way to know which cells were latently infected. Thus, the connection between the oxidative stress response and latency is weak. Instead, it is quite clear that productive HIV-1/SIV infection does trigger an oxidative stress response.

2. In general, the mapped, filtered and quantified RNAseq data should be provided in supplementary tables for all genes and for the subsets of genes shown in the Figures. This is critical to evaluate the specificity of the responses. Statistical inference should be applied to determine if the pathways in question are changed relative to other cellular pathways in the first place.

Minor concerns that should be addressed

3. In Figure 3C, it is not clear what was used to normalize mRNA expression. GAPDH is unsuitable because its own expression changes upon oxidative stress (see for example Avery S. *Biochem. J.*, 2011 434: 201).

4. In Figure 4A, the iron homeostasis gene network in human cells has an opposite trend to simian PBMCs. This needs clarification. In panel B, it appears that the day 14 pi is not significant but this is relative to what? To the 9 days or to 3 days pi? There seem to be robust changes between 3 and 14 days pi.

5. Supplementary Figure 5D is not convincing and probably unnecessary.

6. A simple marker of oxidative stress is phosphorylated eIF2 α . It should be examined during the course of infection

1st Revision - authors' response

16th May 2019

Once again I would like to thank you for providing a timely review with the comments of our manuscript, which we found very useful. While all reviewers raised concerns about the interpretation of some of the results, we were encouraged by the fact that they all agreed that work topic to be of interest and with potentially broad impact, if the conclusions are strengthened.

We have looked carefully at the comments and we are confident that we can ameliorate our manuscript by addressing these points in a reasonable timeframe.

In particular, we now have a chance to sort infected cells in BSL3 conditions with a state-of-the-art sorter. This could allow us to address all the comments which rightly pointed out that non-infected/dead cells in the infected culture could represent a source of potential bias.

Moreover, we have introduced in the lab the FANA silencing technology, which could help us with the PML knock-down experiment, while avoiding the fundamental problems arising from electroporation (or use of lentiviruses) to transfect primary CD4+T cells, as both introduce significant oxidative stress or cell death.

I am sending you here a detailed point by point response and description on how we plan to respond to each of the questions (in blue, in the attached document), if given a chance to revise our manuscript for your journal.

--

Reviewer 1.

In the manuscript entitled "Alterations of redox and iron metabolism accompany development of HIV latency", Luca et al. investigate the effects of HIV infection in primary CD4+ T cell, distinguishing productive phase from latency, focusing on both oxidative stress, iron metabolism and PML nuclear bodies. The authors report induction of oxidative stress and NRF2 signaling upon productive infection that decrease after HIV infection, at time of latency. The authors report an activation of anti-oxidant pathway accompanied by enriched expression of iron import genes based on RNAseq and proteomic analyses, as well as a transient decrease in PML protein and nuclear bodies during productive infection. Although use of primary cells favors physiological relevance of their findings, at present, the data presented is too descriptive and correlative for firm conclusions to be drawn.

Major issues:

Interpretation of the data seems often overstretched and many differences are of borderline significance. For unknown reasons, there are often important basal variations in the uninfected controls (Fig. 3D, 4E, 5D), complicating interpretation of the experiments shown.

Many of the reported effects (induction of oxidative stress by HIV infections, PML restriction of HIV, role of PML in HIV latency) have been, at least partially, described, although previously not well-characterized, reducing impact of the data shown.

It is true that previous studies had focused on some elements of the pathways that we examined in our work. We believe that the main novelty of our manuscript is the attempt to provide a general picture of the influence of redox metabolism in HIV-1 latency establishment, together with the finding that cells surviving productive HIV-1 infection are characterized by enhanced antioxidant and iron import capacity. This was also recognized by Reviewer n3. Moreover, it is true the role of PML in latency maintenance was previously described by us, and the interplay between redox metabolism and PML stability in cancer was described by the group of de Thé. However, the influence of oxidative stress on PML stability in HIV-1 infection is here shown, to our knowledge, for the first time, and so is the role of iron in influencing PML content. We agree with the Reviewer that more evidence is required to validate and deepen our conclusions, and we will try to address all specific questions as described below. We also thank the Reviewer for acknowledging the potentially higher relevance derived from working exclusively with primary cells. This advantage was also a limitation which impaired our ability to perform some functional experiments. In some cases we believe that additional technologies recently adopted in the lab might allow us to satisfactorily address the questions.

1) The mechanism by which HIV replication alters iron metabolism remains still speculative, resulting from complex effects on TfR1 mainly affected by cell culture and weak effects, if any on

FTH-1 or SLC40A1 expression (a decrease in Fig. 4E rather an increase being much more convincing, and starting before latency at D7).

In the pathway that we analyze in the paper, both *in-vitro* and *ex-vivo* data (Figure 4B) point towards upregulation of iron import capacity during transition to latency. In this regard, upregulation of TfR1 at the peak of infection (day 7) was indicated as a key finding, because all of our approaches showed TfR1 as the main iron import protein in CD4+ T-cells (*i.e.* the expression of which could be readily detected in all assays performed). This is in line with the downregulation of FTH-1 and SLC40A1 (Figure 4E and S3D) that the Reviewer notices at day 7, because these proteins respond with divergent trends to the intracellular levels of iron. Specifically, there is more FTH-1 and SLC40A1 and less TfR-1 when the intracellular level of iron is high and more TfR-1 and less FTH-1 and SLC40A1 when iron is low.

We agree with the Reviewer that more can and needs to be done to strengthen these conclusions. In particular, to specifically address this question we plan to:

- 1) Provide quantification analyses for SLC40A1 and FTH-1
- 2) Assess the role of TfR1 upregulation in sustaining the survival of infected cells by promoting synthesis of deoxyribonucleotides through ribonucleotide reductase. While this link was suggested in the paper based on literature, we will analyze it experimentally to address the point highlighted by the Reviewer. To test this we want to perform experiments in which dNTPs are exogenously provided along with deferiprone in order to rescue the survival of infected cells induced by iron chelation, and, conversely, to test the effects of inhibitors of ribonucleotide reductase (such as hydroxyurea) to analyze whether they can induce effects similar of those of iron chelation on infected cells.

We also believe that the experiments requested by Reviewer 2 (see points 3 and 9), which we will attempt, could help to address this point and to further unravel the mechanisms behind these phenomena.

- 2) Although the decrease in PML level upon HIV production or FeCl₃ is clear (Fig. 5 and 6A), the result presented in Fig. 6E attempting to link oxidative stress (or iron imbalance) induced by HIV productive infection to PML degradation is not.

We agree with the reviewer that the experiment performed with iron chelators on HIV-1 infected cells is less clear than the one performed with the iron donor on uninfected cells. This stems mainly from some limitations in the experimental design which, we will try to improve. In particular the main problem lies in the timing of the treatment for the following reasons:

- 1) PML depletion is maximal between 7-9 days post-infection (Figure 5D)
- 2) Iron chelation leads to a significant increase in the mortality of infected cells starting from day 7 post-infection (Figure S6 E).

Thus, in order to avoid the bias of increased mortality of cells treated with the iron chelator, we had decided to analyze PML levels at day 5, where, however, the baseline difference in PML levels between mock-infected and infected cells is lower.

To address the question of the reviewer we plan to change the experimental setup and introduce the iron chelator at day 5 post-infection, while testing PML levels at days 7 or 9 post-infection. This might allow us to preserve a similar viability among the different conditions analyzed while highlighting PML differences in the most appropriate time points.

- 3) Unexpectedly, no inhibition of ROS (using NAC for example) was performed to assess the role of NRF2 on HIV production and on PML degradation. The close reciprocal connections between iron metabolism and oxidative stress renders very challenging attempts to order these events. Similarly, the role of antioxidant response in establishment of HIV latency is not explored (nor in PML restoration).

We agree that these experiments would be quite informative. We did not proceed with NAC because, in our hands, NAC treatment of CD4+ T-cells resulted in discordant effects on the

GSH/GSSG ratio (one main marker of oxidative stress), which was, depending on the experiment, increased or decreased following NAC treatment.

To perform these experiments we plan to:

- 1) combine NAC with other antioxidants to increase the chance of reproducibly decreasing oxidative stress *in vitro* and then check PML levels with or without antioxidant treatment.
 - 2) check the effect of this antioxidant treatment on viral production/integration
- 4) Finally, how HIV mechanistically induces PML decrease is not investigated. This could have allowed the authors to block this HIV effect and assess whether it affects viral production and latency.

Based on previous literature we expect PML decrease to be mediated by sumoylation and/or proteasomal degradation. While blocking proteasome function would be feasible in primary cells and we do plan to include this experiment in our reply, testing sumoylation levels in primary cells would be very challenging due to the very high number of infected cells required. We can, however, try to test sumoylation levels in cell lines infected with HIV-1. These experiments, along with the experiments with antioxidants described in the reply to question 3, would in our opinion allow a *bona fide* reconstruction of the molecular mechanism of PML depletion upon HIV-1-induced oxidative stress.

5) The model of insertion of Fe atoms within PML RING seems much too speculative to be presented at this point.

We agree that the *in-vivo* relevance of the binding model proposed is still unknown. As the models of degradation of PML by arsenic have been heavily investigated (Zhang et al. 2010, Zhou et al. 2015, Wang et al. 2018) and both direct and indirect mechanisms seem to be involved, we thought that a preliminary *in-silico* hypothesis on the mechanism of action of iron could help to initiate further studies on the topic. We would thus try to move this figure to supplementary information and underline its preliminary nature. In case the Reviewer should still find it not fitting, we will remove it.

6) Productive HIV infection is expected to induce cell death, which could account for oxidative stress induction. How the authors avoid these non-specific effects of cell mortality?

Throughout the paper, we tried to reduce the bias deriving from bystander effects, by coupling single molecule FISH and IF to allow for single cell visualization of the target of interest (e.g. Nrf2 or PML).

The Reviewer is correct in saying that cell mortality might influence the generation of oxidative stress. As an additional experiment to isolate the impact of cell death, we will try to sort live cells and test the regulation of antioxidant species in these sorted cells.

However, we believe that completely isolating the effect of cell death would not be possible and could lead to less physiological conditions. Cell death, as well as oxidative stress, are characterized by strong bystander effects, and *in-vivo* acute HIV-1 infection (which is roughly modeled by our productively infected cells) leads to a drop in CD4 counts which includes infected but also many non infected cells. Thus, the possibility that HIV-1 induced oxidative stress might be reinforced by cell death cannot, in our opinion, be completely excluded and cell death might be a relevant *in-vivo* contributor of this effect.

7) In most if not all the immunoblot analyses, the anti-actin loading control is overexposed, precluding any comparison between cell experimental conditions.

We believe that the Reviewer's criticism is directed at the actin of blots in figures 4D, 5E, S4A and S5C. We will repeat these blots with lower exposure times.

8) A2O3 cannot account here as simple "PML depletion" in HIV (Fig. S5D) since arsenic also potently activates oxidative stress. Concentration of A2O3 used on those primary cells is very high (5uM), likely inducing broad unspecific effects, in addition to PML targeting.

The Reviewer is right that arsenic has other effects aside from PML depletion. We did not include PML knock down experiments, because when we tried to perform them, the use of electroporation was associated with extremely high cell mortality in primary cells. The use of lentiviruses for the same purpose is obviously not an option when studying HIV-1.

However, we have recently adopted in the lab the FANA silencing technology which is aimed at knocking down a target gene without the need for electroporation or transfection reagents. This experiment should allow us to specifically isolate the effect of PML depletion.

Reviewer 2.

Luca and colleagues aim at understanding the mechanisms regulating HIV-1 cytotoxicity and transition to latency. The authors combined multiple methods, including RNAseq, Proteomic, flow cytometry and microscopy to demonstrate that HIV-1 infection upregulates antioxidant and iron import pathways, leading to remodeling of promyelocytic leukemia protein nuclear bodies (PML NBs). As PML NBs inhibit viral replication, its decrease in infected cells favors HIV-1 spread. Interestingly, this also makes infected cells more susceptible to iron chelation, providing a potential target to eliminate infected cells. This is an impressive amount of work. Unfortunately, the presentation of the results is confusing, and the data are sometimes over-interpreted. The work is promising, but fails to be fully convincing at this stage.

We are very grateful to this Reviewer for the appreciation of the amount of work herein presented. We hope that taking into account all the Reviewers' comments we can provide a revised, and clearer version of the manuscript, with the appropriate interpretation of the obtained results, which will make our work fully convincing.

General comments:

1) In kinetic experiments day 0 is never shown. This makes difficult the interpretation of the results. The results are sometimes but not always normalized to the mock condition. Overall, it is difficult to know what are the actual levels of expression of the genes/proteins studied.

We did not include day 0, because the main focus of our comparisons was between matched mock-infected and infected samples, and at day 0 no infection markers are yet visible. While it would be difficult to repeat all the analyses performed to include day 0, we can add some experiments where day 0 is considered in order to address this question.

In terms of normalization, we apologize that our description of the results was not always clear. For all experiments where this was applicable, a comparison between mock infection and infection was provided. However, in some specific cases (e.g. Figure 2B) we also provided additional comparisons to highlight the differences between latent and productively infected time points. We will try to clarify better the normalizations used and make sure that the comparisons between mock infection and infection are always clearly stated.

2) The authors attempted to confirm the *in vitro* RNA Seq findings with an *in vivo* dataset in macaques. However, the *in vivo* results were obtained on bulk PBMCs, in which the frequency of infected cells is known to be very low. Thus, the authors should be more cautious when concluding on gene expression in infected cells *in vivo*.

We agree with the reviewer that *in-vivo* the proportion of infected cells is low. As mentioned in the answer to reviewer n1 (please see answer to comment n6) bystander effects are inherently part of oxidative stress and cannot be isolated *in vivo* and only partially *in vitro*. In this regard, we have attempted in our manuscript to highlight the specific contribution of infected cells *in vitro* by coupling FISH and IF (Figure 3B, Figures 5E-G and Figure S5E). By doing so, we believe that we managed to prove that both PML depletion/reformation and Nrf2 nuclear localization are more pronounced in infected cells.

Overall, a complete exclusion of bystander effects would not be feasible and perhaps not physiological, but we will try to clarify this point better in the revised manuscript and to caution on the need to interpret the results in light of the low number of *in-vivo* infected cells as rightly pointed out by the reviewer.

3) Addition of gene knock down or silencing to decrypt the role of specific proteins/pathways would help understanding their exact contribution. For instance, targeting Tfr1, Nrf2, HMOX or PML knock would be of high value.

As mentioned in the reply to question 8 of reviewer n1, knock down strategies are not trivial to apply to primary cells without causing very high cell mortality and/or oxidative stress. We have now introduced FANA silencing technology in our lab to try to overcome the problems derived from electroporation and transfecting agents. We will apply this technology to knock down PML, thus trying to address the questions of both reviewers. As FANA oligos are very expensive and the other knock down targets suggested are anyway associated with oxidative stress, we plan to test at least one or two (Tfr1 and/or Nrf2, because due the central cellular role of these targets they are less likely to be neutralized by compensatory pathways) of the other targets by using siRNAs in T-cell lines. We believe that this could be a reasonable approach to combine insight and feasibility and address the question raised by the reviewer.

4) Figure 1: The proportion of HIV-1 latently infected cells in the in vitro model is unclear. HIV-1 DNA should be expressed as copies per million of cells rather than normalized to day 3. PHA reactivation of Gag- cells at day 14 could be performed to assess latency.

We agree that the proportion of latently infected cells would be an important information to include. Assays testing integrated viral DNA usually rely on standard curves based on cell lines harboring one copy of viral DNA per cell to yield absolute numbers of integrates. As integration sites in these cell lines are typically unique, this approach can lead to biases, and to avoid those, we decided to describe results in relative terms.

Since we are now able to sort cells in BSL-3 conditions we plan to address this question by sorting p24⁺ cells and use them to make a standard curve based on primary cells. This standard curve will then be used for expressing the number of HIV-1 DNA integrates per cell. This will also provide an estimate of the total viral reservoir at day 14 post-infection. In addition to this, the reactivation with α -CD3-CD28 (already in the paper, Figure 1F) and with PHA (suggested by the reviewer) should provide an estimate of the reservoir which can be reactivated.

5) Figure 2B: the authors conclude that the cellular response to oxidative stress is decreased with establishment of latency. They should tune down this conclusion. Figure 2B clearly shows that day 14 is not different from day 9.

We agree that there is little difference between day 9 and 14 and will mention this and tune down this conclusion

6) Associated to Figure 3: The author report that GSH/GSSG ratio is decreased in infected cells versus mock control cells. The data should be shown, and the number of donors clearly stated.

We will include this figure in the revised manuscript.

7) Figure 3E: it is not surprising that delta Env and delta Tat viruses display no difference at day 5. These viruses do not replicate. It is likely that samples from day 5 and 7 do not carry infected cells. The frequency of infected cells in the culture should be depicted. Overall, it is unclear whether viral replication by itself is needed or if it is only a matter of number of infected cells and assay sensitivity.

According to our Alu-PCR data, the delta Env mutant carries infected cells (Figure S2C). In line with the answer to question n4, we will generate a standard curve to derive an absolute number of viral integrates and thus an estimate of the proportion of infected cells. We will also include an additional mutant (delta Nef), which was not added in the first version of the paper for simplicity, and which is able to replicate, but does not lead to increased antioxidant responses in infected cells.

8) Figure 4: How do the author explain the discrepancy between Tfr1 measurement by FACS and IF?

It is true that, among all time points considered in our analyses, day 5 shows some discrepancy. We believe this discrepancy to be due to the inherent variability of working with different sets of donors

(overall six donors) in primary cells with two different assays. We think that the strength of our analysis lies in the fact that both data sets show a biphasic regulation of TfR1 expression in infected cells compared to mock infected ones, with lower relative expression of TfR1 early upon infection followed by an increase along with enhanced viral replication. As day 5 is an intermediate point of the biphasic response, this may explain the variability observed at this time point. We will include a mention of the divergent time point for more clarity.

9) Figure 4C: in the text the authors state "We observed an early decrease in TfR1 expression in infected cells at 3 dpi, which was most likely secondary to intracellular iron release through the action of HMOX-1 " There is no data on intracellular iron release supporting this claim.

The Reviewer is right in that we do not have a direct measure of intracellular iron release. As mentioned in the reply to the question 1 of Reviewer n1, it is generally accepted that TfR1, FTH-1 and SLC40A1 represent reliable surrogate markers of intracellular iron content. The pattern of expression of these markers at day 3 suggests that infected cells are characterized by an early increase in intracellular free iron levels.

While there are kits available for measuring intracellular iron, those that we tested were not reliably working in primary cells, and this appears to be a common phenomenon (Galy B.).

We will avoid references to iron release and mention in the revised version the likely increased iron content as inferred from surrogate markers of iron turnover. Appropriate references for these markers will be included [Harford, J. B., Rouault, T. A. and Klausner, R. D. (1994). The control of cellular iron homeostasis. In: Brock, J. H., Halliday, J. W., Pippard, M. J. and Powell, L. W. eds, Iron Metabolism in Health and Disease. WB Saunders Co.:London, pp. 123±150. Trowbridge, I. A. (1995). Overview of CD71. In: Schlossman, S. F., Boumsell, L., Harlan, J. M., Kishimoto, T., Morimoto, C., Ritz, J., Shaw, S., Silverstein, R., Springer, T., Tedder, T. F., and Todd, R. F. eds, Leukocyte Typing V: White Cells Differentiation Antigens. Oxford University Press: New York, pp. 1139±1141. Pantopoulos, K., Weiss, G. and Hentze, M. W. (1994). Nitric oxide and the post-transcriptional control of cellular iron. Trends Cell Biol. ,4, 82±86. Ward DM, Kaplan J. Ferroportin-mediated iron transport: expression and regulation. *Biochim Biophys Acta*. 2012;1823(9):1426–1433. doi:10.1016/j.bbamcr.2012.03.004].

10) Figure 4E: the increase of FTH-1 "early upon infection" is not convincing. It seems to be the case only at day 5. Quantification with several donors is required to support this claim.

We will include quantifications for this blot (please see also the response to question n1 of Reviewer n1).

11) Figure 4F does not display an "enhanced HIV-1 gag p24 production" but rather an increase in the frequency of Gag+ cells. Moreover, the enhancing effect of Fe is a key result, it should be validated by adding more time points and various doses of viruses.

The reviewer is correct that the phrasing of this sentence was not precise. We will modify it and include additional time points and viral doses.

12) Figure S5: why productively infected cells (GFP+) are not all positive for viral integration?

This is a drawback inherent to dual color constructs, and which could be due to various reasons such as recombination, transcriptional interference or lower content of mKO2 produced under the control of the EF1 α cellular promoter compared to the LTR-driven GFP expression. Issues in this sense have been recently described in the literature (Battivelli et al. Elife 2018).

Despite this limitation, we think that the ability to visualize the productive and latent states, coupled with the analysis on the wild type virus, can provide some useful information on the effects of arsenic-mediated PML depletion on the expression of HIV-1. As our analysis is based on relative comparisons (untreated vs arsenic treated), the final conclusions should not be affected by this limitation of the model.

Reviewer 3

Shytaj et al. studied oxidative stress, iron transport and PML in primary CD4 T cells infected with HIV-1. They used a model of infection that extended for 14 days when HIV-1 latency started to become apparent. Although most cells died over the 14 days period, a few seemed to survive that contained an integrated yet transcriptionally silent provirus. Samples were analysed by RNA Seq and proteomics to compare changes of expression of specific gene pathways (i.e. GO "cellular response to oxidative stress" or "PML pathway"). Overall, the oxidative stress response pathway, controlled by transcription factor Nrf2, was upregulated in productively infected cells but this tended to normalize at 14 days post infection. Similar results were observed in PBMCs from SIVmac infected macaques before and after antiretroviral therapy (ART). Analysis of a subset of 6 oxidative stress genes by qPCR supported the RNA Seq data and infection with two different defective viruses demonstrated that the oxidative stress response was triggered only by viral replication. Iron homeostasis was also examined, showing an overall signature of enhanced intracellular iron uptake during productive infection. Iron controls PML stability and PML has been proposed to promote latency. Thus, the authors examined mRNA and protein levels of PML in their experimental setting. PML degraded during active virus replication but it stabilized at 14 days post-infection. The authors draw on these lines of evidence to argue that HIV-1 replication triggers an oxidative stress response and induce iron uptake in CD4 T cells, converging on the degradation of PML, which in turn may regulate entry into latency.

The study reports a series of interesting observations on the relationship between HIV-1 infection, the oxidative stress response and iron homeostasis, an area that is little investigated. The notion that cells more resistant to oxidative stress might go on to survive and become latently infected is intriguing.

Major concerns.

Unfortunately, the mechanistic connection with latency itself is tenuous and mostly unsupported by the data presented. Indeed, there is no clue that perturbing the oxidative stress response or iron homeostasis affects either the establishment of latency or exit from it. Some of the results seem to be overinterpreted and would benefit from clarification.

We thank the Reviewer for highlighting the novelty of the study. We believe that, also by replying to the comments of the other Reviewers (please see in particular replies to comment 3 of Reviewer n1 and comments 3 and 11 of Reviewer n2) as well as including latency reversal experiments with drugs able to manipulate ROS or iron content, we can address this point.

1) Figure 2A shows a modest upregulation of the oxidative response pathway (mock relative to infected), which appears to be stable until day 14. Figure 2B then shows progressive downregulation of the same pathway when comparing "productive vs latent" cells. However, here they perform the analysis up to day 9, before latency becomes apparent (Figure 1B-D). Furthermore, there is no way the authors can separate latently infected cells from uninfected cells from dying cells in the experimental setting described, even at day 14. In contrast to the data in primary human CD4 T cells, the data on simian PBMCs are clearer but of course again there is no way to know which cells were latently infected. Thus, the connection between the oxidative stress response and latency is weak. Instead, it is quite clear that productive HIV-1/SIV infection does trigger an oxidative stress response.

The Reviewer is right to point out that we did not isolate the effect of latently infected cells. As mentioned in the replies to reviewers n1 and n2 (questions 6 and 2, respectively) bystander effects are important contributors to oxidative stress. In the sections describing Nrf2 activation and PML decrease/reformation we tried to control for these effects by performing single cell FISH/IF experiments. As this approach is, to our knowledge, one of the few tools to tackle this question, we plan to extend it in the revised version by assessing the content of Nrf2 or by using a probe for ROS in latently infected cells.

As for Figure 2B, the follow-up did not stop at day 9, but rather expression and gene enrichment at day 14 were used as a benchmark to compare corresponding values during productive infection. As also in this case our exposition was not clear enough, we will try to improve the clarity of the Figure and the description of this analysis.

2) In general, the mapped, filtered and quantified RNA Seq data should be provided in supplementary tables for all genes and for the subsets of genes shown in the Figures. This is critical to evaluate the specificity of the responses. Statistical inference should be applied to determine if the pathways in question are changed relative to other cellular pathways in the first place.

We will include this information in the revised manuscript.

3) In Figure 3C, it is not clear what was used to normalize mRNA expression. GAPDH is unsuitable because its own expression changes upon oxidative stress (see for example Avery S. *Biochem. J.*, 2011 434: 201).

We used GAPDH for these analysis, but in accordance with the point raised by the reviewer we will include a second housekeeping gene (18S). We already have in part this data available.

4) In Figure 4A, the iron homeostasis gene network in human cells has an opposite trend to simian PBMCs. This needs clarification. In panel B, it appears that the day 14 pi is not significant but this is relative to what? To the 9 days or to 3 days pi? There seem to be robust changes between 3 and 14 days pi.

In our view, the trend is not opposite, rather, in simian PBMCs the GSEA analysis highlights that the leading genes which drive the upregulation of the iron homeostasis pathway upon ART treatment (Figure 4A) are mostly associated with iron import (Figure 4B). As iron homeostasis is regulated by sometimes opposing mechanisms of import and export, we decided to study these two phenomena separately for further analyses by specifically assessing both markers of import (TfR1) and export (SLC40A1).

As more than one reviewer mentioned that our exposition of the data lacked clarity, we will try to improve the description of these results.

For the statistics, the dot size in the enrichment part of Figure 4B represents the Fisher test significance. The Fisher test does not compare days between each other, but rather the enrichment of the genes significantly up-regulated in infected vs matched mock-infected controls relative to the genes of the pathway considered (in this case iron import). Also in this case we will try to improve the clarity of the presentation of these data.

5) Supplementary Figure 5D is not convincing and probably unnecessary.

As other Reviewers have raised different questions about this experiment, our current idea would be to complement this figure with a knock down of PML through FANA oligos (please see answers to questions 7 and 3 of reviewers n1 and 2 respectively). If these additional data should still fail to convince the Reviewer we would then remove this part from the manuscript.

6) A simple marker of oxidative stress is phosphorylated eIF2 α . It should be examined during the course of infection

The suggested marker of oxidative stress is very good and we will check it by western blot.

2nd Editorial Decision

24th May 2019

Thank you for contacting me with a preliminary point-by-point response outlining the scope of a potential revision of your manuscript. I have forwarded your revision proposal to the reviewers and have now received a response from all three original referees. While reviewers #2 and #3 are of the opinion that the proposed revision plan, if successful, would sufficiently improve the manuscript,

reviewer #1 is still not convinced that the new data will sufficiently strengthen the study due to the complex nature of the analysed system.

Since I appreciate from your revision plan and the response of the reviewers that you might be able to address most of the initially indicated issues with the manuscript, you are welcome to submit a revised version of the manuscript if find that you can address all main referee concerns and provide substantial additional support to the proposed mechanism. Please also note that a strong support from the referees will be required to consider the revised manuscript for publication.

2nd Revision - authors' response

24th Nov 2019

Please see next page.

Reviewer 1.

In the manuscript entitled "Alterations of redox and iron metabolism accompany development of HIV latency", Luca et al. investigate the effects of HIV infection in primary CD4+ T cell, distinguishing productive phase from latency, focusing on both oxidative stress, iron metabolism and PML nuclear bodies. The authors report induction of oxidative stress and NRF2 signaling upon productive infection that decrease after HIV infection, at time of latency. The authors report an activation of anti-oxidant pathway accompanied by enriched expression of iron import genes based on RNAseq and proteomic analyses, as well as a transient decrease in PML protein and nuclear bodies during productive infection. Although use of primary cells favors physiological relevance of their findings, at present, the data presented is too descriptive and correlative for firm conclusions to be drawn.

Major issues:

Interpretation of the data seems often overstretched and many differences are of borderline significance. For unknown reasons, there are often important basal variations in the uninfected controls (Fig. 3D, 4E, 5D), complicating interpretation of the experiments shown.

Many of the reported effects (induction of oxidative stress by HIV infections, PML restriction of HIV, role of PML in HIV latency) have been, at least partially, described, although previously not well-characterized, reducing impact of the data shown.

It is true that previous studies had focused on some elements of the pathways that we have examined in our work. We believe that the main novelty of our manuscript is the attempt to provide a general picture of the influence of redox metabolism in HIV-1 latency establishment, together with the finding that cells surviving productive HIV-1 infection are characterized by enhanced antioxidant and iron import capacity. This was also recognized by Reviewer n3. Moreover, it is true that we previously described the role of PML in latency maintenance, while the interplay between redox metabolism and PML stability in cancer was described by the group of de Thé. However, the influence of oxidative stress on PML stability in HIV-1 infection is here shown, to our knowledge, for the first time, and so is the role of iron in influencing PML content. We agree with the Reviewer that more evidence is required to validate and deepen our conclusions, and we tried to address all specific questions as described below.

We also thank the Reviewer for acknowledging the potentially higher relevance derived from working mainly with primary cells. This advantage is, however, accompanied by higher variation between different donors. We tried throughout the paper to minimize the relevance of this physiologic basal variation by including for each analysis and each time point a donor-matched mock infected control. Apart from this, in this revised version, we also addressed some questions on well-validated lymphoid cell line models of latency (J-Lat cells), which allowed us to expand our conclusions while increasing the standardization between biological replicates.

1) The mechanism by which HIV replication alters iron metabolism remains still speculative, resulting from complex effects on Tfr1 mainly affected by cell culture and weak effects, if any on FTH-1 or SLC40A1 expression (a decrease in Fig. 4E rather an increase being much more convincing, and starting before latency at D7).

In the pathway that we analyze in the paper, both *in-vitro* and *ex-vivo* data (*Figure 4B-E of the revised manuscript*) point towards upregulation of iron import capacity during transition to latency, with upregulation of TfR1 and downregulation of FTH-1 and SLC40A1 (*Figure 4E and EV4F of the revised manuscript*) at

peak infection (7-9dpi). This was also the point that that the Reviewer found more convincing. Indeed, these proteins respond with divergent trends to the intracellular levels of iron, with more TfR-1 and less FTH-1 and SLC40A1 when iron is consumed or iron import is required. Following reviewers suggestions, to strengthen these conclusions we:

- Included a protein quantification for FTH-1 and added new blots to corroborate our data for FTH-1 and SLC40A1 (Reply Figure 1A and B; *i.e. Figures 4E and EV4F in the revised manuscript*).
- Clarified the role of iron as a cofactor which explains the need to utilize and import it during viral replication. For this, we expanded the experiment on the effect of iron in increasing the percentage of p24⁺ CD4⁺ T-cells during productive infection (please see reply to question 11 of Reviewer n2), and included a whole new experiment on the potent effect of iron as enhancer, but not inducer, of viral reactivation from latency in two different clones of J-Lat cells (Reply Figure 1C; *i.e. Figure 4G in the revised manuscript*).

We also explored how iron could play this role as cofactor to enhance viral replication. Previous studies had suggested that iron can facilitate HIV-1 replication through two main mechanisms: 1) by increasing viral transcription (Xu *et al.* Retrovirology 2010) and 2) through the role of iron as a cofactor for the function of ribonucleotide reductase (Asbeck *et al.* J. Clinical Virol. 2011). We tested the latter hypothesis in primary CD4⁺ T-cells by using an inhibitor of ribonucleotide reductase (*i.e.* hydroxyurea) along with the iron chelator deferiprone (L1). We observed that iron chelation can synergize with ribonucleotide reductase inhibition in inducing cytotoxicity already after 24 hours, preferentially of p24⁺ cells. This is also in line with the toxic effect of long-term iron chelation in infected cells (*Figure EV4H of the revised manuscript*). These results have been enclosed to this reply, but were not added to the main manuscript since thorough attempts to fully characterize the molecular mechanisms through which iron facilitates HIV-1 replication have already been the focus of several independent manuscripts (*e.g.* van Asbeck *et al.* J. Clinical Virol. 2001; Hoque *et al.* Retrovirology 2009; Xu *et al.* Retrovirology 2010; Debebe *et al.* Molecular Pharmacology 2010).

In this regard, regulation of intracellular iron metabolism during the different phases of HIV-1 infection is the main focus of the iron section of this manuscript. Accordingly, also to address a common criticism of the Reviewers (potential data overinterpretation), we limited our conclusions exclusively to the common denominator of all results obtained, *i.e.* that iron is imported and utilized at peak HIV-1 replication. Specifically, in the revised version we underline that at 7-9 dpi the upregulation of TfR1 in infected cells (coupled with downregulation of SLC40A1 and FTH-1) is driven by the need to import iron. As a consequence, in the oxidized environment induced by the infection (for a measurement of ROS content in infection, please see reply to questions 5 and 11 of Reviewer n2) iron will generate further oxidative stress. This pro-oxidant effect of free iron is well established, due to the fundamental reactions of Fenton chemistry (for an overview of Fenton Chemistry: Wardman *et al.* Radiation Research 1996) and is in line with our data proving that iron is required for PML depletion (Reply Figure 2; *i.e. Figure 5G in the revised manuscript*).

Reply Figure 1. Iron is utilized during productive HIV-1 infection. Panels A,B) representative images of FTH-1 and SLC40A1 (A) and relative protein FTH-1 expression (B) in HIV-1 infected and mock infected primary CD4⁺ T-cells over time. Western blot quantifications in (B) were performed with Fiji-Image J (Schindelin *et al*, 2012), normalized to the housekeeping protein beta-actin and expressed as Log₂ fold change of HIV-1 infected over mock infected cells (mean \pm SEM; n=3). Panel C) Percentage of HIV-1 reactivation (GFP⁺ cells) in J-Lat cells (clones 9.2 and 15.4) left untreated or treated for 48 hours with phorbol ester 12-O-tetradecanoylphorbol-13-acetate (TPA; 5 μ M), the iron donor ferric nitriloacetate (Fe-NTA; 150 μ M) or a combination of the two. Panels D,E) Percentage of viable (D) and p24⁺ (E) CD4⁺ T-cells left untreated or treated for 24 hours with the iron chelator deferiprone (L1; 50 μ M) and/or the ribonucleotide reductase inhibitor hydroxyurea (100 μ M). Cells were treated at 6dpi and assayed by flow cytometry at 7dpi. Viability was measured using a LIVE/DEAD stain.

2) Although the decrease in PML level upon HIV production or FeCl₃ is clear (Fig. 5 and 6A), the result presented in Fig. 6E attempting to link oxidative stress (or iron imbalance) induced by HIV productive infection to PML degradation is not.

We agree with the Reviewer that the experiment performed with iron chelators on HIV-1 infected cells was less clear than the one performed with the iron donor on uninfected cells. This stemmed mainly from some limitations in the experimental design, with the original experiment designed to maximize cell viability over basal PML depletion in infected cells. To address the question of the Reviewer, we repeated the experiment and introduced the iron chelator at 5dpi (instead of 0dpi), while testing PML levels at 7dpi (instead of 5dpi).

The results obtained were similar to those shown in the first version of the paper (Reply Figure 2B; *i.e.* Figure 5G in the revised manuscript), and confirmed the rescue of PML protein expression by iron chelation.

Reply Figure 2. Iron chelation rescues PML depletion upon HIV-1 infection. Panels A,B) PML protein expression at 5dpi (A) or 7dpi (B) in CD4⁺ T-cells infected with HIV-1 or mock infected and left untreated or treated with the iron chelators deferoxamine (+DFOA, 1 μ M) or deferiprone (+L1, 50 μ M). Iron chelators were added at 0dpi (A) or 5dpi (B).

Moreover, to improve the coherence and readability of the manuscript, we now coupled the rescue of PML depletion through iron chelation to that induced by antioxidant supplementation (please see *Figures 5F and G in the revised manuscript* and the response to the next question).

3) Unexpectedly, no inhibition of ROS (using NAC for example) was performed to assess the role of NRF2 on HIV production and on PML degradation. The close reciprocal connections between iron metabolism and oxidative stress renders very challenging attempts to order these events. Similarly, the role of antioxidant response in establishment of HIV latency is not explored (nor in PML restoration).

We thank the Reviewer for these useful suggestions. We performed experiments with two different antioxidants (N-acetylcysteine [NAC] and resveratrol) which showed the ability of antioxidant treatment to rescue PML depletion (Reply Figure 3). Results obtained with NAC treatment were added to the revised manuscript, *Figure 5G*). Moreover, we analyzed the effect of latency reactivation on antioxidant responses (please see response to question 1 of Reviewer n3) and oxidative stress (through ROS content) both in productively and latently infected cells (please see reply to question 5 of Reviewer n2).

Reply Figure 3. Antioxidant treatment rescues PML depletion upon HIV-1 infection. Panels A,B) PML protein expression at 7 dpi in CD4⁺ T-cells infected with HIV-1 or mock infected. Cells were left untreated or treated at 5dpi with resveratrol (10μM, panel A) or NAC (10mM, Panel B).

4) Finally, how HIV mechanistically induces PML decrease is not investigated. This could have allowed the authors to block this HIV effect and assess whether it affects viral production and latency.

Based on previous literature (Sahin et al. Nucleus 2014; Sahin et al. JCB 2014) we explored whether PML decrease could be mediated by sumoylation and/or proteasomal degradation in the setting of HIV-1 infection. To investigate this hypothesis, we first blocked proteasome function using two well established inhibitors, MG132 and bortezomib (Goldberg JCB 2012). We performed this experiment on primary CD4⁺ T-cells as well as in the latently HIV-1 infected cell line J-Lat 9.2, on which we first demonstrated that reactivation of HIV-1 expression leads to depletion of PML protein (Reply Figure 4A, i.e. *Figure 5E in the revised manuscript*). Subsequently, we also show that both proteasome inhibitors could rescue PML NBs degradation in productively infected primary CD4⁺ T-cells and in J-Lat cells in which HIV-1 expression was reactivated from latency (Reply Figure 4B-E, i.e. *Figure EV8 and Figure 7A,B in the revised manuscript*). Furthermore, we showed that inhibition of ubiquitination through the small molecule TAK243, which blocks the ubiquitin activating enzyme (Hyer et al. Nat Med. 2018), can rescue PML NBs number in J-Lat cells upon latency reactivation (Reply Figure 4F,G, i.e. *Figure 7C,D in the revised manuscript*). With this experiment we demonstrated that ubiquitin function is required for PML degradation upon latency reactivation. Moreover, these data offer a missing link between latency reactivation and PML depletion which was not characterized in the original version of the manuscript.

Reply Figure 4. PML degradation induced by HIV-1 replication is mediated by the ubiquitin-proteasome axis.

A) PML protein expression and quantification in latently infected J-Lat 9.2 and 15.4 cells left untreated or in which HIV-1 expression was reactivated for 24 hours with 10 μ M TPA. B) Representative immunofluorescence images and quantifications (C) of PML NBs (green, maximal projection) and mab414/nuclear lamina (red, one Z stack) of CD4⁺ T cells. Mock infected and HIV-1 infected (7 dpi) cells were left untreated or treated for 4 hrs with MG132 (10 μ M) or Bortezomib (100mM). D-G) Representative immunofluorescence images (D,F) and quantifications (E,G) of PML NBs (green, maximal projection) and mab414/nuclear lamina (red, one Z stack) in J-Lat 9.2 cells left untreated (latent), or reactivated for 24 hours with 10 μ M TPA. To block proteasome function (D,E), cells were treated for 4 hrs with MG132 (10 μ M) or Bortezomib (100mM). To block ubiquitination (F,G), cells were treated for 4 hrs using TAK243 (10 μ M). Scale bar 10 μ m. Data were analyzed using One-way ANOVA. Black dash indicates median, n= number of PML NBs. * $P < 0.05$; ** $P < 0.01$; *** $P < 0.001$; **** $P < 0.0001$.

Finally, we investigated the possible colocalization of PML with Small Ubiquitin-like Modifier 2/3 (SUMO 2/3) during latency and upon HIV-1 reactivation. We chose SUMO 2/3 in that PML was previously shown to be preferably an endogenous SUMO 2/3 target (Becker et al. Nat. Struct. Mol. Biol. 2013). By coupling Stimulated emission depletion (STED) and confocal microscopy in J-Lat 9.2 cells, latent or reactivated with TPA (Reply Figure 5A,B, *i.e.* Figure 7 E,F in the revised manuscript) we found that, upon HIV-1 latency reactivation, PML NBs colocalization with SUMO 2/3 was increased. Moreover, we detected increased SUMO2/3 protein (red) colocalizing with PML NBs (green) during productive HIV-1 infection in primary CD4⁺ T cells as compared to mock infected cells (Reply Figure 5C,D, *i.e.* Figure 7 G,H in the revised manuscript).

These data, along with the experiments with antioxidants described in the reply to question 3, allow in our opinion, a *bona fide* reconstruction of the molecular mechanism of PML degradation upon HIV-1 induced oxidative stress.

Reply Figure 5. Enhanced sumoylation in PML NBs upon latency reactivation and productive HIV-1 infection.

A) Representative immunofluorescence images of PML NBs (green) and SUMO 2/3 (red) in J-Llat 9.2 cells left untreated (latent), or reactivated for 24 hours with 10 μ M TPA as imaged by STED microscopy. Scale bar 2 μ m. Data quantification of confocal imaging from the same experiment (B). C) Representative immunofluorescence images and quantification (D) of PML NBs (green, maximal projection) and SUMO 2/3 (red, maximal projection) during mock and productive HIV-1 infection of primary CD4⁺ T-cells at 7dpi. Scale bar 10 μ m. Data were analyzed using unpaired nonparametric Mann-Whitney test. Black dash indicates median, n= number of PML NBs. * $P < 0.05$; ** $P < 0.01$; *** $P < 0.001$; **** $P < 0.0001$.

5) The model of insertion of Fe atoms within PML RING seems much too speculative to be presented at this point.

We agree that the *in-vivo* relevance of the binding model proposed is still unknown. As the models of degradation of PML by arsenic have been heavily investigated (Zhang *et al.* Science 2010, Zhou *et al.* J. Biol. Chem. 2015, Wang *et al.* Nat. Commun. 2018) and both direct and indirect mechanisms seem to be involved, we thought that a preliminary *in-silico* hypothesis on the mechanism of action of iron could help to initiate further studies on the topic. In the revised version of the manuscript this figure was moved to the supplementary information (Figure EV5G in the revised manuscript). Moreover, the result itself is now

presented as a mere suggestion which fits with the role of iron as a ROS enhancer (please see also the response to question n1 and response to question n9 of Reviewer 2).

In case the Reviewer should still find this panel not appropriate, we will remove it.

6) Productive HIV infection is expected to induce cell death, which could account for oxidative stress induction. How the authors avoid these non-specific effects of cell mortality?

Throughout the paper, we tried to reduce the bias deriving from bystander effects, by coupling single molecule FISH and immunofluorescence to allow for single cell visualization of the target of interest (e.g. Nrf2 or PML: *Figures 3F and 6A-C of the Revised manuscript* and ROS: Reply Figure 11; i.e. *Figure 3B of the Revised manuscript*).

The Reviewer is correct in pointing out that cell mortality might influence the generation of oxidative stress. To limit the impact of cell death and address this question, we isolated live cells at three different time points (3, 7 and 14 dpi, Reply Figure 6A, i.e. *Figure EV3C in the revised manuscript*) and validated the main results of the paper (namely regulation of antioxidant gene expression and PML protein content) which could be affected by the cytotoxicity of replicating HIV-1. Our new experiments show that, also when considering only the cell population enriched for viability, transcription of antioxidant genes in infected cells peaks along with the highest level of viral replication (7dpi, Reply Figure 6B, i.e. *Figure EV3D in the revised manuscript*) and is accompanied by depletion of the PML protein (Reply Figure 6C, i.e. *Figure EV7A in the revised manuscript*). These effects are reversed when viral replication ceases (14dpi). These new results, along with our single molecule FISH experiments (Reply Figure 11, i.e. *Figure 3B of the Revised manuscript*), allow to prove that oxidative stress generation and PML depletion are a hallmark associated with viral replication and not a mere consequence of cell death or bystander effects.

Despite this, the possibility that HIV-1 induced oxidative stress might be reinforced by cell death cannot be completely excluded and cell death might also be a relevant *in-vivo* contributor of this effect.

Reply Figure 6. Regulation of antioxidant genes and PML gene and protein expression in primary CD4⁺ T-cells enriched for viability. A) example of the enrichment efficiency of viable cells. Dead cells were removed by magnetic selection of Annexin V⁺ cells, and analyzed by flow cytometry after LIVE/DEAD staining. B,C) mRNA level of main antioxidant genes regulated by Nrf2 (B) and protein level of PML (C) during the transition from productive (3-7 dpi) to latent (14 dpi) infection, as measured by qPCR and western blot, respectively. Assays were performed on CD4⁺ T-cells previously enriched for viability. Raw data in (B) were first normalized using 18S as housekeeping control and then expressed as Log₂ fold mRNA expression in infected vs mock infected cells, calculated as in (Livak & Schmittgen, 2001). Data are expressed as mean±SEM of three donors and were analyzed by two-way ANOVA followed by Tukey's post-test for multiple comparisons. ** *P*<0.01; ****P*<0.001; *****P*<0.0001. Trx= thioredoxin; NQO1= NAD(P)H [quinone] dehydrogenase 1; HMOX-1= heme oxygenase 1; G6PD= glucose-6-phosphate dehydrogenase; GCLC= glutamate-cysteine ligase; TrxR1= thioredoxin reductase 1.

7) In most if not all the immunoblot analyses, the anti-actin loading control is overexposed, precluding any comparison between cell experimental conditions.

We believe that the Reviewer's criticism was mainly directed at the actin of blots in Figure 4D, 5E, and S5C of the original manuscript. We repeated these blots with lower exposure times (Reply Figure 7 and Reply Figure 1A, *i.e.* Figures 5D and 4E in the revised manuscript). Please note that the arsenic blot (Reply Figure 7, right panel), albeit included in this answer, was not included in the revised version of the paper as the whole supplementary file was removed, as per request of Reviewer n3 (question 5).

Moreover we tried to apply lower exposure times to the new blots added in the revised version of the manuscript (please see replies to questions 1-4 of this Reviewer and reply to question 6 of Reviewer 3).

Reply Figure 7. PML expression in activated CD4⁺ T-cells: infected with HIV-1 or mock infected (left) or left untreated or treated for 48 hr with 5μM As₂O₃ (right).

8) A2O3 cannot account here as simple "PML depletion" in HIV (Fig. S5D) since arsenic also potently activates oxidative stress. Concentration of A2O3 used on those primary cells is very high (5uM), likely inducing broad unspecific effects, in addition to PML targeting.

The Reviewer is right that arsenic has other effects aside from PML depletion. As another Reviewer criticized the whole supplementary figure with these experiments (question 5 of Reviewer 3), we have removed these data from the revised version of the manuscript.

In the initial version, we did not include PML knock down experiments, because when we tried to perform them, the use of electroporation was associated with extremely high cell mortality in primary cells.

In order to address this question we used the recently developed FANA oligo technology (for a recent example, Takahashi et al. Molecular Therapy 2019) to silence PML in primary CD4⁺ T-cells (Reply Figure 8A-C).

While we were not able to achieve full silencing of PML, we did obtain a partial depletion with two of the three FANA oligos used (FANA oligos 2 and 3; Reply Figure 9A,B; *i.e.* Figure EV6 in the revised manuscript). A small but consistent increase in the percentage of p24⁺ cells was observed only upon treatment with FANA oligos 2 and 3 in all four donors tested (Reply Figure 8C; Figure EV6C in the revised manuscript). This experimental setup does present some difficulties, mainly due to the incomplete knock down and to the cost which limits the amount of cells that can be used. Consequently, limited space is dedicated to these results and their interpretation in the revised manuscript.

However, despite its limitations, this is, to our knowledge, the first attempt to analyze the effect of PML silencing upon HIV-1 infection in primary cells. Therefore, we included this experiment in the revised version as a support of the main topic of the paper.

Reply Figure 8. Effect of PML knock down at 11dpi on the establishment of HIV-1 latency in primary CD4⁺ T-cells. PML mRNA (A) and protein (B) expression in activated CD4⁺T-cells left untreated or treated for 72 hours with non targeting oligos or three different FANA oligos designed to silence PML. Data were measured by real-time PCR (A) and western blot (B). For (A), raw data were first normalized using 18S as housekeeping control and then expressed as fold mRNA expression in FANA PML vs non targeting oligo calculated as in (Livak & Schmittgen, 2001) (n=3, mean±SEM). (C) Frequency of p24⁺CD4⁺ T-cells at 14dpi. Cells were treated at 11dpi with non targeting oligos or FANA oligos designed to silence PML. Data were measured by flow cytometry. (n=4, mean±range).

Reviewer 2.

Luca and colleagues aim at understanding the mechanisms regulating HIV-1 cytotoxicity and transition to latency. The authors combined multiple methods, including RNAseq, Proteomic, flow cytometry and microscopy to demonstrate that HIV-1 infection upregulates antioxidant and iron import pathways, leading to remodeling of promyelocytic leukemia protein nuclear bodies (PML NBs). As PML NBs inhibit viral replication, its decrease in infected cells favors HIV-1 spread. Interestingly, this also makes infected cells more susceptible to iron chelation, providing a potential target to eliminate infected cells. This is an impressive amount of work. Unfortunately, the presentation of the results is confusing, and the data are sometimes over-interpreted. The work is promising, but fails to be fully convincing at this stage.

We are grateful to this Reviewer for the appreciation of the amount of work herein presented. We took into account all Reviewers' comments and we hope that his revised version has managed to improve our work and made it fully convincing.

General comments:

1) In kinetic experiments day 0 is never shown. This makes difficult the interpretation of the results. The results are sometimes but not always normalized to the mock condition. Overall, it is difficult to know what are the actual levels of expression of the genes/proteins studied.

In the original version of the manuscript, we did not include day 0, because the focus of our comparisons was between matched mock-infected and infected samples. Indeed, at day 0 no infection markers are yet visible. However, we agree that it could be useful to provide an indication of baseline values for markers for which absolute expression can be measured (such as in western blot experiments). While it would have been difficult to repeat all the analyses performed to include day 0, we included this time point for the western blots showing time courses which were added in this revised version.

In particular, day 0 expression is now shown for FTH-1, SLC40A1, PML and eIF2alpha proteins (please see Reply Figure 1A, Reply Figure 7 and Reply Figure 17, *i.e. Figures 4E, 5D and EV3A in the revised manuscript*).

From these experiments, two main conclusions can be drawn:

- The relative expression of each marker at day 0 between mock infected and HIV-1 infected cells is, as expected, comparable.
- eIF2alpha phosphorylation (a marker suggested by Reviewer 3 in his question 6) is strongly dependent on the activation status of cells (*i.e.* at day 0, closer to the activation stimulus provided by α -CD3/CD28 beads, the absolute expression of the marker is much higher).

Thus, we agree with the Reviewer that this inclusion is helpful to highlight markers that are solely (or mostly) altered by the infection, among markers that are dependent on the infection, but also significantly influenced by cell activation. We thank the Reviewer for this useful suggestion.

As for the normalization, we apologize that our description of the results was not always clear. For all experiments where this was applicable, a comparison between mock infection and HIV-1 infection was provided. Only in one case (Figure 2B of the original manuscript) we also provided additional comparisons to highlight the differences between latent and productively infected time points.

To streamline our data exposition, in this revised version, we show only comparisons between corresponding time points of infected vs mock infected cells of the same donors (unless additional treatments have been performed on the cells). We believe this comparison to be the most informative, because it allows to account for the different activation status of cells over time and for donor-dependent differences in baseline values, which are unavoidable when working with primary cells.

2) The authors attempted to confirm the *in vitro* RNAseq findings with an *in vivo* dataset in macaques. However, the *in vivo* results were obtained on bulk PBMCs, in which the frequency of infected cells is known to be very low. Thus, the authors should be more cautious when concluding on gene expression in infected cells *in vivo*.

We agree with the Reviewer that the *in-vivo* proportion of infected cells is low. As also mentioned in the answer to question n6 of Reviewer n1, bystander effects, which could contribute to the effects observed in macaque PBMCs, are inherently part of oxidative stress and cannot be isolated *in vivo* and can only partially be excluded *in vitro*. In this regard, we have attempted to highlight the specific contribution of infected cells *in vitro* by coupling FISH and immunofluorescence (*e.g.* Nrf2 or PML, *Figures 3F and 6A-C of the revised*

manuscript and ROS, Reply Figure 11; *i.e. Figure 3B of the revised manuscript*). By doing so, we believe that we managed to prove that both PML depletion/reformation and Nrf2 nuclear localization are more pronounced in infected cells. Moreover, in the revised version of the manuscript we have included experiments performed on live cells, to control for the non specific effect produced by dead/dying cells on the upregulation of antioxidant defenses (Reply Figure 6, *i.e. Figures EV3C,D and Figure EV7A in the revised manuscript*). However, we agree that a complete exclusion of bystander effects would not be feasible and perhaps not physiological. We tried to clarify this point in the revised manuscript and to caution on the need to interpret the results in light of the low number of *in-vivo* infected cells as rightly pointed out by the Reviewer. In particular, in the revised discussion we now write:

“Although ex-vivo data from SIVmac infected macaques were in good agreement with the main findings of our work, the low frequency of infected cells in vivo suggests an important role for bystander effects in the phenomena that we observed. However, despite this potential confounding factor, our single cell analyses support the idea that productively infected cells are the drivers of ROS generation and PML degradation.”

3) Addition of gene knock down or silencing to decrypt the role of specific proteins/pathways would help understanding their exact contribution. For instance, targeting Tfr1, Nrf2, HMOX or PML knock would be of high value.

As mentioned in the reply to question 8 of Reviewer n1, knock down strategies are not trivial to apply to primary cells without causing very high cell mortality and/or oxidative stress. To address this question we have silenced HMOX-1 in Jurkat T-cells (Reply Figure 9; *i.e. Figures EVA and 4A in the revised manuscript*) and, by using the FANA silencing technology, PML in primary CD4⁺ T-cells (Reply Figure 8; *i.e. Figure EV6 in the revised manuscript*).

We chose to silence HMOX-1 as this gene was the most upregulated among the Nrf2 targets analyzed, both during productive infection and upon latency reactivation in J-Lat cells (please see Reply Figure 2B, Reply Figure 16, *i.e. Figures 3G,I and Figures EV3B,D in the revised manuscript*). Our results show that HMOX-1 silencing is accompanied by increased HIV-1 expression (Reply Figure 9B, *i.e. Figures 4A in the revised manuscript*) and decreased viability of infected cells (Reply Figure 9C, *i.e. Figures 4A in the revised manuscript*). This is in line with the data of the original manuscript and supports a model in which HMOX-1 upregulation at peak HIV-1 infection is a cellular response to limit viral replication and favor cell survival.

Reply Figure 9. Effect of HMOX-1 knock down on viability and HIV-1 gag mRNA transcription in Jurkat T-cells. Panels A-C) Relative HMOX-1 mRNA expression (panel A), HIV-1 gag mRNA expression (panel B) and cell viability (panel C) in Jurkat T-cells transfected with siRNAs targeting HMOX-1 as compared to control siRNAs. Cells were transfected with 300nM siRNAs by electroporation, infected 24 hours post-transfection with HIV-1_{NL4-3} and assayed 72 hours post-transfection. Data were analyzed by unpaired t-test and are expressed as mean±SEM (n=3).

4) Figure 1: The proportion of HIV-1 latently infected cells in the in vitro model is unclear. HIV-1 DNA should be expressed as copies per million of cells rather than normalized to day 3. PHA reactivation of Gag- cells at day 14 could be performed to assess latency.

We agree that the proportion of latently infected cells is an important information to include. Assays testing integrated viral DNA usually rely on standard curves based on cell lines harboring one copy of viral DNA per cell to yield absolute numbers of integrates. As integration sites in these cell lines are typically unique, this approach can lead to biases, and to avoid those, we decided to describe results in relative terms in the original version of the manuscript.

In this revised version, to address this question and make a standard curve of integrated DNA, we sorted primary p24⁺ CD4⁺ T-cells. The standard curve obtained from DNA of sorted cells was then employed to express the level of integrated HIV-1 DNA as Log₁₀ copies of integrates per 10⁶ cells (Reply Figure 10A-C; *i.e.* Figure EV1 and 1D in the revised manuscript). This analysis confirmed that HIV-1 DNA persists in our model at 14dpi (Reply Figure 10C, *i.e.* Figure 1D in the revised manuscript).

To address the question about reactivation with PHA, we could not use sorted gag negative cells, as intracellular staining for p24 requires cell permeabilization, which would thus not be compatible with further culturing of sorted cells. However, we have included a PHA reactivation of the infected cell culture at 14 dpi (Reply Figure 10D, *i.e.* Figure EV1C in the revised manuscript), and we have described in more detail the features of the model in the latent time point. In particular, we now write in the revised manuscript:

*“In this model, CD4⁺ T-cells are activated through stimulation of CD3/CD28 receptors, infected with HIV-1_{NL4-3}, and monitored over time until returning to a resting state. This time course mimics distinct features of HIV-1 infection in vivo, *i.e.* a rapid initial growth of viral replication accompanied by cell death or establishment of a small pool of cells harboring integrated viral DNA, a fraction of which can be reactivated.”*

Reply Figure 10. Absolute quantification of integrated HIV-1 DNA and latency reactivation with PHA in primary CD4⁺ T-cells. Panel A) FACS sorting strategy of primary p24⁺ CD4⁺ T-cells. Panel B) Linear correlation between delta Ct values and Log₁₀ HIV-1 DNA copies/10⁶ CD4⁺ T-cells. The correlation was calculated based on serial dilutions of the initial input of DNA from sorted p24⁺ CD4⁺ T-cell. Delta Ct values were calculated as in Tan *et al.* J Virol 2006. Serial dilutions of sorted DNA from p24⁺ cells were done using DNA from mock infected cells. Panel C) Number of copies of integrated HIV-1 DNA over time in HIV-1 infected CD4⁺ T-cells. D) reactivation of HIV-1 in primary CD4⁺ T-cells at 14 dpi with 10 μg/mL PHA, measured as proportion of p24⁺ CD4⁺ T-cells by flow cytometry.

5) Figure 2B: the authors conclude that the cellular response to oxidative stress is decreased with establishment of latency. They should tune down this conclusion. Figure 2B clearly shows that day 14 is not different from day 9.

We agree that there is little difference in the graph between day 9 and 14 and, in accordance with the suggestion, we did not stress this decrease in the revised version of the paper. On the other hand, our revised data prove that a partial renormalization of oxidative stress is observed upon transition to latency (Reply Figure 11; *i.e.* Figure 3B in the revised manuscript). In this new experiment we have measured the content of reactive oxygen species (ROS) in mock infected and HIV-1 infected cells during productive (7dpi) and latent (14dpi) infection (Reply Figure 11). In order to increase the relevance of this analysis we further coupled HIV-1 DNA FISH to immunofluorescence for ROS, to focus the measurement of ROS only to cells harboring the virus. This additional analysis also showed that ROS content, and thus oxidative stress, is significantly higher in HIV-1 infected cells during productive infection, and is normalized, at least in part, in latently infected cells.

Reply Figure 11. ROS content in primary CD4⁺ T cells is higher in productively HIV-1 infected as compared to latently and mock infected cells. A,B) representative picture (A) and quantification (B,C) of ROS content in HIV-1 infected or mock infected primary CD4⁺ T-cells at 7dpi (productive infection) or 14dpi (latent infection). ROS content was measured by immunofluorescence, HIV-1 DNA⁺ cells were identified by FISH. Values displayed in (B,C) were calculated as nuclear corrected total cell fluorescence (CTCF as in (McCloy *et al*, 2014)) and analyzed by Kruskal-Wallis test followed by Dunn's post-test (B) or by Mann-Whitney test (C). Black dash indicates median, n= number of cells. ** $P < 0.0001$.**

6) Associated to Figure 3: The author report that GSH/GSSG ratio is decreased in infected cells versus mock control cells. The data should be shown, and the number of donors clearly stated.

We have included this figure in the revised manuscript (Reply Figure 12; *i.e.* Figure 3A in the revised manuscript).

Reply Figure 12. Total to oxidized glutathione ratio (GSH/GSSG) in mock infected or HIV-1 infected CD4⁺ T-cells (7dpi, n. of donors=4). Data are expressed as mean±SEM and were analyzed by paired t-test.

7) Figure 3E: it is not surprising that delta Env and delta Tat viruses display no difference at day 5. These viruses do not replicate. It is likely that samples from day 5 and 7 do not carry infected cells. The frequency of infected cells in the culture should be depicted. Overall, it is unclear whether viral replication by itself is needed or if it is only a matter of number of infected cells and assay sensitivity.

The Reviewer is right that at least one of the mutant viruses (delta tat) carries very little integrated DNA. In order to get a quantitative complement to our Alu-PCR data, we performed absolute quantifications of integrated HIV-1 DNA, using the same experimental setup described in Reply Figure 10. We also tried to include a third mutant (delta Nef), to compare it to the other mutants used in the original version of the paper (delta tat and delta env).

Also in this case, the criticism of the Reviewer stands true, and given the extensive new information (*i.e.* ROS FISH and latency reactivation experiments) now included on this topic in the revised manuscript, we decided not to include the experiments with mutant viruses as they could be potentially skewed by their lower level of integration compared to the wild type.

Reply Figure 13. Integrated viral DNA in CD4⁺ T-cells infected with HIV-1_{NL4-3} mutated in the *nef*, *tat* or *env* genes. Number of copies of integrated HIV-1 DNA were obtained from serial dilutions of sorted p24⁺ CD4⁺ T-cells (similarly to Reply Figure 10) and expressed as Log₁₀ copies/million CD4⁺ T-cells (mean±SEM; n. of donors= 3).

8) Figure 4: How do the author explain the discrepancy between TfR1 measurement by FACS and IF?

It is true that, among all time points considered in our analyses, day 5 shows some discrepancy. We believe this discrepancy to be due to the inherent variability of working with different sets of donors (overall six donors) of primary cells with two different assays. We think that the strength of our analysis lies in the fact that both data sets show an inversion of the trend of TfR1 expression in HIV-1 infected, compared to mock infected cells when viral replication increases. As day 5 is an intermediate point in this viral replication buildup, this may explain the variability observed at this time point.

As mentioned in the reply to question n1 of Reviewer n1 and to the question below, we have now narrowed the interpretation of iron metabolism regulation to be fully adherent to the mechanism which could be highlighted by all experimental approaches: that iron import is increased at peak HIV-1 replication. Please see below for more detail.

9) Figure 4C: in the text the authors state "We observed an early decrease in TfR1 expression in infected cells at 3 dpi, which was most likely secondary to intracellular iron release through the action of HMOX-1 " There is no data on intracellular iron release supporting this claim.

The Reviewer is right in that we do not have a direct measure of intracellular iron release. As also mentioned in the reply to the question 1 of Reviewer n1, it is generally accepted that TfR1, FTH-1 and SLC40A1 represent reliable surrogate markers of intracellular iron content (Harford *et al.* Iron Metabolism in Health and Disease 1994; Trowbridge, I.A. Leukocyte Typing V: White Cells Differentiation Antigens 1995; Pantopoulos K. *et al.* Trends Cell Biol 1994; Ward DM & Kaplan J. Biochim Biophys Acta. 2012).

The pattern of expression of these markers at 3dpi, now further corroborated by the new western blot experiments (Reply Figure 1, *i.e.* Figures 4E-C and Figure EV4F) suggests that infected cells are characterized by an early increase in intracellular iron levels. However, while there are kits available for

measuring intracellular iron, those that we tested were not reliably working in primary cells, and this appears to be a common phenomenon (Galy B.).

In order to address the risk, highlighted by the reviewers, of data overinterpretation, we have attenuated the interpretation of the iron metabolism data, confining it to the conclusions that could be univocally drawn from all different analyses performed.

Briefly, in the revised version, we conclude that increased iron import capacity of infected cells at peak viral replication (TfR1 increase between 7-9 dpi) sustains intracellular iron uptake and thus can enhance PML depletion (for the role of iron on PML, please see *Figure EV5 D-G in the revised manuscript*), through further ROS generation, a well established product of the classic Fenton reaction (Wardman *et al.* Radiation Research 1996). This increased iron import is associated with iron utilization by infected cells (as suggested by SLC40A1 and FTH-1 decrease at 7dpi) due to the role of iron as cofactor to sustain viral replication (as shown by our Reply Figure 1D, Reply Figure 14 and several previous lines of evidence: Asbeck *et al.* J. Clinical Virol. 2001; Hoque *et al.* Retrovirology 2009; Xu *et al.* Retrovirology 2010; Debebe *et al.* Molecular Pharmacology 2010). Furthermore, upon transition to a resting (latent) state, iron import is decreased, in line with downregulation of TfR1 expression in resting cells (*Figure EV4A of the revised manuscript*) and its pro-oxidant effect through the Fenton reaction reduced, in line with the lower content of ROS in latently infected cells (Reply Figure 11, *i.e.* *Figures 3B-D in the revised manuscript*).

We believe that the revised data and their interpretation are sufficient to coherently explain PML dynamics during the different stages of the infection, while remaining adherent to the conclusions that can be drawn by all available experimental evidence.

10) Figure 4E: the increase of FTH-1 "early upon infection" is not convincing. It seems to be the case only at day 5. Quantification with several donors is required to support this claim.

We thank the Reviewer for pointing out this potential concern, which was also raised by Reviewer n1. We have now included quantifications for this blot (Reply Figure 1A, *i.e.* *Figure 4E in the revised manuscript*). Moreover, as pointed out in the previous answer, we have revised the manuscript to base our core conclusions only on the evidence that could be drawn by all experimental approaches and already deemed generally acceptable by all Reviewers. Please refer to the previous two answers for more details.

11) Figure 4F does not display an "enhanced HIV-1 gag p24 production" but rather an increase in the frequency of Gag+ cells. Moreover, the enhancing effect of Fe is a key result, it should be validated by adding more time points and various doses of viruses.

We apologize for our phrasing, which was not precise. We have rewritten the description of these data and included additional time points and viral doses (Reply Figure 14; *i.e.* *Figure 4F in the revised manuscript*). The data show that incubation with iron can increase the frequency of p24⁺ CD4⁺ T-cells, with the effect being more visible when the baseline percentage of p24⁺ cells is higher (*i.e.* samples at 5dpi infected with an input HIV-1 dose of 20ng p24/million cells). This evidence suggests the ability of iron to increase ongoing replication, rather than induce *de novo* viral production. To further characterize the role of iron in the context of HIV-1 replication, we analyzed the effect of iron on latency reactivation in two different clones of J-Lat cells (Reply Figure 1D, *i.e.* *Figure 4G in the revised manuscript*). This experiment proved that, while iron *per se* was only modestly able to induce viral reactivation, coupling iron to the reactivating agent TPA could significantly enhance the amount of viral reactivation obtained. Of note, to reduce the possibility that incomplete iron solubilization/uptake was the cause of absence of reactivation by iron alone, we used the iron donor ferric nitriloacetate (Fe-NTA) which is known to significantly increase intracellular iron content on a broad pH range (Byrd & Hornitz JCI 1991).

Overall our data show, in line with previous evidence, that iron is utilized during ongoing HIV-1 replication and are in agreement with active iron import at peak productive infection in our primary CD4⁺ T-cell model (between 7-9dpi, please refer to the answer to question n9 for further details).

Reply Figure 14. Effect of iron during productive infection. Percentage of intracellular gag p24⁺ CD4⁺ T-cells infected with three different concentrations of HIV-1_{NL4-3} (0.2, 2 and 20 ng p24/million CD4⁺ T-cells). Cells were left untreated or treated for 48hr with the iron donor ferric nitriloacetate (Fe-NTA) at 150μM concentration. Intracellular p24 was assayed by flow cytometry at 5 and 9dpi (n=3; mean ± range).

12) Figure S5: why productively infected cells (GFP+) are not all positive for viral integration?

The Reviewer is correct in noticing this. It is a drawback inherent to dual color constructs, which could be due to various reasons such as recombination, transcriptional interference or lower content of Kousoubira Orange Dye (mKO2) produced under the control of the EF1α cellular promoter, compared to the LTR-driven GFP expression. Issues in this sense have been recently described in the literature (Battivelli et al. Elife 2018).

As the supplementary Figure containing this panel was criticized for different reasons by all Reviewers, it has now been removed and substituted by the results obtained by silencing PML in HIV-1 infected primary CD4⁺ T-cells (Reply Figure 8, *i.e.* Figure EV6 in the revised manuscript).

Reviewer 3

Shytaj et al. studied oxidative stress, iron transport and PML in primary CD4 T cells infected with HIV-1. They used a model of infection that extended for 14 days when HIV-1 latency started to become apparent. Although most cells died over the 14 days period, a few seemed to survive that contained an integrated yet transcriptionally silent provirus. Samples were analysed by RNAseq and proteomics to compare changes of expression of specific gene pathways (i.e. GO "cellular response to oxidative stress" or "PML pathway"). Overall, the oxidative stress response pathway, controlled by transcription factor Nrf2, was upregulated in productively infected cells but this tended to normalize at 14 days post infection. Similar results were observed in PBMCs from SIVmac infected macaques before and after antiretroviral therapy (ART). Analysis of a subset of 6 oxidative stress genes by qPCR supported the RNAseq data and infection with two different defective viruses demonstrated that the oxidative stress response was triggered only by viral replication. Iron homeostasis was also examined, showing an overall signature of enhanced intracellular iron uptake during productive infection. Iron controls PML stability and PML has been proposed to promote latency. Thus, the authors examined mRNA and protein levels of PML in their experimental setting. PML degraded during active virus replication but it stabilized at 14 days post-infection. The authors draw on these lines of evidence to argue that HIV-1 replication triggers an oxidative stress response and induce iron uptake in CD4 T cells, converging on the degradation of PML, which in turn may regulate entry into latency.

The study reports a series of interesting observations on the relationship between HIV-1 infection, the oxidative stress response and iron homeostasis, an area that is little investigated. The notion that cells more resistant to oxidative stress might go on to survive and become latently infected is intriguing.

We thank the Reviewer for highlighting the novelty of the study and for the in depth summary of the main conclusions and strengths of the work. In line with this comment, we think that Reply Figure 9 (*i.e. Figure 4A in the revised manuscript*) showing the effect of HMOX-1 knock down in infected cells, further strengthened the notion that antioxidant upregulation upon productive HIV-1 infection favors survival of infected cells.

Major concerns.

Unfortunately, the mechanistic connection with latency itself is tenuous and mostly unsupported by the data presented. Indeed, there is no clue that perturbing the oxidative stress response or iron homeostasis affects either the establishment of latency or exit from it. Some of the results seem to be overinterpreted and would benefit from clarification.

The Reviewer highlights a major potential issue of the manuscript. We believe that the new data can satisfactorily address this concern. In particular, Reply Figure 1C (*i.e. Figure 4G in the revised manuscript*) shows the effect of iron as a cofactor to enhance latency reactivation. Moreover, Reply Figure 11 (*i.e. Figures 3B-D in the revised manuscript*) shows increased oxidative stress (ROS content) in productively, but not latently infected cells. Reply Figure 15 (*i.e. Figure 3I in the revised manuscript*) displays the upregulation of antioxidant gene expression upon latency reactivation in J-Lat cells. Finally, Reply Figure 4A (*i.e. Figures 5E in the revised manuscript*) links reactivation from latency with PML degradation. We believe that this new set of evidence coherently connects oxidative stress, antioxidant responses, iron import and PML stability in all infection stages, be it productive, latent or upon latency reactivation.

Reply Figure 15. Upregulation of antioxidant gene expression upon HIV-1 reactivation from latency. Relative mRNA expression of Nrf2 downstream antioxidant targets in J-Lat 9.2 cells left untreated or treated for 24h with TPA. Data were obtained by qPCR, normalized using 18S as housekeeping control and then expressed as Log₂ fold mRNA expression in TPA-reactivated vs latent cells (calculated as in: Livak & Schmittgen, 2001). Data are expressed as mean±SEM and were analyzed by paired t-test. *** $P < 0.001$. Trx= thioredoxin; NQO1= NAD(P)H [quinone] dehydrogenase 1; HMOX-1= heme oxygenase 1; G6PD= glucose-6-phosphate dehydrogenase; GCLC= glutamate-cysteine ligase; TrxR1= thioredoxin reductase 1.

1) Figure 2A shows a modest upregulation of the oxidative response pathway (mock relative to infected), which appears to be stable until day 14. Figure 2B then shows progressive downregulation of the same pathway when comparing "productive vs latent" cells. However, here they perform the analysis up to day 9, before latency becomes apparent (Figure 1B-D). Furthermore, there is no way the authors can separate latently infected cells from uninfected cells from dying cells in the experimental setting described, even at day 14. In contrast to the data in primary human CD4 T cells, the data on simian PBMCs are clearer but of course again there is no way to know which cells were latently infected. Thus, the connection between the oxidative stress response and latency is weak. Instead, it is quite clear that productive HIV-1/SIV infection does trigger an oxidative stress response.

The Reviewer is right to point out that in the data presented in the original manuscript we did not isolate the antioxidant regulation in latently infected cells or in live cells. As mentioned in the replies to Reviewers n1 and n2 (questions 6 and 2, respectively) bystander effects are important contributors to oxidative stress. In the sections describing Nrf2 activation and PML decrease/reformation we tried to control for these effects by coupling single cell FISH and immunofluorescence (*Figures 3F and 6A-C in the revised manuscript*). As this approach is, to our knowledge, one of the few tools to tackle this question, we extended it to measure the content of ROS in productively and latently infected cells (Reply Figure 11, *i.e. Figures 3B-D in the revised manuscript*). This analysis confirmed that ROS content is upregulated in productively infected cells. Moreover, we showed upregulation of antioxidant responses upon HIV-1 latency reactivation (Reply Figure 15, *Figure 3I in the revised manuscript*), thus proving that latency is indeed associated with decreased expression of antioxidant responses. Finally, we validated all our main findings on primary CD4⁺ T-cells enriched for viability (Reply Figure 6, *i.e. Figure EV3C,D and EV7A in the revised manuscript*) to address this question and question 6 of Reviewer n1.

Finally, for Figure 2B, the follow-up did not stop at day 9, but rather expression and gene enrichment at day 14 were used as a benchmark to compare corresponding values during productive infection.

As our exposition was not clear and this comparison was criticized also by Reviewer n2 (question 1) we have removed it from the revised manuscript.

2) In general, the mapped, filtered and quantified RNAseq data should be provided in supplementary tables for all genes and for the subsets of genes shown in the Figures. This is critical to evaluate the specificity of the responses. Statistical inference should be applied to determine if the pathways in question are changed relative to other cellular pathways in the first place.

We included as supplementary tables the normalized expression values for all genes of the investigated pathways, both for the *in-vitro* and *ex-vivo* data (please see *Tables EV1, EV2, EV3 and EV4 in the revised manuscript*).

In addition, statistical analyses now include a test for enrichment of differentially expressed genes (DEGs) corrected for multiple testing using the entire list of pathways included in the repositories employed (unbiased approach). Apart from providing a q-value to assess significant enrichment of DEGs in the pathway of interest, this analysis ranks this q-value of enrichment in the pathway of interest and compares it to those of all pathways available in the repository (e.g. 4436 pathways in the Gene Ontology - Biological Process collection). This ranking is now presented in the revised manuscript as a percentile value and provides an estimate of the relative enrichment of DEGs in the pathway of interest as compared to all other cellular pathways available.

For example in the first subchapter of the results we write:

“To estimate the biological impact of this enrichment when compared to other cellular pathways modulated by HIV-1, we analyzed all significantly up-regulated genes in infection at 7 dpi using the entire Gene Ontology - Biological Process collection available at MSigDb (containing 4436 gene sets). The enrichment of the genes included in the antioxidant pathway considered retained its statistical significance after correction for multiple testing (q-val = 1.71E-08; 6th percentile of all 4436 gene sets ranked by q-values).”

Finally, p-values calculated using GSEA enrichment analyses for *ex-vivo* data have been kept in the revised version.

3) In Figure 3C, it is not clear what was used to normalize mRNA expression. GAPDH is unsuitable because its own expression changes upon oxidative stress (see for example Avery S. *Biochem. J*, 2011 434: 201).

We thank the Reviewer for pointing out this potentially confounding factor. In this original version of the manuscript we used GAPDH to normalize qPCR data, but in accordance with this comment, we now included a second housekeeping gene, 18S (Reply Figure 6B, Reply Figure 15 and Reply Figure 16, *i.e. Figures 1C, 3G, 3I, 5D and EV3B,D in the revised manuscript*). Where possible we showed normalizations with both housekeeping genes in the same figure. In some cases, to avoid overcrowding of the graphs, we added the data normalized using 18S in the main Figure and moved the ones normalized using GAPDH to the Supplementary Information. Overall, trends observed using 18S to normalize the data, were similar to those obtained using GAPDH.

Reply Figure 16. Normalization of mRNA levels using the housekeeping genes 18S and GAPDH. Panels A,B) Comparison of the relative mRNA level of HIV-1 gag (A) Nrf2-regulated antioxidant genes (B) or PML (C) normalized using GAPDH (A,C) or 18S (A-C) as housekeeping gene. Raw data were first normalized using the housekeeping control and then expressed fold change over 3dpi (A) or Log₂ fold mRNA expression in infected vs mock infected cells, calculated as in (Livak & Schmittgen, 2001). Data are displayed as mean±SEM of three donors and were analyzed by two way ANOVA followed by Tukey's post-test for multiple comparisons (B,C). * $P < 0.05$; ** $P < 0.01$; *** $P < 0.001$; **** $P < 0.0001$. Trx= thioredoxin; NQO1= NAD(P)H [quinone] dehydrogenase 1; HMOX-1= heme oxygenase 1; G6PD= glucose-6-phosphate dehydrogenase; GCLC= glutamate-cysteine ligase; TrxR1= thioredoxin reductase 1.

4) In Figure 4A, the iron homeostasis gene network in human cells has an opposite trend to simian PBMCs. This needs clarification. In panel B, it appears that the day 14 pi is not significant but this is relative to what? To the 9 days or to 3 days pi? There seem to be robust changes between 3 and 14 days pi.

In our view, the trend is not opposite, rather, both RNA-Seq analyses show enrichment of iron homeostasis genes, but in *in-vitro* infection the peak enrichment (compared to mock infected cells) is reached in the time point immediately before latency, while in simian PBMCs the enrichment is observed after ART. Our rationale for showing these analysis was to obtain an initial assessment on whether iron homeostasis can be perturbed by the infection. However, the iron homeostasis pathway includes genes that act by opposing mechanisms of import and export. Thus, our readouts to clarify the mechanism of iron regulation are those discriminating between iron import (e.g. Tfr1) level and export (SLC40A1). To account for the Reviewer's comment, and as the iron homeostasis gene set does not per se allow to draw firm conclusions, we have moved the analysis of this pathway to the supplementary information (Figure EV4B in the revised manuscript).

For the statistics, the dot size in the enrichment plots depicts the results of the Fisher test of significance. The Fisher test herein employed does not compare days between each other, but rather, for each given day, the enrichment of differentially expressed genes (DEGs) significantly up-regulated in infected vs matched mock-infected controls in the pathway of interest relative to the DEGs of all pathways.

5) Supplementary Figure 5D is not convincing and probably unnecessary.

We have removed the figure as per Reviewer's request.

To also address the questions of the other Reviewers on this topic (question 8 of Reviewer n1 and question 3 of Reviewer n2) we have substituted this figure with a new one (Reply Figure 8, i.e. Figure EV6 in the

revised manuscript) showing the effect of knock down of PML in primary CD4⁺ T-cells. Although the silencing of PML in this setting was not complete, the data showed that only treatment with FANA oligos effective in decreasing PML content was associated with increased proportion of p24⁺ CD4⁺ T-cells. For more details, please refer to Reply Figure 8.

6) A simple marker of oxidative stress is phosphorylated eIF2a. It should be examined during the course of infection

We thank the Reviewer for this suggestion. We have analyzed this marker in primary CD4⁺ T-cells (Reply Figure 17; *i.e.* *Figure EV3A in the revised manuscript*), including the time point 0dpi to also account for question 1 of Reviewer n2. Our results show that:

- The level of phosphorylated eIF2alpha is higher in HIV-1 infected as compared to mock infected CD4⁺ T-cells
- Absolute levels of eIF2alpha phosphorylation are higher when cells are more activated (that is, at 0dpi, closer to the initial stimulation with α -CD3/CD28 antibodies).

This is in line with the role of cellular activation in modulating stress factors and, in our view, confirms the validity and importance of using mock infected controls matched in terms of donor and time point for each analysis. More detail is also provided in answer to question n1 of Reviewer n2.

Reply Figure 17. Time course of the expression of eIF2 α protein (phosphorylated and non) in HIV-1 infected and mock infected primary CD4⁺ T-cells. Protein expression was assessed by western blot.

Thank you for submitting a revised version of your manuscript. It has now been seen by all original referees, who find that their main concerns have been addressed and are now broadly in favour of publication of the manuscript after a minor revision. There are also a few editorial issues that have to be addressed before I can extend formal acceptance of the manuscript.

REFeree REPORTS:

Referee #1:

In general, I find that the authors have done quite a good job in the revision. This considerable amount of data is more carefully presented and discussed. The interplay between HIV replication, oxidative stress, iron metabolism and PML interesting. The fact that most observations were made in primary cells strengthens the quality and the impact of this study.

Many of my comments have been addressed. I would nevertheless down-tone a few points in the presentation of the data:

- In the experiment where J-LAT activation by TPA induces NRF2 targets, a control with a non-infected T cell treated by TPA is needed.
- Did HMOX1 extinction alter redox status? This may be why it increases HIV1 replication.
- I would not show in main figures data that are not statistically significant (5B)
- Figure 1, and even perhaps 2, could be moved to supplementary.

Referee #2:

The authors made an extensive work and greatly improved the manuscript. The report still lacks nfr2 of Tfr1 knock-out/down to make a definitive link between iron metabolism, PML depletion and survival, but overall the data presented are of interest. The authors should be more moderate in their final model.

Minor Comments:

1. The authors should be more cautious while presenting their model (Figure 7i), as some links are still speculative.
3. Page 5: "Nrf2 showed that the nuclear content of Nrf2 was higher in cells actively producing the virus (Fig 3D)". "Fig 3D" should read "Fig 3F".
4. Page 6, it should be clearly stated that HMOX-1 silencing is performed in J-LAT.
5. Page 7: "(in-vitro data q-value for enriched genes at 9 dpi= 0.027; ex-vivo GSEA p-value= 0.035; Fig 4C)". The author refer to Fig EV4C instead of 4C.
6. Page 7: "In addition, the two iron transport proteins that could be detected in our proteomic analysis (transferrin receptor 1 (TfR1) and Phosphatidylinositol Binding Clathrin Assembly ProteinPICALM) showed a similar trend (Fig EV4C right panel)" There is no data specific of TfR1 and PICALM in figure EV4C
8. Page 7. The relevance of measuring Tfr1 internalization should be explained. If the authors want to use this data to assess Tfr1 activity, a quantification should be provided.
9. Page 7: the function of FTH-1 should be briefly explained.
10. Page 7: "Overall, the expression of these markers shows increased iron import". "suggests" is more appropriate than "shows"

11. Page 7: "iron import is increased during productive infection" the authors should stick on their data and conclude on Tfr1, FTH-1 and SLC40A1 expression rather than on iron import.

12. Page 7: "thus facilitating cell survival before latency establishment." "Possibly" would be more appropriate than "thus".

13. Page 8: "PML NBs have been previously described as possible restriction factors of HIV-1 We thus analyzed the proportion of productively infected cells in our primary CD4+ T-cell model where PML was silenced with the recently developed FANA antisense oligo technology (Souleimanian et al, 2012) Albeit full silencing could not be achieved in primary cells, only FANA oligos inducing partial depletion of PML were associated with higher proportion of p24+ T-cells in the culture (Fig EV6)." Several dots are missing in this paragraph.

14. Figure 7: The colocalization between PML and SUMO does not provide an evidence that PML is sumoylated. The conclusion (page 10) that HIV induces the sumoylation of PML should thus be tuned down and restricted to PML NBs. The authors may discuss the possible targets of SUMO (including PML) and how sumoylation addresses PML to the proteasome.

15. Figure EV6: The number of individual donors of primary CD4 T cell should be provided in addition to the number of biological replicates.

Referee #3:

Luca et al. present a revised version of their manuscript "Alterations of redox iron metabolism accompany development of HIV-1 latency". The authors have addressed my original concerns and overall the paper is now better. The link between oxidative stress, iron transport and PML turnover is clearer and more convincingly supported by the data. The manuscript does raise the intriguing question if latently infected cells have chronically elevated oxidative stress responses, which may be used as a potential biomarker. I particularly appreciated the authors use of FISH to detect viral DNA and distinguish infected from uninfected cells. I have two remaining issues:

1. The conclusions about the link between PML and latency would be strengthened if the authors tested the effect of the protease inhibitors on latency reactivation. The imaging and biochemical data shown in Figure 7 are convincing and should be linked functionally to HIV-1 latency. It seems a fairly straightforward experiment to conduct.

2. Eyeballing the RNAseq data in dataset 1, the difference in normalized gene expression between infected and mock treated cells were not so striking for most genes in the Iron Homeostasis Pathway or the PML Pathway. It would be helpful to show a heatmap for each pathway (including oxidative stress) for easy comparison. Also, please indicate if the numbers shown in the datasets are Log2.

3rd Revision - authors' response

1st Feb 2020

Please see next page.

Referee #1:

In general, I find that the authors have done quite a good job in the revision. This considerable amount of data is more carefully presented and discussed. The interplay between HIV replication, oxidative stress, iron metabolism and PML interesting. The fact that most observations were made in primary cells strengthens the quality and the impact of this study.

Many of my comments have been addressed. I would nevertheless down-tone a few points in the presentation of the data:

We thank the Reviewer for the encouraging and positive feedback.

- In the experiment where J-LAT activation by TPA induces NRF2 targets, a control with a non-infected T cell treated by TPA is needed.

We agree that this control is important to highlight the specific effect of HIV-1 expression and have now included it. The data show no significant change in the expression of the Nrf2 targets considered when uninfected Jurkat T-cells are treated with TPA.

Reply Figure 1. Relative mRNA expression of Nrf2 downstream antioxidant targets in Jurkat T-cells left untreated or treated for 24h with TPA. Data were normalized using 18S as housekeeping control and expressed as Log₂ fold mRNA expression in TPA treated vs untreated cells as in (Livak & Schmittgen, 2001). Data are shown as mean±SEM of 3 replicates. NQO1= NAD(P)H [quinone] dehydrogenase 1; HMOX-1= heme oxygenase 1; G6PD= glucose-6-phosphate dehydrogenase; GCLC= glutamate-cysteine ligase; TrxR1= thioredoxin reductase 1; Trx= thioredoxin.

- Did HMOX1 extinction alter redox status? This may be why it increases HIV1 replication.

The Reviewer is correct in noticing this. To knock down HMOX-1 (and Nrf2 in the revised version of the paper) we introduced siRNAs by electroporation of Jurkat T-cells. This procedure is known to induce oxidative stress (e.g. Bonnafous *et al.* Biochim Biophys Acta 1999) and renders difficult reliable quantification of ROS content, as also control cells electroporated with non targeting siRNAs will have significantly increased ROS. Considering this background ROS production, despite the technical limitations associated with this experiment, a specific contribution of HMOX-1 knock down beyond mere ROS generation is likely.

- I would not show in main figures data that are not statistically significant (5B)

We have moved this panel to the supplementary material.

- Figure 1, and even perhaps 2, could be moved to supplementary.

We have moved Figure 1 to the supplementary material.

Referee #2:

The authors made an extensive work and greatly improved the manuscript. The report still lacks *nfr2* or *Tfr1* knock-out/down to make a definitive link between iron metabolism, PML depletion and survival, but overall the data presented are of interest. The authors should be more moderate in their final model.

We thank the Reviewer for the supportive comments. We have performed experiments to knock down both *Nrf2* and *Tfr1* (using a similar experimental setup to that previously done with HMOX-1). For *Nrf2* we obtained acceptable knock down efficiency and included the data in the revised manuscript (**Reply Figure 2**; *i.e.* **Fig. EV3B** and **EV3C** in the Revised manuscript). Specifically, our results show increased transcription of HIV-1 upon *Nrf2* KD (in line with the previously demonstrated role of *Nrf2* in decreasing HIV-1 mRNA expression [H.-S. Zhang et al. 2009]).

Reply Figure 2. (A,B) *Nrf2* (A) and *gag* (B) mRNA expression in HIV-1 infected J-Tag cells transfected with non targeting siRNA (NC) or siRNAs targeting the *Nrf2* gene. Cells were infected 24hr post-transfection and assayed for mRNA expression by qPCR 48hr post-infection. Data (mean±SEM; n=4 technical replicates) were normalized using 18S as housekeeping gene, expressed as fold change in siRNA *Nrf2* treated cells vs NC control and analyzed by unpaired *t*-test.

Our attempts to knock down *Tfr1*, instead, were not successful (**Reply Figure 3**). This is likely due to the requirement of *Tfr1* expression by proliferating cells (both healthy and cancer cells, *e.g.* Jian *et al.* Cancer Research 2008), in line with their increased iron consumption. In general, we could not find any published results showing effective knock down of *Tfr1* in proliferating T-cell lines.

Reply Figure 3. Tfr1 mRNA expression in J-Tag cells transfected with non targeting siRNA or siRNAs targeting the *Tfr1* mRNA. Cells were collected 72 hours post-transfection and analyzed by qPCR. Data (mean±SEM; n=3) were normalized using the NC control.

Minor Comments:

1. The authors should be more cautious while presenting their model (Figure 7i), as some links are still speculative.

In line with this comment, we have modified the Figure (please see figure 6i in the Revised manuscript). In particular, changes of iron content in cells are now not shown, and the effect of ROS or, possibly, iron on PML is not anymore linked to direct cysteine oxidation. The Fenton reaction was retained as our data show increased ROS content during productive infection (please see Figure 2B-D in the Revised manuscript). Therefore, in an oxidized intracellular environment, this chemical reaction is a necessary consequence of ROS and iron interaction.

3. Page 5: "Nrf2 showed that the nuclear content of Nrf2 was higher in cells actively producing the virus (Fig 3D)". "Fig 3D" should read "Fig 3F".

We have corrected the mistake.

4. Page 6, it should be clearly stated that HMOX-1 silencing is performed in J-LAT.

We have now specified that HMOX-1 silencing was done in a cell line (J-Tag cells infected with *wild type* HIV-1). This cell line was selected over J-Lats because:

- a) It allowed using a *wild type* virus for the infection.
- b) Electroporating J-Lat cells leads to spontaneous and robust GFP production.

5. Page 7: "(in-vitro data q-value for enriched genes at 9 dpi= 0.027; ex-vivo GSEA p-value= 0.035; Fig 4C)". The author refer to Fig EV4C instead of 4C.

We apologize for the confusion, the correct mention would have been to Figure 4B. As one panel of former Figure 4B has now been moved to the supplementary material (in line with the suggestion of Reviewer n1) we have updated the sentence accordingly (in-vitro data q-value for enriched genes at 9 dpi= 0.027; ex-vivo GSEA p-value= 0.035; **Fig 3B,C and Fig EV4F**).

6. Page 7: "In addition, the two iron transport proteins that could be detected in our proteomic analysis (transferrin receptor 1 (TfR1) and Phosphatidylinositol Binding Clathrin Assembly ProteinPICALM) showed a similar trend (Fig EV4C right panel)" There is no data specific of TfR1 and PICALM in figure EV4C

The boxplots depicted in Figure EV4G of the Revised version of the manuscript display the median and interquartile range of the relative (HIV-1 infected vs mock infected) TfR1 and PICALM protein expression. This is due to the fact that only these two proteins, among the members of the *GO iron ion import* pathway, could be detected in our proteomic array.

In order to provide further information on the expression of TfR1 and PICALM in proteomics, we included a table (Table EV4 in the Revised manuscript), which separately shows, for each time point, the Log₂ fold change expression of TfR1 and PICALM in HIV-1 infected as compared to mock infected cells.

8. Page 7. The relevance of measuring TfR1 internalization should be explained. If the authors want to use this data to assess TfR1 activity, a quantification should be provided.

Our purpose in showing TfR1 subcellular localization was to prove that increased TfR1 expression in infected cells was also accompanied by TfR1 internalization (and thus that highly expressed TfR1 in infected cells was functional). We agree with the Reviewer that claims of increased TfR1 activity would require a quantification, but this is not easy to apply to STED images.

As this panel was simply a support to our main readouts of TfR1 expression (RNA-Seq, proteomic, immunofluorescence and FACS) we have now removed it to avoid potential data overinterpretation.

9. Page 7: the function of FTH-1 should be briefly explained.

We have included a clarification of the function of FTH-1, the paragraph now reads:

“Apart from TfR1 upregulation, productively infected cells were characterized by significantly decreased expression of FTH-1 (Fig 3E). As FTH-1 binds iron converting it to a non reactive form (Muckenthaler et al, 2017), its downregulation suggests reduced intracellular storage. This reduced storage, however, was also accompanied by lower expression of SLC40A1, suggesting decreased iron export (Fig EV4F).”

10. Page 7: "Overall, the expression of these markers shows increased iron import". "suggests" is more appropriate than "shows"

We have modified this sentence according to the Reviewer's suggestion.

11. Page 7: "iron import is increased during productive infection" the authors should stick on their data and conclude on TfR1, FTH-1 and SLC40A1 expression rather than on iron import.

12. Page 7: "thus facilitating cell survival before latency establishment." "Possibly" would be more appropriate than "thus".

In line with these two comments, we have modified this conclusion as follows:

*“Overall, these data show that iron can **be exploited** as a co-factor by replicating HIV-1 and that iron import **capacity** is increased during productive infection, **possibly** facilitating cell survival before latency establishment.”*

We hope that this can reflect fairly the findings described in the subchapter while providing a concise summary which is more suited for the final sentence.

13. Page 8: "PML NBs have been previously described as possible restriction factors of HIV-1 We thus analyzed the proportion of productively infected cells in our primary CD4+ T-cell model where PML was silenced with the recently developed FANA antisense oligo technology (Souleimanian et al, 2012) Albeit full silencing could not be achieved in primary cells, only FANA oligos inducing partial depletion of PML were associated with higher proportion of p24+ T-cells in the culture (Fig EV6)." Several dots are missing in this paragraph.

We thank the Reviewer for noticing this, and we have now corrected this sentence.

14. Figure 7: The colocalization between PML and SUMO does not provide an evidence that PML is sumoylated. The conclusion (page 10) that HIV induces the sumoylation of PML should thus be tuned down and restricted to PML NBs. The authors may discuss the possible targets of SUMO (including PML) and how sumoylation addresses PML to the proteasome.

We agree that our experiments can formally prove only increased sumoylation of PML nuclear bodies. We have now corrected the text to restrict each mention of sumoylation to nuclear bodies.

15. Figure EV6: The number of individual donors of primary CD4 T cell should be provided in addition to the number of biological replicates.

For all experiments on primary cells, each biological replicate indicates a different donor. We have used the term biological replicates to conform with the Journal guidelines.

Referee #3:

Luca et al. present a revised version of their manuscript "Alterations of redox iron metabolism accompany development of HIV-1 latency". The authors have addressed my original concerns and overall the paper is now better. The link between oxidative stress, iron transport and PML turnover is clearer and more convincingly supported by the data. The manuscript does raise the intriguing question if latently infected cells have chronically elevated oxidative stress responses, which may be used as a potential biomarker. I particularly appreciated the authors use of FISH to detect viral DNA and distinguish infected from uninfected cells. I have two remaining issues:

We thank the Reviewer for the positive comments and for highlighting the relevance of single cell FISH-IF experiments.

1. The conclusions about the link between PML and latency would be strengthened if the authors tested the effect of the protease inhibitors on latency reactivation. The imaging and biochemical data shown in Figure 7 are convincing and should be linked functionally to HIV-1 latency. It seems are fairly straightforward experiment to conduct.

We agree that this is an important experiment to link proteasome function and latency. To answer this question we pretreated J-Lat 9.2 cells with the proteasome inhibitor MG132 and then induced HIV-1 reactivation with TPA (**Reply Figure 4**; i.e. **Fig. EV8D,E** in the Revised manuscript). In order to limit possible biases of non-specific effects induced by proteasome inhibition on GFP turnover, we analyzed both GFP expression (by FACS) and *gag* transcription (by real-time PCR). Both analyses showed that proteasome inhibition decreased the extent of TPA-induced reactivation. Despite some limitations, due to the numerous proteins that are stabilized by proteasome inhibition, we have included this experiment in the supplementary material (**Fig EV8D and E** in the Revised manuscript).

Reply Figure 4. Effect of proteasome inhibition on HIV-1 latency reactivation. (A, B) GFP protein (A) or relative *gag* mRNA (B) expression in J-Lat 9.2 cells in which HIV-1 was latent or reactivated with TPA for 24 hours. Before TPA reactivation, cells were left untreated or pre-treated for 4 hours with MG132. GFP and *gag* expression were measured by flow cytometry and real-time PCR, respectively. Real-time PCR data were normalized using 18S as housekeeping gene. After normalization data were expressed as fold mRNA expression over untreated as in (Livak & Schmittgen, 2001). Data were analyzed using the non parametric Friedman test. * $P < 0.05$; ** $P < 0.01$.

2. Eyeballing the RNAseq data in dataset 1, the difference in normalized gene expression between infected and mock treated cells were not so striking for most genes in the Iron Homeostasis Pathway or the PML Pathway. It would be helpful to show a heatmap for each pathway (including oxidative stress) for easy comparison. Also, please indicate if the numbers shown in the datasets are Log₂.

The values included in the EV tables datasets are Log₂. We have now specified this in the captions (please see captions of **Tables EV1, 2, 3 and 5** in the Revised manuscript).

As per Reviewer's request, we have now included heatmaps of all datasets for infected and mock infected cells (please see **Fig EV2A, EV4C, EV6A and Fig 3C** in the Revised manuscript).

4th Editorial Decision

10th Feb 2020

Thank you for addressing the issues indicated in the previous decision letter and for submitting source data for your manuscript. The original Fig. EV3A image is fine to use. Unfortunately, while going through the source data I noticed several issues that have to be fixed before I can formally accept the manuscript.

1) Most importantly, there are several cases of beautification in the actin blots for figures 2H, EV3A and EV4J, where background is either removed or added. Please check and adjust main and EV figures to faithfully reflect the original experiment.

2) Minor source data issues, please check:

- In source data for Fig. 5E the bounding box is not in the right place

- Source data for EV4J SD is mislabelled as for EV4F

- Source data for Appendix Fog S2 is mislabelled as for Fig EV7

- Source data for Appendix Fog S3 is mislabelled as for Fig EV8

3) Thank you for submitting the .docx file of manuscript text. The file has now been checked by our data editor Vivian Killet. Please check her comments in the figure legend section and add the requested information.

Please let me know if you have any questions about these remaining issues. We will re-check the updated figures upon submission to make sure that everything is in order before we forward the files to production.

4th Revision - authors' response

11th Feb 2020

Thank you for the update on our manuscript. We have corrected the minor issues of the source data and added the information requested by V. Killet.

Regarding the actin signals of the blots mentioned, we would like to clarify better the procedure that we followed to prepare all images (both Figures and Source data).

The Source data have been prepared using pictures to which the "overlay marker image" function had been applied, whenever this was possible. This function is commonly available on the instrument employed, the "Intas ECL Chemostar Imager" and allows to see the actual membrane in the background, for example how it was cut. We reasoned that this would be useful to know, especially in experiments including membrane stripping (such as the experiment of Figure 2H).

The panels for the Figures have been prepared using the same picture, but without applying the "overlay marker image" function. Thus, only the signal generated by the ECL reaction is present in these images.

Typically, there is no difference between the figure with or without marker overlay. However, if the physical characteristics of the membrane should have some peculiarity (e.g. wet membrane or plastic membrane holder in Figures EV3A and EV4J, cut membrane in Figure 2H) this will not be acquired in the non-overlaid images. This is not, however, a result of beautification, but a consequence of the fact that the instrument only acquires the signal produced by the ECL (the only "true" western blot signal).

Thus, in order to make sure that we can correctly address your question, we would like to submit to your attention two/three separate raw data images, at different exposures, for each of the blots mentioned in your question. For each image we provide both the "overlay" and "standard" version. When the brightness has been reduced to address your previous question (please see reply to your comment n9 in Revision n2), we have indicated this in the file name.

To the best of our understanding, each of the separate images provided fully reflects the whole western blot signal and result of the experiment. We would appreciate your guidance on which images to select to comply with the EMBO guidelines

5th Editorial Decision

14th Feb 2020

Thank you for implementing the final edits in your manuscript and providing additional source data together with the explanation on the detected source data discrepancies. We had so far not encountered this type of overlay in previous publications. I appreciate your explanation, and everything looks in order now. I am pleased to inform you that your manuscript has been accepted for publication.

Corresponding Author Name:

Journal Submitted to:

Manuscript Number: